# Generalization Analysis for Supervised Contrastive Representation Learning under Non-IID Settings

**Nong Minh Hieu** [1]   **Antoine Ledent** [1]

## Abstract

Contrastive Representation Learning (CRL) has achieved impressive success in various domains in recent years. Nevertheless, the theoretical understanding of the generalization behavior of CRL has remained limited. Moreover, to the best of our knowledge, the current literature only analyzes generalization bounds under the assumption that the data tuples used for contrastive learning are independently and identically distributed. However, in practice, we are often limited to a fixed pool of reusable labeled data points, making it inevitable to recycle data across tuples to create sufficiently large datasets. Therefore, the tuple-wise independence condition imposed by previous works is invalidated. In this paper, we provide a generalization analysis for the CRL framework under non-$i.i.d.$ settings that adheres to practice more realistically. Drawing inspiration from the literature on U-statistics, we derive generalization bounds which indicate that the required number of samples in each class scales as the logarithm of the covering number of the class of learnable feature representations associated to that class. Next, we apply our main results to derive excess risk bounds for common function classes such as linear maps and neural networks.

## 1. Introduction

The performance of many machine learning (ML) algorithms is often significantly influenced by the quality of data representations. For example, in multi-class classification problems, it is typically desirable for data points within the same class to exhibit proximity, reflecting intra-class compactness, while maintaining sufficient separation from data points of other classes, thereby ensuring inter-class separability under an appropriate distance metric. Motivated by this intuition, CRL has been utilized in numerous ML tasks as a pre-processing step to improve data representations. Essentially, the goal of CRL is to learn a representation function $f : \mathcal{X} \to \mathbb{R}^d$ that maps data from an input space $\mathcal{X}$ to a representation space (potentially of lower dimension). The underlying strategy for representation learning is by pulling together similar pairs of data points $(x, x^+)$ while pushing apart dissimilar pairs $(x, x_i^-)$ via minimizing a certain contrastive loss function $\ell$. For instance, given an input tuple $(x, x^+, \{x_i^-\}_{i=1}^k)$ where $k$ is the number of negative samples, the logistic loss (otherwise known as N-pair loss (Sohn, 2016)) $\ell : \mathbb{R}^k \to \mathbb{R}_+$ is defined as follows:

$$\ell(\mathbf{v}) = \log\left(1 + \sum_{i=1}^{k} \exp(-\mathbf{v}_i)\right), \tag{1}$$
$$\text{where } \mathbf{v}_i = f(x)^\top f(x^+) - f(x)^\top f(x_i^-).$$

The application of CRL spans a wide variety of ML disciplines, including computer vision (Chen et al., 2020; He et al., 2019; Gidaris et al., 2018), graph representation learning (Hassani & Khasahmadi, 2020; Zhu et al., 2020; Velickovic et al., 2019), natural language processing (Gao et al., 2021; Zhang et al., 2021; Reimers & Gurevych, 2021) and time series forecasting (Lee et al., 2024; Yang et al., 2022; Nie1 et al., 2023; Eldele et al., 2021). The increasing empirical success naturally inspires multiple theoretical works (Arora et al., 2019; Lei et al., 2023; Hieu et al., 2024) dedicated to studying the generalization behavior of the CRL framework. However, the analyses performed by previous works have only explored the $i.i.d.$ settings where input tuples are independently and identically distributed (cf. Section 3.2). In reality, training datasets are often limited to fixed pools of labeled samples (Sohn, 2016; van den Oord et al., 2019; Khosla et al., 2020). Therefore, in order to create sufficiently large datasets for contrastive learning, it is a standard practice to "recycle" data points across input tuples (cf. Section 3.3). Effectively, it is possible for the same data points to appear multiple times in different input tuples, invalidating the independence criterion. As a result, previous results on generalization bounds for CRL might not comply with most practical use cases where data is limited.

---

[1]School of Computing and Information Systems, Singapore Management University. Correspondence to: Nong Minh Hieu <mh.nong.2024@phdcs.smu.edu.sg>.

*Proceedings of the 42nd International Conference on Machine Learning*, Vancouver, Canada. PMLR 267, 2025. Copyright 2025 by the author(s).

In this work, we propose a revised theoretical model and establish generalization bounds for the CRL framework under non-$i.i.d.$ conditions, adhering more closely to the standard practice compared to existing results. Our key contributions are listed as follows:

1. **A revised theoretical framework for CRL**: Previous analyses performed for CRL mostly rely on the theoretical model proposed by Arora et al. (2019), which requires an $i.i.d.$ assumption across input tuples used for empirical risk minimization (ERM). However, we argue that this approach does not abide by the set-ups of most practical use cases. As an alternative, we propose a modified framework where ERM is performed using a small subset of input tuples assembled from a fixed pool of labeled data points (cf. Section 3.3).

2. **Generalization bound for the empirical minimizer of U-Statistics**: We proposed a U-Statistics formulation (cf. Equation 10) for the population unsupervised risk (cf. Definition 3.1). We proved that, with high probability and when the dataset is well-balanced, the generalization gap between the performance of the empirical U-Statistics minimizer and the Bayes risk grows in the order of $\mathcal{O}(1/\sqrt{\widetilde{N}})$ where $\widetilde{N}$ scales like $\mathcal{O}(N/|\mathcal{C}|)$ for small values of $k$ and $\mathcal{O}(N/k)$ when $k$ is large (cf. Theorem 5.1 and Theorem 5.2).

3. **Generalization bound for the empirical minimizer of a sub-sampled risk**: We derive a generalization bound for the empirical minimizer of the "sub-sampled" risk, i.e., the average of some contrastive loss evaluated over a small subset of non-$i.i.d.$ tuples (cf. Equation 7). We prove that, with high probability, the gap between the performance of the empirical sub-sampled risk minimizer and the Bayes risk scales in the order of $\mathcal{O}(1/\sqrt{\widetilde{N}} + 1/\sqrt{M})$ where $M$ is the number of tuples subsampled for training.

4. **Applications of the theoretical results**: We apply our results to obtain bounds for common classes of representation functions such as linear maps and deep neural networks.

## 2. Related Work

**Empirical minimization of U-Statistics**: U-Statistics were first introduced by Hoeffding (1948) as a class of statistics to estimate parameters via the use of symmetric kernels. The concentration properties of U-Statistics was later rigorously studied by Gine & Zinn (1984) and Arcones & Gine (1993). These works proposed concentration inequalities and limit theorems for U-Processes, setting an concrete foundation for subsequent works in statistical learning theory. In generalization analyses where loss functions rely on two or more samples in the dataset, formulations of U-Statistics are often used to create unbiased estimators for the population risk. In such cases, the empirical risk minimizer is obtained via the minimization of U-Statistics. Techniques in U-Statistics have been applied in various analyses for generalization performance of common ML tasks, including bipartite ranking (Clémençon et al., 2008), metric and similarity learning (Cao et al., 2016; Jin et al., 2009), pairwise learning (Lei et al., 2020; 2018), clustering (Li & Liu, 2021) and semi-supervised learning (HaoChen et al., 2021).

**Theoretical analysis of CRL**: In the seminal work of Arora et al. (2019), the foundation for the theoretical understanding of the CRL framework was initially established. Their main contributions are threefold: (1) A formal definition of the population unsupervised risk and derivation of bounds on the excess unsupervised risk, (2) a rigorous analysis of the class-collision phenomenon, which shows how the repetition of anchor-positive classes among the negatives can distort learning and finally, (3) an analysis of the relationship between the unsupervised risk and the downstream classification risk. While their theoretical contribution has set a strong foundation and has been rigorously utilized by further works, the bounds derived by Arora et al. (2019) are limited by a strong dependency of $\mathcal{O}(\sqrt{k})$ on the number of negative samples. This dependency was later improved to at most logarithmic by Lei et al. (2023) using more advanced complexity measures like worst-case Rademacher complexity, fat-shattering dimension and arguments related to $\ell^{\infty}$-Lipschitzness of loss functions inspired by prior works in multi-class classification (Lei et al., 2019; Wu et al., 2021; Mustafa et al., 2021). The theoretical model designed by Arora et al. (2019) was later extended to results in other regimes of CRL such as adversarial contrastive learning (Zou & Liu, 2023; Ghanooni et al., 2024), PAC-Bayes analysis (Nozawa et al., 2020), de-biased CRL (Chuang et al., 2020) and CRL using DNNs (Alves & Ledent, 2024; Hieu et al., 2024). Notably, we emphasize that all of the aforementioned works target learning setups with $i.i.d.$ tuples.

**Other lines of research in CRL**: Other directions in the theoretical analysis of CRL have also been explored. For example, in Wang et al. (2022), the effect of data augmentation was studied using augmentation overlap theory. The authors suggested that aggressive augmentation can implicitly enhance intra-class compactness. In Huang et al. (2023), the effect of unsupervised risk on downstream classification was investigated by deriving bounds for the error rate of nearest-neighbor classifiers built on top of learned representations using misalignment probability. In Bao et al. (2022), the authors showed that the unsupervised risk is a surrogate of the supervised risk by deriving upper and lower bounds for the surrogate gap, which is the difference between the supervised and unsupervised risk. Further efforts for understanding the effect of representation quality to downstream

tasks have also been made (Arora et al., 2019; Chuang et al., 2020; Li & Liu, 2021; Bao et al., 2022). Crucially, these works primarily focus on studying the influence of the *population* unsupervised contrastive risk on the *population* downstream classification risk. On the other hand, the analysis in this paper targets generalization bounds on *excess* contrastive risk. Aside from the works mentioned above, another line of research studies the effect of *negative sampling* on the supervised downstream task. For example, in Ash et al. (2022), the authors show that while increasing the number of negatives can be initially advantageous, too many negative samples can potentially hinder the performance of downstream classifiers. The work of Awasthi et al. (2022) builds upon this by showing that the decline of downstream performance as number of negatives increases can be theoretically alleviated under certain situations (e.g., when class distribution is balanced). Despite earlier efforts, empirical experiments tend to suggest that amount of negatives usually correlates positively with downstream performance. In Nozawa & Sato (2021), the authors proposed a novel framework based on the coupon collector's problem that bridges the gap between theory and practice, showing that as the number of negatives increases, the unsupervised risk is more likely to contain the supervised risk. We also note that all of the results in this direction concern population-level risks.

## 3. Problem Formulation

### 3.1. Problem Set-up

We denote $\mathcal{X}$ as the space of input vectors and $\mathcal{C}$ be the finite set of all classes. Suppose that $\rho$ is the discrete probability measure over $\mathcal{C}$ (for all $c \in \mathcal{C}$, $\rho(c)$ denotes that occurrence probability of class $c$). For any class $c \in \mathcal{C}$, we denote $\mathcal{D}_c$ as the class-conditional distribution of input vectors over $\mathcal{X}$ given that the vectors belong to class $c$. On the other hand, we define $\bar{\mathcal{D}}_c$ as the distribution of input vectors in $\mathcal{X}$ given that the vectors do not belong to class $c$. Specifically, $\bar{\mathcal{D}}_c$ is defined as follows:

$$\bar{\mathcal{D}}_c(x) = \frac{\sum_{z \in \mathcal{C}, z \neq c} \rho(z) \mathcal{D}_z(x)}{1 - \rho(c)}, \forall x \in \mathcal{X}. \quad (2)$$

Basically, for $x \in \mathcal{X}$, $\bar{\mathcal{D}}_c(x)$ quantifies the probability that $x$ does not belong to class $c$. Let $\mathcal{F}$ denote the class of representation functions $f : \mathcal{X} \to \mathbb{R}^d$ for $d \geq 1$. For $f \in \mathcal{F}$, we denote $\mathrm{L}_{\mathrm{un}}(f)$ as the (population) unsupervised risk of $f$ (cf. Definition 3.1). In this work, we are interested in bounding the excess risk of the form $\mathrm{L}_{\mathrm{un}}(\hat{f}) - \inf_{f \in \mathcal{F}} \mathrm{L}_{\mathrm{un}}(f)$ where $\hat{f}$ is a minimizer of an empirical risk assessed on some dataset.

### 3.2. A Theoretical Framework under IID Settings

We start by revisiting the key definitions from the theoretical framework proposed by Arora et al. (2019).

**Definition 3.1** (Unsupervised Risk). Let $\ell : \mathbb{R}^k \to \mathbb{R}_+$ be an unsupervised loss function (E.g., Hinge or Logistic losses) where $k$ is the number of negative examples and $f : \mathcal{X} \to \mathbb{R}^d$ be a representation function. The population unsupervised risk for $f$ is defined as follows:

$$\mathrm{L}_{\mathrm{un}}(f) = \qquad\qquad\qquad (3)$$

$$\underbrace{\mathbb{E}_{c \sim \rho} \mathbb{E}_{\substack{\mathbf{x}, \mathbf{x}^+ \sim \mathcal{D}_c^2 \\ \mathbf{x}_i^- \sim \bar{\mathcal{D}}_c^k}} \left[ \ell\left( \left\{ f(\mathbf{x})^\top \left[ f(\mathbf{x}^+) - f(\mathbf{x}_i^-) \right] \right\}_{i=1}^k \right) \right]}_{\mathrm{L}_{\mathrm{un}}(f|c)}.$$

By the law of total expectation, we can decompose the population unsupervised risk as the weighted sum of the class-specific unsupervised risks $\mathrm{L}_{\mathrm{un}}(f|c)$ as follows:

$$\mathrm{L}_{\mathrm{un}}(f) = \sum_{c \in \mathcal{C}} \rho(c) \mathrm{L}_{\mathrm{un}}(f|c). \quad (4)$$

**Remark 3.2.** We note that the risk in Definition 3.1 is slightly different from Arora et al. (2019). In the latter, the negatives are drawn from $\bar{\mathcal{D}} = \sum_{c \in \mathcal{C}} \rho(c) \mathcal{D}_c$, which allows the positive class to re-appear among the negatives, effectively allowing class-collision.

A natural choice of algorithm for determining a representation function with low expected unsupervised risk is through empirical risk minimization (ERM). In the analyses done by Arora et al. (2019); Lei et al. (2023); Hieu et al. (2024), the empirical risk minimizer is identified via an ERM approach. Specifically, let $S = \left\{ (\mathbf{x}_j, \mathbf{x}_j^+, \mathbf{x}_{j1}^-, \ldots, \mathbf{x}_{jk}^-) \right\}_{j=1}^n$ be a dataset of $n$ independently and identically (*i.i.d.*) drawn input tuples from an unknown distribution. The empirical risk minimizer is then defined as $\hat{f}_n = \arg\min_{f \in \mathcal{F}} \hat{\mathrm{L}}_{\mathrm{un}}(f)$ where $\hat{\mathrm{L}}_{\mathrm{un}}(f)$ is defined as:

$$\hat{\mathrm{L}}_{\mathrm{un}}(f) = \frac{1}{n} \sum_{j=1}^n \ell\left( \left\{ f(\mathbf{x}_j)^\top \left[ f(\mathbf{x}_j^+) - f(\mathbf{x}_{ji}^-) \right] \right\}_{i=1}^k \right), \quad (5)$$

In line with most of the learning theory literature, the generalization gap $\mathrm{L}_{\mathrm{un}}(\hat{f}_n) - \inf_{f \in \mathcal{F}} \mathrm{L}_{\mathrm{un}}(f)$ can be conveniently upper bounded via controlling the Rademacher complexity of the loss class $\ell \circ \mathcal{F}$:

$$\ell \circ \mathcal{F} = \left\{ (\mathbf{x}, \mathbf{x}^+, \mathbf{x}_1^-, \ldots, \mathbf{x}_k^-) \mapsto \right.$$
$$\left. \ell\left( \left\{ f(\mathbf{x})^\top \left[ f(\mathbf{x}^+) - f(\mathbf{x}_i^-) \right] \right\}_{i=1}^k \right) : f \in \mathcal{F} \right\}. \quad (6)$$

However, Rademacher complexity based bounds, which arise from the symmetrization trick, are only immediately applicable under the assumption that the tuples in $S$ are all independent. Under regimes where tuples are sampled in a way that violates this assumption, more nuanced approaches are required before sample complexity bounds can be derived.

## 3.3. A Revised Framework under non-IID Settings

An issue with the framework described in Section 3.2 is that $S$ is defined as a dataset sampled $i.i.d.$ from a distribution of tuples. However, in practice, we do not observe whole input tuples independently due to the cost of data collection. Instead, training of representation functions is often limited to a fixed pool $S = \left\{ (\mathbf{x}_j, \mathbf{y}_j) \right\}_{j=1}^{N} \subset \mathcal{X} \times \mathcal{C}$ of input vectors and their corresponding labels, which are drawn $i.i.d.$ from a joint distribution over $\mathcal{X}$ and $\mathcal{C}$ (for each $c \in \mathcal{C}$, we denote $N_c^+$ as the number of samples that belong to class $c$, i.e., $\sum_{c \in \mathcal{C}} N_c^+ = N$). One possible alternative is to evaluate the empirical risk over all possible tuples that one can create from the dataset $S$. Unfortunately, this approach is computationally expensive because as the number of labeled samples N increases, the number of all possible tuples grows in the order of $\mathcal{O}\left[ \sum_{c \in \mathcal{C}} (N_c^+)^2 \cdot (N - N_c^+)^k \right]$[1]. Alternatively, a more reasonable approach is to use only a small subset of valid tuples. Let $\mathcal{T}_{\text{sub}}$ denote the subset of tuples that we use for training, the procedure for sub-sampling input tuples to train representation functions can be described as follows:

1. For $1 \leqslant j \leqslant M$, collect M tuples of the form $(\mathbf{x}_j, \mathbf{x}_j^+, \mathbf{x}_{j1}^-, \ldots, \mathbf{x}_{jk}^-)$ as follows:

    (i) Select a class $c \in \mathcal{C}$ with probability $\widehat{\rho}(c) = \frac{N_c^+}{N}$.
    (ii) Select $\mathbf{x}_j, \mathbf{x}_j^+$ uniformly (without replacement) among samples belonging to class $c$. We call these two data points the "anchor-positive pair".
    (iii) Select $\mathbf{x}_{j1}^-, \ldots, \mathbf{x}_{jk}^-$ uniformly (without replacement) among samples belonging to classes other than $c$. We call these "negative samples".
    (iv) Add the tuple $(\mathbf{x}_j, \mathbf{x}_j^+, \mathbf{x}_{j1}^-, \ldots, \mathbf{x}_{jk}^-)$ to $\mathcal{T}_{\text{sub}}$.

2. Finally, repeat the above steps M times independently.

Using the collected set of tuples $\mathcal{T}_{\text{sub}}$ described above, the empirical risk evaluated for every representation function $f \in \mathcal{F}$ is computed as:

$$\widehat{\mathcal{L}}(f; \mathcal{T}_{\text{sub}}) = \tag{7}$$
$$\frac{1}{M} \sum_{j=1}^{M} \ell\left( \left\{ f(\mathbf{x}_j)^\top \left[ f(\mathbf{x}_j^+) - f(\mathbf{x}_{ji}^-) \right] \right\}_{i=1}^{k} \right),$$

which we refer to as "sub-sampled empirical risk" since it is only evaluated on a small subset of valid tuples. In this revised framework, we are interested in the performance of the sub-sampled empirical risk minimizer, which we define as $\widehat{f}_{\text{sub}} = \arg\min_{f \in \mathcal{F}} \widehat{\mathcal{L}}(f; \mathcal{T}_{\text{sub}})$.

---

[1]For a given class $c$, a valid tuple is chosen by picking two samples from class $c$ and $k$ samples from other classes. Hence, the total number of tuples possibly formed for each class is $2 \binom{N_c^+}{2} \binom{N - N_c^+}{k} \in \mathcal{O}\left[ (N_c^+)^2 \cdot (N - N_c^+)^k \right]$.

## 4. Proof Strategy

As mentioned in the previous section, we are interested in how well the empirical sub-sampled risk minimizer $\widehat{f}_{\text{sub}}$ generalizes to testing data. In particular, we are concerned with bounding the excess risk $L_{\text{un}}(\widehat{f}_{\text{sub}}) - \inf_{f \in \mathcal{F}} L_{\text{un}}(f)$. However, as noted in Section 3.3, this is not immediately straightforward because the set of sub-sampled tuples $\mathcal{T}_{\text{sub}}$ are not $i.i.d.$ Therefore, we first formulate an (asymptotically) unbiased estimator (U-Statistics), denoted as $\mathcal{U}_N(f)$, for the population unsupervised risk $L_{\text{un}}(f)$. Then, we can decompose the generalization gap as (cf. Theorem E.1):

$$\frac{1}{2} \left[ L_{\text{un}}(\widehat{f}_{\text{sub}}) - \inf_{f \in \mathcal{F}} L_{\text{un}}(f) \right] \leqslant$$
$$\sup_{f \in \mathcal{F}} \left| \widehat{\mathcal{L}}(f; \mathcal{T}_{\text{sub}}) - \mathcal{U}_N(f) \right| + \sup_{f \in \mathcal{F}} \left| \mathcal{U}_N(f) - L_{\text{un}}(f) \right|.$$

As a result, the task of bounding $L_{\text{un}}(\widehat{f}_{\text{sub}}) - \inf_{f \in \mathcal{F}} L_{\text{un}}(f)$ translates naturally to the sub-tasks of deriving uniform bounds for the absolute deviations between:

1. The sub-sampled empirical risk (Eqn. 7) and the formulation of U-Statistics (Eqn. 10), and

2. The U-Statistics (Eqn. 10) and the population unsupervised risk (Eqn. 3).

In the following sections, we will provide general strategies to tackle both types of deviations. Our proof will rely heavily on the U-Statistics decoupling technique (Arcones & Gine, 1993; de la Peña & Giné, 1998).

### 4.1. Bounding the U-Statistics to Population Unsupervised Risk Deviation

**Overview of U-Statistics**: Given $S = \{\mathbf{x}_1, \ldots, \mathbf{x}_n\}$ drawn $i.i.d.$ from a distribution $\mathcal{P}$ over an input space $\mathcal{X}$. Let $h : \mathcal{X}^m \to \mathbb{R}$ be a symmetric kernel in its arguments, i.e., $h(x_1, \ldots, x_m)$ is unchanged for any permutations of $x_1, \ldots, x_m \in \mathcal{X}$. Then, a natural estimate for the parameter $\theta = \mathbb{E}_{\mathbf{x}_1, \ldots, \mathbf{x}_m \sim \mathcal{P}^m} [h(\mathbf{x}_1, \ldots, \mathbf{x}_m)]$ can be the average of the kernel $h$ over all possible $m$-tuples (selected without replacement) that can be formed from $S$ (which is the one-sample U-Statistics of order $m$ for the kernel $h$), denoted $U_n(h)$. Formally:

$$U_n(h) = \frac{1}{\binom{n}{m}} \sum_{i_1, \ldots, i_m \in C_m[n]} h(\mathbf{x}_{i_1}, \ldots, \mathbf{x}_{i_m}), \tag{8}$$

where $C_m[n]$ denotes the set of $m$-tuples selected (without replacement) from $[n]$ without order ($m$-combinations). We call $U_n(h)$ an one-sample U-Statistics of order $m$. When the kernel $h$ is asymmetric with respect to the arguments, the average is taken over the set of $m$-tuples selected from $[n]$ *with order* instead ($m$-permutations).

**Remark 4.1.** $U_n(h)$ is indeed an unbiased estimator for $\theta$. However, since the formula averages over a set of non-$i.i.d.$ tuples, it is impossible to analyze the concentration properties of $U_n(h)$ without applying further processing.

Our proof relies on the decoupling technique of U-Statistics discussed in de la Peña & Giné (1998). Specifically, the goal is to decouple the U-Statistics as an average of mean over independent "blocks" of samples so that we can apply concentration inequalities (E.g. McDiarmid (1989)). Let $q = \lfloor n/m \rfloor$ (which represents the maximum number of $i.i.d.$ tuples possibly created from $S$), the U-Statistics given in Eqn. (8) can be re-formulated as follows:

$$
U_n(h) = \frac{1}{n!} \sum_{\pi \in S[n]} V_\pi(S),
$$

$$
V_\pi(S) = \frac{1}{q} \sum_{i=1}^{q} h(\mathbf{x}_{\pi[mi-m+1]}, \ldots, \mathbf{x}_{\pi[mi]}),
$$

(9)

and $S[n] = \left\{ \pi : [n] \to [n] \,\middle|\, \pi \text{ is bijective} \right\}$ denotes the set of all possible permutations of the indices $[n] = \{1, \ldots, n\}$.

The techniques related to decoupling U-Statistics has been applied for bipartite ranking in Clémençon et al. (2008), which involves one-sample U-Statistics with order 2. Even though the authors pointed out that their results can be naturally extended to $m$-order U-Statistics for any $m \geq 2$, it is not as direct to formulate a $(k+2)$-order U-Statistics in the case of CRL. Specifically, in Clémençon et al. (2008), the selection of valid tuples is straightforward by choosing (without replacement) $m$ elements arbitrarily from a single random sample $S$. On the other hand, the problem setup of CRL involves $|\mathcal{C}|$ random samples with random sizes. Furthermore, given an arbitrary $(k+2)$-tuple selected (without replacement) from $\mathcal{S}$, the validity of the tuple is restricted by the classes of its elements (as described in Section 3.3).

Therefore, instead of formulating a $(k+2)$-order U-Statistics to estimate $L_{un}(f)$ directly, our approach is to formulate multiple class-wise U-Statistics $\mathcal{U}_N(f|c)$ to estimate each class-conditional risk $L_{un}(f|c)$ separately then combine the estimations. Specifically, Let $\mathcal{T}$ be the set of all possible valid tuples (not necessarily $i.i.d.$, repetition of data points across tuples allowed) that can be formed from the dataset $\mathcal{S}$. Additionally, denote $\mathcal{T} = \bigcup_{c \in \mathcal{C}} \mathcal{T}_c$ where for each $c \in \mathcal{C}$, $\mathcal{T}_c$ denotes the set of valid tuples whose anchor-positive pairs belong to class $c$. Thus, $|\mathcal{T}_c| = 2\binom{N_c^+}{2}\binom{N-N_c^+}{k}$. The core strategy of our proofs relies on the following U-Statistics formulation for the population unsupervised risk $L_{un}(f)$:

$$
\mathcal{U}_N(f) = \sum_{c \in \mathcal{C}} \frac{N_c^+}{N} \mathcal{U}_N(f|c),
$$

$$
\mathcal{U}_N(f|c) = \mathbf{1}\{N_c \geq 1\} U_N(f|c)
$$

(10)

where $N_c = \min(\lfloor N_c^+/2 \rfloor, \lfloor (N - N_c^+)/k \rfloor)$ and $U_N(f|c)$ denotes the U-Statistics that estimates the conditional unsupervised risk $L_{un}(f|c)$, i.e., the average of the contrastive loss $\ell$ over valid tuples in $\mathcal{T}_c$ (cf. Appendix D, Eqn. 27). The indicator involving $N_c$ specifies whether there are enough samples to form at least one tuple where the anchor-positive pair belongs to class $c$. When $N_c = 0$ (i.e., $\mathcal{T}_c = \varnothing$), the dataset $\mathcal{S}$ lacks either anchor-positive pairs or negative samples. In this case, we set the estimation for $L_{un}(f|c)$ to 0, indicating the absence of information about the unsupervised risk conditioned on class $c$.

### 4.2. Bounding the Sub-sampled Empirical Risk to U-Statistics Deviation

Suppose that the dataset $\mathcal{S}$ is given. The sampling procedure described earlier in Section 3.3 is equivalent to selecting M elements (with replacement) from the pool of all possible tuples $\mathcal{T}$ such that the probability of selecting a tuple whose anchor-positive pair belongs to class $c \in \mathcal{C}$ is proportional to $\frac{N_c^+}{N}$ - the unbiased estimator of the class probability $\rho_c$. In other words, it is equivalent to drawing M tuples independently from the discrete distribution $\nu$ over $\mathcal{T}$ such that for any tuple $T \in \mathcal{T}$, the mass assigned to $T$ via $\nu$ is:

$$
\nu(T) = \frac{N_c^+/N}{|\mathcal{T}_c|}, \quad \text{if} \quad T \in \mathcal{T}_c. \tag{11}
$$

Let $\widetilde{\mathcal{C}} = \left\{ c \in \mathcal{C} : N_c \geq 1 \right\}$, which denotes the subset of classes such that $\forall c \in \widetilde{\mathcal{C}}, \mathcal{T}_c \neq \varnothing$. Then, we have:

$$
\underset{\mathcal{T}_{sub} \sim \nu^M}{\mathbb{E}}\left[ \hat{\mathcal{L}}(f; \mathcal{T}_{sub}) \,\middle|\, \mathcal{S} \right] = \underbrace{\sum_{c \in \widetilde{\mathcal{C}}} \frac{N_c^+}{N} U_N(f|c)}_{\mathcal{U}_N(f)}. \tag{12}
$$

In other words, given a fixed draw of the labeled dataset $\mathcal{S}$, the sub-sampled empirical risk, which is evaluated on a subset of tuples drawn $i.i.d.$ from the distribution $\nu$, concentrates to the proposed U-Statistics formulation in Eqn. (10). Hence, we can make use of classic results in learning theory on Rademacher complexity bounds (cf. Proposition F.3) to obtain high probability results on the excess risk of $\hat{f}_{sub}$, conditionally given the draw of the labeled dataset $\mathcal{S}$[2].

## 5. Main Results

**Notations and Settings**: Let $\mathcal{S} = \left\{ (\mathbf{x}_j, \mathbf{y}_j) \right\}_{j=1}^{N}$ be the labeled dataset and $\mathcal{T}$ be the set of all possible valid tuples that can be formed from $\mathcal{S}$. Additionally, for each $c \in \mathcal{C}$:

---

[2]The conditioning can be easily eliminated using the tower rule (cf. Theorem E.1).

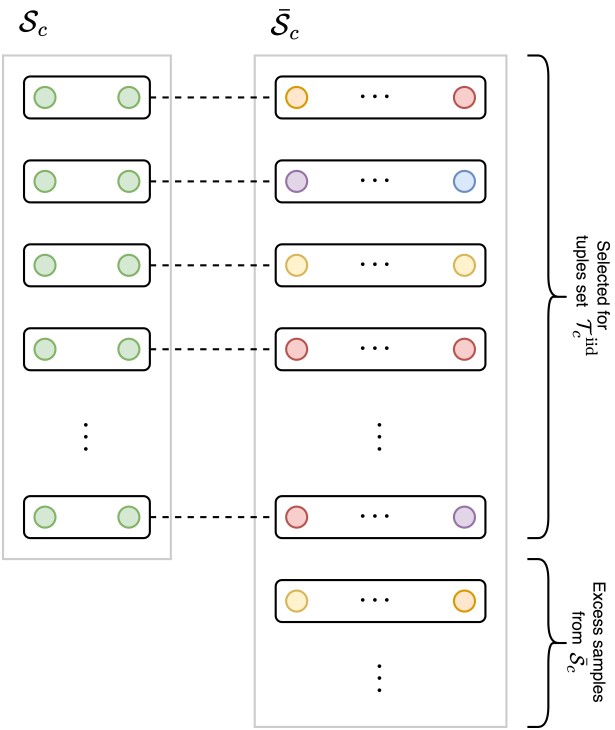

$\mathcal{S}_c$        $\bar{\mathcal{S}}_c$

Selected for tuples set $\mathcal{T}_c^{\text{iid}}$

Excess samples from $\bar{\mathcal{S}}_c$

*Figure 1.* An illustration of the tuples selection process for $\mathcal{T}_c^{\text{iid}}$. In this case, there are excess samples from $\bar{\mathcal{S}}_c$ that are left unused. However, the other way around where there are excess samples from $\mathcal{S}_c$ is also possible (E.g. for very large values of $k$).

- Let $\mathcal{S}_c \subset \mathcal{S}$ the be set of data points belonging to class $c$ (i.e., $\mathcal{S} = \bigcup_{c \in \mathcal{C}} \mathcal{S}_c$). Also, we denote $\bar{\mathcal{S}}_c = \mathcal{S} \backslash \mathcal{S}_c$.

- Let $\mathrm{N}_c^+ = |\mathcal{S}_c|$ and $\mathrm{N}_c^- = |\bar{\mathcal{S}}_c| = \mathrm{N} - \mathrm{N}_c^+$. Also, we define $\mathrm{N}_c = \min(\lfloor \mathrm{N}_c^+/2 \rfloor, \lfloor \mathrm{N}_c^-/k \rfloor)$.

- Let $\mathcal{T}_c^{\text{iid}}$ be the set of independent tuples selected in a "greedy" way such that each tuple's anchor-positive pair belongs to class $c$. Specifically, we select tuples for $\mathcal{T}_c^{\text{iid}}$ by matching 2-element disjoint blocks from $\mathcal{S}_c$ and $k$-element disjoint blocks from $\bar{\mathcal{S}}_c$ until one of the two sets runs out of blocks to select (cf. Figure 1). Therefore, when $\mathcal{T}_c^{\text{iid}} \neq \varnothing$, $|\mathcal{T}_c^{\text{iid}}| = \mathrm{N}_c$.

Furthermore, for any subset of tuples $\mathcal{T}' \subset \mathcal{T}$, we denote $\widehat{\mathfrak{R}}_{\mathcal{T}'}(\ell \circ \mathcal{F})$ as the empirical Rademacher complexity of the loss class $\ell \circ \mathcal{F}$ (Eqn. (6)) restricted to $\mathcal{T}'$. Formally, if we denote $\mathcal{T}' = \left\{ (\mathbf{x}_j, \mathbf{x}_j^+, \mathbf{x}_{j1}^-, \ldots, \mathbf{x}_{jk}^-) \right\}_{j=1}^n$, we have:

$$\widehat{\mathfrak{R}}_{\mathcal{T}'}(\ell \circ \mathcal{F}) = \\ \mathbb{E}_{\boldsymbol{\Sigma}_n} \left[ \sup_{f \in \mathcal{F}} \left| \frac{1}{n} \sum_{j=1}^n \sigma_j \ell\left( \left\{ f_j^\top \left[ f_j^+ - f_{ji}^- \right] \right\}_{i=1}^k \right) \right| \right], \quad (13)$$

where $f(\mathbf{x}_j), f(\mathbf{x}_j^+), f(\mathbf{x}_{ji}^-)$ are denoted as $f_j, f_j^+, f_{ji}^-$ for all $1 \leqslant j \leqslant n, 1 \leqslant i \leqslant k$ for notational brevity and we denote $\boldsymbol{\Sigma}_n = \left\{ \sigma_1, \ldots, \sigma_n \right\}$ as the vector of independent Rademacher variables. Additionally, when $\mathcal{T}' = \varnothing$, we define $\widehat{\mathfrak{R}}_{\mathcal{T}'}(\ell \circ \mathcal{F}) = 0$ for completeness.

**Assumption**: We assume that for all classes $c \in \mathcal{C}$, whenever $\mathcal{T}_c^{\text{iid}} \neq \varnothing$ (i.e., $\mathrm{N}_c \geqslant 1$), we can bound the empirical Rademacher complexity $\widehat{\mathfrak{R}}_{\mathcal{T}_c^{\text{iid}}}(\ell \circ \mathcal{F})$ of the loss class as:

$$\widehat{\mathfrak{R}}_{\mathcal{T}_c^{\text{iid}}}(\ell \circ \mathcal{F}) \leqslant \frac{K_{\mathcal{F},c}}{\sqrt{\mathrm{N}_c}}, \quad (14)$$

where $K_{\mathcal{F},c}$ is an expression that depends on both the function class $\mathcal{F}$ and the class $c$ itself. For example, by the Dudley entropy integral (Theorem F.5), we can bound $\widehat{\mathfrak{R}}_{\mathcal{T}_c^{\text{iid}}}(\ell \circ \mathcal{F})$ with $K_{\mathcal{F},c}$ as an expression that involves the $L_2$-covering number (cf. Definition D.11) of $\ell \circ \mathcal{F}$ restricted to the subset of independent tuples $\mathcal{T}_c^{\text{iid}}$.

In the results that follow, for a given representation function $\hat{f} \in \mathcal{F}$, we denote $\mathrm{ER}_{\mathrm{un}}(\hat{f}) = \mathrm{L}_{\mathrm{un}}(\hat{f}) - \inf_{f \in \mathcal{F}} \mathrm{L}_{\mathrm{un}}(f)$ for notational brevity (short for "excess risk").

### 5.1. Generalization Bound for Empirical Minimizer of U-Statistics

**Theorem 5.1** (cf. Theorem D.10)**.** *Let $\mathcal{F}$ be a class of representation functions and $\ell : \mathbb{R}^k \to [0, \mathcal{M}]$ be a bounded contrastive loss. Let $\hat{f}_{\mathcal{U}} = \arg\min_{f \in \mathcal{F}} \mathcal{U}_{\mathrm{N}}(f)$. Then, for any $\delta \in (0, 1)$, with probability of at least $1 - \delta$:*

$$\mathrm{ER}_{\mathrm{un}}(\hat{f}_{\mathcal{U}}) \leqslant \mathcal{O}\left[ \sum_{c \in \mathcal{C}} \rho(c) \frac{K_{\mathcal{F},c}}{\sqrt{\widetilde{\mathrm{N}}}} + \mathcal{M} \sqrt{\frac{\ln |\mathcal{C}|/\delta}{\widetilde{\mathrm{N}}}} \right], \quad (15)$$

*where $\widetilde{\mathrm{N}} = \mathrm{N} \min\left( \frac{\min_{c \in \mathcal{C}} \rho(c)}{2}, \frac{1 - \max_{c \in \mathcal{C}} \rho(c)}{k} \right)$.*

To understand the intuition of Theorem 5.1, let us assume that the underlying distribution over the labels set $\mathcal{C}$ is perfectly balanced, meaning $\rho(c) = |\mathcal{C}|^{-1}, \forall c \in \mathcal{C}$. Under this assumption, we have $\widetilde{\mathrm{N}} = \mathrm{N} \max[2|\mathcal{C}|, k|\mathcal{C}|/(|\mathcal{C}| - 1)]^{-1}$, which means that the bound scales as $\mathcal{O}(\sqrt{|\mathcal{C}|/\mathrm{N}})$ for $k \leqslant 2|\mathcal{C}| - 2$ and as $\mathcal{O}(\sqrt{k/\mathrm{N}})$ for larger values of $k$. In other words, for a given desired generalization gap, the sample complexity of the model exhibits a square-root dependency on either the number of classes or on the number of negative samples. For example, if we consider a class of neural networks with $\mathcal{W}$ parameters in total, the sample complexity to reach a desired generalization gap of $\epsilon > 0$ scales in the order of $\widetilde{\mathcal{O}}\left[ \frac{\mathcal{W}k}{\epsilon^2} \right]$ (cf. Table 1 or Theorem D.17) for large values of $k$ and when the distribution over $\mathcal{C}$ is perfectly balanced.

At first glance, the bound may seem worse than that of previous works by Lei et al. (2023) and Hieu et al. (2024)

*Table 1.* Summary of generalization bounds for $\widehat{f}_{\mathcal{U}}$ and $\widehat{f}_{\text{sub}}$ when the class of representation functions are linear maps and neural networks. The $\widetilde{\mathcal{O}}$ notation hides poly-logarithmic terms of all relevant quantities. For linear maps, $N, \mathcal{M}, \eta, s, a, b, k$ and $d$ are hidden. For neural networks, $\eta, N, L, b, \{\xi_l\}_{l=1}^L$ and $\{s_l\}_{l=1}^L$ are hidden.

| $\widehat{f}$ | **Function Classes** | **Generalization Bounds** $\text{ER}_{\text{un}}(\widehat{f})$ | **Relevant Results** |
|---|---|---|---|
| $\widehat{f}_{\mathcal{U}}$ | $\mathcal{F}_{\text{lin}}$ (Eqn. (17)) | $\widetilde{\mathcal{O}}\left[ \frac{\eta s a b^2}{\sqrt{\widetilde{N}}} + \mathcal{M}\sqrt{\frac{\ln |\mathcal{C}|/\delta}{\widetilde{N}}} \right]$ | Theorem D.15 |
| $\widehat{f}_{\mathcal{U}}$ | $\mathcal{F}_{\text{nn}}^{\mathcal{A}}$ (Eqn. (18)) | $\widetilde{\mathcal{O}}\left[ \frac{\mathcal{M}\mathcal{W}^{\frac{1}{2}}}{\sqrt{\widetilde{N}}} + \mathcal{M}\sqrt{\frac{\ln |\mathcal{C}|/\delta}{\widetilde{N}}} \right]$ | Theorem D.17 |
| $\widehat{f}_{\text{sub}}$ | $\mathcal{F}_{\text{lin}}$ (Eqn. (17)) | $\widetilde{\mathcal{O}}\left[ \eta s a b^2 \left( \sqrt{\frac{1}{\widetilde{N}}} + \sqrt{\frac{1}{M}} \right) + \mathcal{M}\left( \sqrt{\frac{\ln |\mathcal{C}|/\delta}{\widetilde{N}}} + \sqrt{\frac{\ln 1/\delta}{M}} \right) \right]$ | Theorem E.2 |
| $\widehat{f}_{\text{sub}}$ | $\mathcal{F}_{\text{nn}}^{\mathcal{A}}$ (Eqn. (18)) | $\widetilde{\mathcal{O}}\left[ \mathcal{M}\mathcal{W}^{\frac{1}{2}} \left( \sqrt{\frac{1}{\widetilde{N}}} + \sqrt{\frac{1}{M}} \right) + \mathcal{M}\left( \sqrt{\frac{\ln |\mathcal{C}|/\delta}{\widetilde{N}}} + \sqrt{\frac{\ln 1/\delta}{M}} \right) \right]$ | Theorem E.3 |

where one of the primary strengths of their results is the logarithmic dependency on the tuples size $k$. However, we note that the results in the above references express sample complexity in terms of *tuples* rather than the number of labeled data points. Specifically, their generalization bounds scale in the order of $\mathcal{O}(1/\sqrt{n})$ where $n$ is the number of tuples drawn $i.i.d.$ from an unknown distribution of tuples. In other words, there are $N = nk$ labeled data points in total. Therefore, under the regime where the label distribution is perfectly balanced and there is a large number of negative samples, our bound behaves similarly as the bounds derived by prior works.

One key drawback of the bound in Theorem 5.1 is its square-root dependency on the number of classes $|\mathcal{C}|$ when $k$ is small. This arises from analyzing the concentration of each U-Statistics $\mathcal{U}_N(f|c)$ around their corresponding conditional unsupervised risk $L_{\text{un}}(f|c)$ independently. As a result, the sample size required to achieve a desired generalization gap has to be large enough such that for all $c \in \mathcal{C}$, the deviations $|\mathcal{U}_N(f|c) - L_{\text{un}}(f|c)|$ are small even though the probabilities for most classes are low (especially for large number of classes $|\mathcal{C}|$), making their contribution to the overall unsupervised risk $L_{\text{un}}(f)$ negligible. Consequently, the minimum occurrence probability $\min_{c \in \mathcal{C}} \rho(c)$ is introduced into the sample complexity as a result of minimizing the deviations between $\mathcal{U}_N(f|c)$ and $L_{\text{un}}(f|c)$ across all classes equally.

We conjecture that there is a certain U-Statistics formulation that accounts for the concentration of all classes at once. However, we note that this is a major technical challenge as it is non-trivial to formulate a $(k+2)$-order U-Statistics to estimate $L_{\text{un}}(f)$ directly. This is primarily due to the fact that a randomly selected $(k+2)$-tuple from the pool of labeled data points $\mathcal{S}$ is not automatically a valid input for the loss function $\ell$, making it complicated to apply techniques in de la Peña & Giné (1998) and Clémençon et al. (2008).

### 5.2. Generalization Bound for Empirical Sub-sampled Risk Minimizer

**Theorem 5.2** (cf. Theorem E.1). *Let $\mathcal{F}$ be a class of representation functions and $\ell : \mathbb{R}^k \to [0, \mathcal{M}]$ be a bounded contrastive loss. Let $\widehat{f}_{\text{sub}} = \arg\min_{f \in \mathcal{F}} \widehat{\mathcal{L}}(f; \mathcal{T}_{\text{sub}})$. Then, for any $\delta \in (0,1)$, with probability of at least $1 - \delta$:*

$$\text{ER}_{\text{un}}(\widehat{f}_{\text{sub}}) \leqslant \mathcal{O}\left[ \sum_{c \in \mathcal{C}} \rho(c) \frac{K_{\mathcal{F},c}}{\sqrt{\widetilde{N}}} + \widehat{\mathfrak{R}}_{\mathcal{T}_{\text{sub}}}(\ell \circ \mathcal{F}) \right. \tag{16}$$
$$\left. + \mathcal{M}\sqrt{\frac{\ln 1/\delta}{M}} + \mathcal{M}\sqrt{\frac{\ln |\mathcal{C}|/\delta}{\widetilde{N}}} \right],$$

*where $\widetilde{N} = N \min\left( \frac{\min_{c \in \mathcal{C}} \rho(c)}{2}, \frac{1 - \max_{c \in \mathcal{C}} \rho(c)}{k} \right)$.*

From the above theorem, there is a striking resemblance to the result in theorem 5.1 except there is an additional cost of order $\mathcal{O}\left( \sqrt{\frac{\ln 1/\delta}{M}} \right)$. Although there is an additional empirical Rademacher complexity $\widehat{\mathfrak{R}}_{\mathcal{T}_{\text{sub}}}(\ell \circ \mathcal{F})$ appearing in the bound, by Theorem F.5, we know that it also scales in the order of $\mathcal{O}(1/\sqrt{M})$. The implication of this result is straightforward: under the sub-sampling regime, when the number of sub-sampled tuples is large, the performance discrepancy between the empirical sub-sampled risk minimizer $\widehat{f}_{\text{sub}}$ and $\widehat{f}_{\mathcal{U}}$ is negligible. As a result, under circumstances when the amount of labeled data points available is too large, making it computationally impossible to train on all valid tuples, the training of representation functions can be improved by simply increasing the number of (non-independent) tuples. We further corroborate this intuition with a numerical experiment (Figure 2), which shows that the performance of the sub-sampled empirical risk minimizer tends to the performance of the full U-Statistics minimizer as the number of sub-sampled tuples $M$ increases.

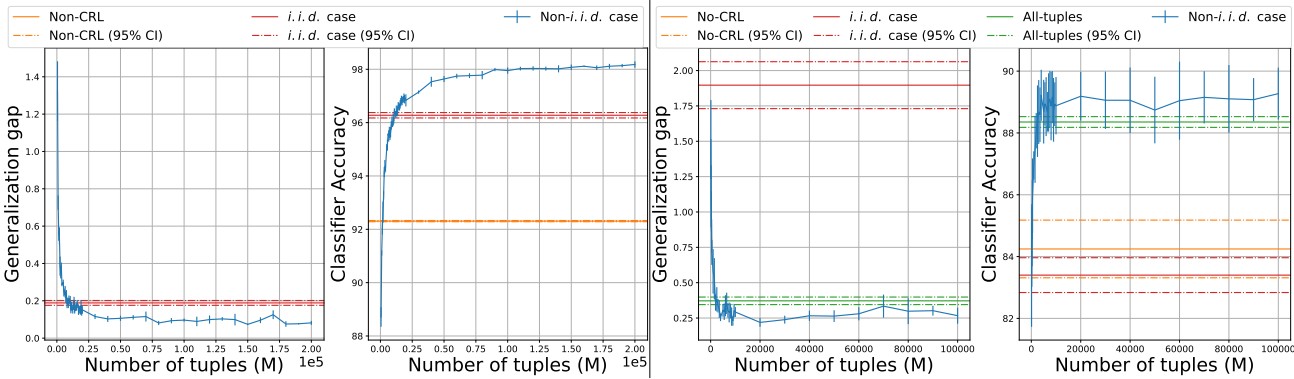

*Figure 2.* Summary of results for experiments with the MNIST dataset. On the left, we have the results for $n = 10000$. On the right, we have the results for $n = 100$ as well as the additional result for the *all-tuples* regime.

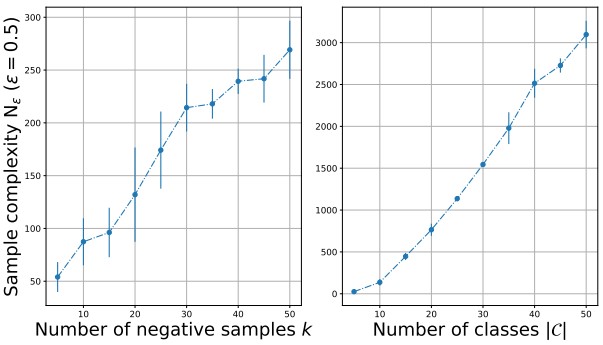

*Figure 3.* Summary of results for synthetic data experiments on the relationship between $|\mathcal{C}|, k$ and the sample complexity.

### 5.3. Generalization Bounds for Common Classes of Representation Functions

Suppose that the contrastive loss function $\ell : \mathbb{R}^k \to [0, \mathcal{M}]$ is $\ell^\infty$-Lipschitz with constant $\eta > 0$. Specifically, for all $\mathbf{v}, \mathbf{v}' \in \mathbb{R}^k$, $|\ell(\mathbf{v}) - \ell(\mathbf{v}')| \leqslant \eta \|\mathbf{v} - \mathbf{v}'\|_\infty$. Also, suppose that $\sup_{\mathbf{x} \in \mathcal{S}} \|\mathbf{x}\|_2 \leqslant b$ with probability one (with respect to the draw of dataset $\mathcal{S}$) for some $b \in \mathbb{R}_+$. In this section, we apply our main theorems to derive generalization bounds for common classes of representation functions such as linear maps and neural networks. We provide a summary of our main results in Table 1 and a brief description of the considered function classes below.

**Linear Functions**: For a matrix $A \in \mathbb{R}^{m \times d}$, we define the $\| \cdot \|_{2,1}$ norm as $\|A\|_{2,1} = \sum_{i=1}^d \|A_{\cdot,i}\|_2$, i.e., the sum of column Euclidean norms and $\| \cdot \|_\sigma$ as the spectral norm. We define the class of linear functions as follows:

$$\mathcal{F}_{\text{lin}} = \Big\{ x \mapsto Ax : A \in \mathbb{R}^{d \times m}, \\ \|A^\top\|_{2,1} \leqslant a, \|A\|_\sigma \leqslant s \Big\}. \tag{17}$$

**Neural Networks**: Let $L \geqslant 1$ and $d_0, d_1, \ldots, d_L$ be known positive integers representing layer widths. Furthermore, let $\mathcal{W} = \sum_{l=1}^L d_l$, which is the total number of parameters in the neural networks. For $1 \leqslant 1 \leqslant L$, we define the sets of matrices $\mathcal{B}_l = \Big\{ A^{(l)} \in \mathbb{R}^{d_{l-1} \times d_l} : \|A^{(l)}\|_\sigma \leqslant s_l \Big\}$ as the layer-wise parameter spaces where $\{s_l\}_{l=1}^L$ is a sequence of known real positive numbers. Let $\{\varphi_l : \mathbb{R}^{d_l} \to \mathbb{R}^{d_l}\}_{l=1}^L$ be a sequence of activation functions that are fixed a-priori which are $\ell^2$-Lipschitz with constants $\{\xi_l\}_{l=1}^L$, i.e., $\forall \mathbf{x}, \mathbf{x}' \in \mathbb{R}^{d_l}$ and $1 \leqslant l \leqslant L$, we have $\|\varphi_l(\mathbf{x}) - \varphi_l(\mathbf{x}')\| \leqslant \xi_l \|\mathbf{x} - \mathbf{x}'\|_2$. Then, we are interested in the class of neural networks $\mathcal{F}_{\text{nn}}^{\mathcal{A}}$ defined as follows:

$$\mathcal{F}_{\text{nn}}^{\mathcal{A}} = \mathcal{F}_L \circ \mathcal{F}_{L-1} \circ \cdots \circ \mathcal{F}_1, \tag{18}$$

$$\text{where } \mathcal{F}_l = \Big\{ x \mapsto \varphi_l\Big(A^{(l)}x\Big) : A^{(l)} \in \mathcal{B}_l \Big\}, \forall l \in [L].$$

## 6. Experiments

In this section, we describe the numerical experiments on both synthetic and open sourced datasets to empirically verify our main results. Specifically, we aim to provide empirical evidence to corroborate three hypotheses. Firstly, the sample complexity, i.e., the number of labeled examples required to reach a desirable performance, for the U-Statistics minimizer corresponds (linearly) to the number of negative samples and the number of classes. Secondly, as the number of sub-sampled tuples increases, the performance of the sub-sampled empirical risk minimizer approaches that of the full U-Statistics minimizer. Finally, the performance of models trained on tuples with recycled data (non-*i.i.d.* tuples) can outperform both models trained on independent tuples and models trained directly with the non-contrastive cross-entropy loss. This observation is in line with the existing literature (Khosla et al., 2020). In fact, the benefit of recycling data is most present when there are too few labeled data points to construct disjoint tuples. For all experiments detailed below, we fix $\mathcal{F}$ as a class of shallow

neural networks (with a fixed number of layers $L = 2$). We summarize our experiment results in Figure 3 and Figure 2.

## 6.1. Performance of $\widehat{f}_{\mathrm{sub}}$ as M Increases

In this section, we describe two experiments conducted on the MNIST dataset. Firstly, we sample from the original dataset $n$ independent (disjoint) valid tuples. Then, for each of the following regimes, we train the neural networks for 5 random weights initializations:

- **Independent tuples**: Using only the selected $n$ independent tuples as the training dataset.

- **Sub-sampled tuples**: Among all the $n(k + 2)$ labeled data points from the previously selected $n$ disjoint tuples, we sub-sample M tuples and use them as the training dataset.

For each regime and random weight initialization, we also train and evaluate a linear classifier on top of the learned representations extracted from the original MNIST dataset. We conduct the experiments with $n = 100$ and $n = 10000$. For the case of $n = 100$, we also compare the above regimes with the performance of the neural networks when trained on all possible valid tuples. The results of both experiments are summarized in Figure 2. For both cases, the results show that the sub-sampling regime helps with the performance for both supervised and unsupervised tasks. Noticeably, when the number of labeled examples ($n = 100$) is small, the result of reusing samples across tuples outperforms the $i.i.d.$ regime extremely fast, which cements the practical validity of the learning setting considered. Moreover, for $n = 100$, we observe that the performance of the sub-sampling regime closely approximates the all-tuples regime. This observation is in line with the implications from Theorem 5.2. Furthermore, the model trained on sub-sampled tuples quickly outperforms the fully independent tuples regime as M increases, highlighting the benefit of recycling data when the amount of labeled examples is limited.

## 6.2. Correlation Between Sample Complexity and Values of $|\mathcal{C}|, k$

Let us denote $\mathrm{N}_\epsilon$ as the minimum number of samples required to achieve a desired generalization gap $\epsilon > 0$. We conducted two ablation studies to investigate the correlation between $\mathrm{N}_\epsilon$ and different values of $|\mathcal{C}|$ and $k$.

**Initialization**: For each value of N, $|\mathcal{C}|$ and $k$, we generate the sample sizes $\{\mathrm{N}_c^+\}_{c \in \mathcal{C}} \sim \mathrm{Multinom}(\mathrm{N}, \{|\mathcal{C}|^{-1}\}_{c \in \mathcal{C}})$ (assuming perfectly balanced classes condition) and random Gaussian centers $\mathbf{g}_c \in \mathbb{R}^{128}$ corresponding to each class $c \in \mathcal{C}$. Then, for each class $c \in \mathcal{C}$, we generate the samples from that class as $\mathcal{S}_c \sim \mathcal{N}(\mathbf{g}_c, \sigma^2)^{\mathrm{N}_c^+}$ with a fixed standard deviation of $\sigma = 0.1$.

**Training**: We create 5 random generations of Gaussian datasets for each configuration of $|\mathcal{C}|$ and $k$ values. Then, for each configuration and dataset, we find the sample complexity $\mathrm{N}_\epsilon$ corresponding to a desired gap $\epsilon$ using a binary search approach within a fixed values range of N. For each search, we train the neural network with $\mathrm{M} = \mathrm{N}^2$ sub-sampled tuples so that the performance is approximately close to that of $\widehat{f}_{\mathcal{U}}$. Fixing $\epsilon = 0.5$, we conduct the ablation studies with the following values ranges of $|\mathcal{C}|$ and $k$:

- **Ablation study 1**: Fix $k = 3$ and increase the values of $|\mathcal{C}|$ from 5 to 50 with an interval of 5.

- **Ablation study 2**: Fix $|\mathcal{C}| = 5$ and increase the values of $k$ from 5 to 50 with an interval of 5.

For each configuration, we average the sample complexities over the random dataset generations and plot the results in Figure 3. For both experiments, the sample complexities display linear relationships with values of $k$ and $|\mathcal{C}|$. These observations are consistent with the result in Theorem 5.1.

## 7. Conclusion & Future Work

In this work, we derive generalization bounds for the CRL framework when training is limited to a fixed pool of reusable labeled examples. We provide two main results on the excess risk bounds for the U-Statistics minimizer $\widehat{f}_{\mathcal{U}}$ and the empirical sub-sampled risk minimizer $\widehat{f}_{\mathrm{sub}}$. We show that under the assumption that the class distribution is perfectly balanced, our results for the U-Statistics minimizer behave similarly to the previous analyses conducted for the $i.i.d.$ regime. Furthermore, we prove both theoretically and empirically that under the sub-sampling regime, as the number of sub-sampled tuples increases, the performance of $\widehat{f}_{\mathrm{sub}}$ is approximately close to $\widehat{f}_{\mathcal{U}}$. Thereby, verifying the validity of the common practice of recycling labeled samples across input tuples. Finally, we apply our main results to derive specific generalization bounds for common function classes such as linear maps and neural networks.

In our experiments, we demonstrate the advantages of recycling samples in different tuples, and confirm the superiority of supervised contrastive learning over direct training with the cross-entropy loss.

We also note that our excess risk bounds possess a dependency of $\mathcal{O}(1/\sqrt{\rho_{\min}})$ where $\rho_{\min}$ is the probability associated to the rarest class. This can potentially be overly pessimistic in cases when there are a lot of small categories in the true class distribution. This dependency arises because all the class-wise U-Statistics $\mathcal{U}_{\mathrm{N}}(f|c)$ need to concentrate uniformly. Therefore, a possible future direction is to improve this dependence by designing an estimation strategy which captures joint concentration across small classes.

## Impact Statement

This work is primarily theoretical and we cannot foresee any potential implications that need to be addressed.

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

# Supplementary Materials : Generalization Analysis for Supervised Contrastive Representation Learning under Non-IID Settings

## A. Table of Notations

Table 2: Table of notations for quick reference.

| Notation | Description |
|---|---|
| **Basic Setups** | |
| Let $\mathcal{X}$ be the input space and $\mathcal{C}$ be the finite set of labels endowed with the probability measure $\rho$. | |
| Let $\mathcal{S} = \left\{ (\mathbf{x}_j, \mathbf{y}_j) \right\}_{j=1}^{N}$ be a labeled dataset sampled $i.i.d.$ from a distribution over $\mathcal{X} \times \mathcal{C}$. | |
| We fix a class of representation $\mathcal{F}$, for every $c \in \mathcal{C}$, we have the following notations: | |
| $\rho(c)$ | The probability mass assigned to class $c$, i.e., the occurrence probability of class $c$. |
| $\mathcal{D}_c$ | The distribution over $\mathcal{X}$ of data points belonging to class $c$. |
| $\bar{\mathcal{D}}_c$ | The distribution over $\mathcal{X}$ of data points not belonging to class $c$. |
| $\mathcal{S}_c$ | The subset of $\mathcal{S}$ that contains instances of class $c \in \mathcal{C}$. |
| $\bar{\mathcal{S}}_c$ | The subset of $\mathcal{S}$ that contains instances not belonging to class $c \in \mathcal{C}$. $\bar{\mathcal{S}}_c = \mathcal{S} \backslash \mathcal{S}_c$. |
| $\mathbf{x}_i^{(c)} / \mathbf{x}_j^{(\bar{c})}$ | The $i^{th}$ and $j^{th}$ elements of $\mathcal{S}_c$ and $\bar{\mathcal{S}}_c$, respectively. |
| $f_{c,i} / f_{\bar{c},j}$ | The shortcuts for $f(\mathbf{x}_i^{(c)})$ and $f(\mathbf{x}_j^{(\bar{c})})$ where $f \in \mathcal{F}$, respectively. |
| $N_c^+$ | Number of samples that belong to class $c$. We have $N_c^+ \sim \text{Bin}(N, \rho(c))$. |
| $N_c^-$ | Number of samples that does not belong to class $c$. $N_c^- = N - N_c^+$. |
| $N_c$ | Number of possible $i.i.d.$ tuples with positive pairs in class $c$. $N_c = \min(\lfloor N_c^+/2 \rfloor, \lfloor N_c^-/k \rfloor)$. |
| $\overline{N}_c$ | $\overline{N}_c = \max(1, N_c)$. |
| $\ell \circ \mathcal{F}$ | $\ell \circ \mathcal{F} = \left\{ (\mathbf{x}, \mathbf{x}^+, \mathbf{x}_{1:k}^-) \mapsto \ell\left( \left\{ f(\mathbf{x})^\top \left[ f(\mathbf{x}^+) - f(\mathbf{x}_i^-) \right] \right\}_{i=1}^{k} \right) : f \in \mathcal{F} \right\}$ - loss function class. |
| **Combinatorics** | |
| $[n]$ | $[n] = \{1, 2, \ldots, n\}$ where $n \in \mathbb{N}$. |
| $S[n]$ | The set of all permutations (shuffles) of the indices set $[n]$. |
| $P_m[n]$ | The set of all permutations of $m$ elements from the indices set $[n]$. |
| $C_m[n]$ | The set of all combinations of $m$ elements from the indices set $[n]$. |
| **Datasets & Auxiliary Datasets** | |

| | |
|---|---|
| $\mathcal{T}$ | The set of all possible valid tuples. $\mathcal{T} = \bigcup_{c \in \mathcal{C}} \mathcal{T}_c$. |
| $\mathcal{T}_c$ | The set of all valid tuples whose anchor-positive pairs belong to $c \in \mathcal{C}$ (Eqn. (20)). |
| $\mathcal{T}_c^{\mathrm{iid}}$ | The set of $i.i.d.$ tuples whose anchor-positive pairs belong to $c \in \mathcal{C}$ (Eqn. (21)). |
| $\widetilde{\mathcal{T}}_c^{\mathrm{iid}}$ | The set of all vectors including anchor, positive and negative samples from $\mathcal{T}_c^{\mathrm{iid}}$ (Eqn. (22)). |
| $\mathcal{T}_{\mathrm{sub}}$ | The subset of tuples sampled $i.i.d.$ from distribution $\nu$ over $\mathcal{T}$ (Eqn. (52)). |
| $\nu$ | The discrete distribution over $\mathcal{T}$ such that $\nu(T) = \frac{\mathrm{N}_c^+/\mathrm{N}}{|\mathcal{T}_c|}$ if $T \in \mathcal{T}_c$. |

### Risk, Empirical Risk & U-Statistics

Let $\ell : \mathbb{R}^k \to [0, \mathcal{M}]$ be a contrastive loss function and $\mathcal{F}$ be a class of representation functions.

| | |
|---|---|
| $\mathrm{L}_{\mathrm{un}}(f|c)$ | $\mathrm{L}_{\mathrm{un}}(f|c) = \mathbb{E}_{\substack{\mathbf{x},\mathbf{x}^+ \sim \mathcal{D}_c^2 \\ \mathbf{x}_{1:k}^- \sim \bar{\mathcal{D}}_c^k}} \Big[ \ell\Big( \big\{ f(\mathbf{x})^\top [f(\mathbf{x}^+) - f(\mathbf{x}_i^-)] \big\}_{i=1}^k \Big) \Big]$. |
| $\mathrm{L}_{\mathrm{un}}(f)$ | $\mathrm{L}_{\mathrm{un}}(f) = \mathbb{E}_{c \sim \rho} \mathbb{E}_{\substack{\mathbf{x},\mathbf{x}^+ \sim \mathcal{D}_c^2 \\ \mathbf{x}_{1:k}^- \sim \bar{\mathcal{D}}_c^k}} \Big[ \ell\Big( \big\{ f(\mathbf{x})^\top [f(\mathbf{x}^+) - f(\mathbf{x}_i^-)] \big\}_{i=1}^k \Big) \Big] = \sum_{c \in \mathcal{C}} \rho(c) \mathrm{L}_{\mathrm{un}}(f|c)$. |
| $\mathrm{U}_{\mathrm{N}}(f|c)$ | $\mathrm{U}_{\mathrm{N}}(f|c) = \frac{1}{|\mathcal{T}_c|} \sum_{\substack{j_1,j_2 \in \mathrm{P}_2[\mathrm{N}_c^+] \\ l_1,\dots,l_k \in \mathrm{C}_k[\mathrm{N}_c^-]}} \ell\Big( \big\{ f_{c,j_1}^\top [f_{c,j_2} - f_{\bar{c},l_i}] \big\}_{i=1}^k \Big)$ if $\mathcal{T}_c \neq \varnothing$. |
| $\mathcal{U}_{\mathrm{N}}(f|c)$ | $\mathcal{U}_{\mathrm{N}}(f|c) = \mathrm{U}_{\mathrm{N}}(f|c)$ if $\mathcal{T}_c \neq \varnothing$ and $\mathcal{U}_{\mathrm{N}}(f|c) = 0$ otherwise. |
| $\mathcal{U}_{\mathrm{N}}(f)$ | $\mathcal{U}_{\mathrm{N}}(f) = \sum_{c \in \mathcal{C}} \frac{\mathrm{N}_c^+}{\mathrm{N}} \mathcal{U}_{\mathrm{N}}(f|c)$. |
| $\widehat{\mathcal{L}}(f; \mathcal{T}_{\mathrm{sub}})$ | $\widehat{\mathcal{L}}(f; \mathcal{T}_{\mathrm{sub}}) = \frac{1}{|\mathcal{T}_{\mathrm{sub}}|} \sum_{(\mathbf{x}_j, \mathbf{x}_j^+, \mathbf{x}_{ji}^-) \in \mathcal{T}_{\mathrm{sub}}} \ell\Big( \big\{ f(\mathbf{x}_j)^\top [f(\mathbf{x}_j^+) - f(\mathbf{x}_{ji}^-)] \big\}_{i=1}^k \Big)$. |
| $\widehat{f}_{\mathcal{U}}$ | $\widehat{f}_{\mathcal{U}} = \arg\min_{f \in \mathcal{F}} \mathcal{U}_{\mathrm{N}}(f)$. |
| $\widehat{f}_{\mathrm{sub}}$ | $\widehat{f}_{\mathrm{sub}} = \arg\min_{f \in \mathcal{F}} \widehat{\mathcal{L}}(f; \mathcal{T}_{\mathrm{sub}})$. |

### Notations for Rademacher Complexity

In the following notations, suppose that we have a distribution $\mathcal{P}$ over an input space $\mathcal{Z}$.

Let $S = \{\mathbf{z}_1, \dots, \mathbf{z}_n\}$ be drawn $i.i.d.$ from $\mathcal{P}$ and let $\mathcal{G}$ be a function class.

| | |
|---|---|
| $\mathbf{\Sigma}_n$ | $\mathbf{\Sigma}_n = \{\sigma_1, \dots, \sigma_n\}$ is the sequence of $n$ $i.i.d.$ Rademacher variables. |
| $\mathcal{R}_{\mathcal{G}}^{S, \mathbf{\Sigma}_n}$ | A random variable depending on $S, \mathbf{\Sigma}_n$ and the class $\mathcal{G}$. $\mathcal{R}_{\mathcal{G}}^{S, \mathbf{\Sigma}_n} = \sup_{g \in \mathcal{G}} \Big| \frac{1}{n} \sum_{j=1}^n \sigma_j g(\mathbf{z}_j) \Big|$. |
| $\widehat{\mathfrak{R}}_S(\mathcal{G})$ | The empirical Rademacher Complexity of $\mathcal{G}$ restricted to $S$. $\widehat{\mathfrak{R}}_S(\mathcal{G}) = \mathbb{E}_{\mathbf{\Sigma}_n | S} \Big[ \mathcal{R}_{\mathcal{G}}^{S, \mathbf{\Sigma}_n} \Big]$ when $n \geqslant 1$. |
| | and $\widehat{\mathfrak{R}}_S(\mathcal{G}) = 0$ when $n = 0$ (in other words, $S = \varnothing$). |
| $\mathfrak{R}_n(\mathcal{G})$ | The expected Rademacher Complexity of $\mathcal{G}$. $\mathfrak{R}_n(\mathcal{G}) = \mathbb{E}_S \Big[ \widehat{\mathfrak{R}}_S(\mathcal{G}) \Big]$. |

### Specific Rademacher Complexities Used in The Main Results

| | |
|---|---|
| $\mathcal{R}_{\mathcal{F}}^{\mathcal{T}_c^{\mathrm{iid}}, \mathbf{\Sigma}_{\mathrm{N}_c}}$ | $\mathcal{R}_{\mathcal{F}}^{\mathcal{T}_c^{\mathrm{iid}}, \mathbf{\Sigma}_{\mathrm{N}_c}} = \sup_{f \in \mathcal{F}} \Big| \frac{1}{\mathrm{N}_c} \sum_{j=1}^{\mathrm{N}_c} \sigma_j \ell\Big( \big\{ f_{c,2j-1}^\top \big[ f_{c,2j} - f_{\bar{c}, kj-k+i} \big] \big\}_{i=1}^k \Big) \Big|$ |
| $\widehat{\mathfrak{R}}_{\mathcal{T}_c^{\mathrm{iid}}}(\ell \circ \mathcal{F})$ | $\widehat{\mathfrak{R}}_{\mathcal{T}_c^{\mathrm{iid}}}(\ell \circ \mathcal{F}) = \begin{cases} \mathbb{E}_{\mathbf{\Sigma}_{\mathrm{N}_c} | \mathcal{S}, \mathrm{N}_c^+} \Big[ \mathcal{R}_{\mathcal{F}}^{\mathcal{T}_c^{\mathrm{iid}}, \mathbf{\Sigma}_{\mathrm{N}_c}} \Big] & \text{when } \mathrm{N}_c \geqslant 1 \\ 0 & \text{when } \mathrm{N}_c = 0 \end{cases}$ |
| $\mathfrak{R}_{\mathrm{N}_c}(\ell \circ \mathcal{F})$ | $\mathfrak{R}_{\mathrm{N}_c}(\ell \circ \mathcal{F}) = \mathbb{E}_{\mathcal{S} | \mathrm{N}_c^+} \big[ \widehat{\mathfrak{R}}_{\mathcal{T}_c^{\mathrm{iid}}}(\ell \circ \mathcal{F}) \big]$ |

### Constants & Estimators

| | |
|---|---|
| $k$ | The number of negative samples. |
| $\mathcal{M}$ | The upper bound on the unsupervised loss function $\ell : \mathbb{R}^k \to \mathbb{R}_+$. |
| $b$ | Bound on input's $\ell^2$ norm. $\forall \mathbf{x} \in \mathcal{S} : \|\mathbf{x}\|_2 \leqslant b$ with probability one. |
| $\eta$ | The $\ell^\infty$-Lipschitz constant of the loss function $\ell : \mathbb{R}^k \to \mathbb{R}_+$. |
| $\rho_{\min}$ | Minimum class occurrence probability $\rho_{\min} = \min_{c \in \mathcal{C}} \rho(c)$. |
| $\rho_{\max}$ | Maximum class occurrence probability $\rho_{\max} = \max_{c \in \mathcal{C}} \rho(c)$. |
| $\Lambda$ | For $\delta \in (0,1)$, $\Lambda = \sqrt{\frac{3 \ln 4|\mathcal{C}|/\delta}{\mathrm{N}\rho_{\min}}}$. |
| $\widehat{\mathrm{N}}_c$ | $\widehat{\mathrm{N}}_c = \max \left\{ 1, \min \left( \left\lfloor \frac{\mathrm{N}\rho(c)(1-\Lambda)}{2} \right\rfloor, \left\lfloor \frac{\mathrm{N}(1-\rho(c))(1-\Lambda)}{k} \right\rfloor \right) \right\}$. |
| $\widetilde{\mathrm{N}}$ | $\widetilde{\mathrm{N}} = \mathrm{N} \min \left( \frac{\rho_{\min}}{2}, \frac{1-\rho_{\max}}{k} \right)$. |

### Class of Linear Functions

$$\mathcal{F}_{\mathrm{lin}} = \left\{ x \mapsto Ax : A \in \mathbb{R}^{d \times m}, \|A^\top\|_{2,1} \leqslant a, \|A\|_\sigma \leqslant s \right\}.$$

| | |
|---|---|
| $\|\cdot\|_{2,1}$ | For $A \in \mathbb{R}^{d \times m}$, $\|A\|_{2,1} = \sum_{i=1}^d \|A_{\cdot,i}\|_2$, i.e., sum of column Euclidean norms. |
| $\|\cdot\|_\sigma$ | For $A \in \mathbb{R}^{d \times m}$, $\|A\|_\sigma$ is the spectral norm, i.e., the largest singular value of $A$. |

### Class of Neural Networks

$$\mathcal{F}_{\mathrm{nn}}^{\mathcal{A}} = \mathcal{F}_L \circ \mathcal{F}_{L-1} \circ \cdots \circ \mathcal{F}_1, \text{ where } \mathcal{F}_l = \left\{ x \mapsto \varphi_l \left( A^{(l)} x \right) : A^{(l)} \in \mathcal{B}_l \right\}.$$

| | |
|---|---|
| $L$ | Number of layers in the neural networks. |
| $d_0$ | Dimensionality of the input layer. |
| $d_l, l \in [L]$ | Dimensionality of the $l^{th}$ hidden layer. |
| $s_l, l \in [L]$ | Bounds on the $l^{th}$ layer's weight matrices' spectral norm. |
| $\mathcal{B}_l, l \in [L]$ | Space of $l^{th}$ layer's weight matrices. $\mathcal{B}_l = \left\{ A^{(l)} \in \mathbb{R}^{d_{l-1} \times d_l} : \|A^{(l)}\|_\sigma \leqslant s_l \right\}$. |
| $\xi_l, l \in [L]$ | $\ell^2$-Lipschitz constant of the $l^{th}$ layer's activation function. |
| $\varphi_l, l \in [L]$ | $\varphi_l : \mathbb{R}^{d_l} \to \mathbb{R}^{d_l}$ is the $l^{th}$ layer's activation such that $\forall \mathbf{x}, \mathbf{x}' \in \mathbb{R}^{d_l} \|\varphi_l(\mathbf{x}) - \varphi(\mathbf{x}')\|_2 \leqslant \xi_l \|\mathbf{x} - \mathbf{x}'\|_2$. |

### Covering Numbers

Let $\mathcal{G} = \left\{ g : \mathcal{Z} \to \mathbb{R}^d \right\}$ be a class of real/vector-valued functions (with $d \geqslant 1$) from an input space $\mathcal{Z}$.

Let $S = \{\mathbf{z}_1, \dots, \mathbf{z}_n\} \subset \mathcal{Z}$ be a dataset.

| | |
|---|---|
| $\mathcal{N}(\mathcal{G}, \epsilon, \|\cdot\|)$ | The covering number of a class $\mathcal{G}$ with granularity $\epsilon$ with respect to a norm $\|\cdot\|$. Specifically, if there exists a (minimum internal) cover $\mathcal{C}_{\mathcal{G}} \subset \mathcal{G}$ where $\forall g \in \mathcal{G}, \exists f \in \mathcal{C}_{\mathcal{G}} : \|f - g\| \leqslant \epsilon$, then $|\mathcal{C}_{\mathcal{G}}| = \mathcal{N}(\mathcal{G}, \epsilon, \|\cdot\|)$. |
| $\|\cdot\|_{\mathrm{L}_2(S)}$ | $\forall g, \tilde{g} \in \mathcal{G} : \|g - \tilde{g}\|_{\mathrm{L}_2(S)} = \left( \frac{1}{n} \sum_{j=1}^n \|g(\mathbf{z}_j) - \tilde{g}(\mathbf{z}_j)\|_2^2 \right)^{1/2}$. |
| $\|\cdot\|_{\mathrm{L}_{\infty,2}(S)}$ | $\forall g, \tilde{g} \in \mathcal{G} : \|g - \tilde{g}\|_{\mathrm{L}_{\infty,2}(S)} = \max_{\mathbf{z}_j \in S} \|g(\mathbf{z}_j) - \tilde{g}(\mathbf{z}_j)\|_2$. |

## B. Useful Notations

**1. Notations on datasets & tuples sets**: As a reiteration of the main text, we lay out formal definitions for the notations used throughout this work. Suppose that we are given a dataset $\mathcal{S} = \left\{ (\mathbf{x}_j, \mathbf{y}_j) \right\}_{j=1}^{N}$ drawn $i.i.d.$ from a joint distribution over the input and label spaces $\mathcal{X} \times \mathcal{C}$. We denote $\mathcal{S} = \bigcup_{c \in \mathcal{C}} \mathcal{S}_c$ where $\mathcal{S}_c$ is the set of data points belonging to class $c$:

$$\mathcal{S}_c = \left\{ (\mathbf{x}_j, \mathbf{y}_j) \in \mathcal{S} : \mathbf{y}_j = c \right\}. \tag{19}$$

With a slight abuse of notation, we also denote $\mathcal{S}_c$ as the set containing only the data points (without labels). Furthermore, for each $c \in \mathcal{C}$, we also define $\bar{\mathcal{S}}_c = \mathcal{S} \backslash \mathcal{S}_c$, i.e., the set of instances not belonging to class $c$. For every $c \in \mathcal{C}$ and $f \in \mathcal{F}$ (where $\mathcal{F}$ is the class of representation functions), we denote:

1. $N_c^+ = |\mathcal{S}_c|$, $N_c^- = |\bar{\mathcal{S}}_c|$ and $N_c = \min(\lfloor N_c^+/2 \rfloor, \lfloor N_c^-/k \rfloor)$.

2. $\mathbf{x}_i^{(c)}$ as the $i^{th}$ element of $\mathcal{S}_c$.

3. $\mathbf{x}_j^{(\bar{c})}$ as the $j^{th}$ element of $\bar{\mathcal{S}}_c$.

4. $f(\mathbf{x}_i^{(c)}) = f_{c,i}$ and $f(\mathbf{x}_j^{(\bar{c})}) = f_{\bar{c},j}$ for notational brevity.

Let $\mathcal{T} = \bigcup_{c \in \mathcal{C}} \mathcal{T}_c$ where $\mathcal{T}_c$ denotes the set of tuples whose anchor-positive pairs belong to class $c \in \mathcal{C}$. Formally:

$$\forall c \in \mathcal{C} : \mathcal{T}_c = \left\{ \left( \mathbf{x}_{j_1}^{(c)}, \mathbf{x}_{j_2}^{(c)}, \mathbf{x}_{l_1}^{(\bar{c})}, \ldots, \mathbf{x}_{l_k}^{(\bar{c})} \right) : 1 \leqslant j_1, j_2 \leqslant N_c^+, 1 \leqslant l_1, \ldots, l_k \leqslant N_c^- \right\}. \tag{20}$$

Additionally, for all $c \in \mathcal{C}$, we also define $\mathcal{T}_c^{\text{iid}}$ as the set of independent tuples selected in a "greedy" way such that every tuple in $\mathcal{T}_c^{\text{iid}}$ has an anchor-positive pair belonging to class $c$ (See also Figure 1 for visual illustration). Formally:

$$\forall c \in \mathcal{C} : \mathcal{T}_c^{\text{iid}} = \left\{ \left( \mathbf{x}_{2j-1}^{(c)}, \mathbf{x}_{2j}^{(c)}, \mathbf{x}_{kj-k+1}^{(\bar{c})}, \ldots, \mathbf{x}_{kj}^{(\bar{c})} \right) \right\}_{j=1}^{N_c}. \tag{21}$$

Furthermore, as we progress through the proofs, we also often use the auxiliary datasets $\widetilde{\mathcal{T}}_c^{\text{iid}}$, which contains all vectors including anchors, positive and negative samples from the tuples set $\mathcal{T}_c^{\text{iid}}$. Formally:

$$\forall c \in \mathcal{C} : \widetilde{\mathcal{T}}_c^{\text{iid}} = \bigcup_{T \in \mathcal{T}_c^{\text{iid}}} \bigcup_{\mathbf{x} \in T} \{\mathbf{x}\}. \tag{22}$$

**2. Notations on loss function class**: Let $\ell : \mathbb{R}^k \to [0, \mathcal{M}]$ be a contrastive loss function and $\mathcal{F}$ be a class of representation functions. With a slight abuse of notation (of the composition operator "$\circ$"), we define the class of loss functions as $\ell \circ \mathcal{F}$ formally as follows:

$$\ell \circ \mathcal{F} = \left\{ (\mathbf{x}, \mathbf{x}^+, \mathbf{x}_i^-, \ldots, \mathbf{x}_k^-) \mapsto \ell \left( \left\{ f(\mathbf{x})^\top \left( f(\mathbf{x}^+) - f(\mathbf{x}_i^-) \right) \right\}_{i=1}^k \right) : f \in \mathcal{F} \right\}. \tag{23}$$

**3. Notations related to Rademacher complexitites**: Now, we are ready to formally define the quantities related to Rademacher complexity used throughout the proofs. For the following definitions, we use the notation $\mathbb{E}_X[\cdot]$ to refer to the expectation taken over the distribution of a random variable $X$ and $\mathbb{E}_{X|Y}[\cdot]$ to refer to the conditional expectation over the distribution of $X$ conditioned on another random variable $Y$. For every $c \in \mathcal{C}$, let $\boldsymbol{\Sigma}_{N_c} = \left\{ \sigma_1, \ldots, \sigma_{N_c} \right\}$ be the vector of $N_c$ independent Rademacher variables and $\mathcal{T}_c^{\text{iid}}$ be define in (21), we have:

$$\mathcal{R}_{\mathcal{F}}^{\mathcal{T}_c^{\text{iid}}, \boldsymbol{\Sigma}_{N_c}} = \sup_{f \in \mathcal{F}} \left| \frac{1}{N_c} \sum_{j=1}^{N_c} \sigma_j \ell \left( \left\{ f_{c,2j-1}^\top \left[ f_{c,2j} - f_{\bar{c},kj-k+i} \right] \right\}_{i=1}^k \right) \right|, \qquad \text{(when } N_c \geqslant 1\text{)}$$

$$\widehat{\mathfrak{R}}_{\mathcal{T}_c^{\text{iid}}}(\ell \circ \mathcal{F}) = \begin{cases} \mathbb{E}_{\boldsymbol{\Sigma}_{N_c}|\mathcal{S}, N_c^+} \left[ \mathcal{R}_{\mathcal{F}}^{\mathcal{T}_c^{\text{iid}}, \boldsymbol{\Sigma}_{N_c}} \right] & \text{when } N_c \geqslant 1 \\ 0 & \text{when } N_c = 0 \end{cases}, \qquad \text{(Empirical Rademacher Complexity)} \tag{24}$$

$$\mathfrak{R}_{N_c}(\ell \circ \mathcal{F}) = \mathbb{E}_{\mathcal{S}|N_c^+} [\widehat{\mathfrak{R}}_{\mathcal{T}_c^{\text{iid}}}(\ell \circ \mathcal{F})]. \qquad \text{(Expected Rademacher Complexity)}$$

The vector $\mathbf{N} = \{N_c^+\}_{c \in \mathcal{C}}$ represents the sample sizes and $\mathbf{N} \sim \mathrm{Multinom}(N, \{\rho(c)\}_{c \in \mathcal{C}})$. Hence, even though we denote $\mathfrak{R}_{\mathbf{N}_c}(\ell \circ \mathcal{F})$ as the "expected" Rademacher complexity, it is still a random variable due to the randomness in $\mathbf{N}$. However, as we will see in subsequent proofs of the main results, we will eventually handle the randomness by analysing the concentration of $\mathbf{N}$ around the respective expectations $\{N\rho(c)\}_{c \in \mathcal{C}}$.

Throughout the main results, we will often encounter the assumption that for all $c \in \mathcal{C}$, the empirical Rademacher complexities $\overline{N}_c^{\frac{1}{2}} \widehat{\mathfrak{R}}_{\widetilde{\mathcal{T}}_c^{\mathrm{iid}}}(\ell \circ \mathcal{F})$ are upper bounded by some $K_{\mathcal{F},c}$ depending on both the class of representation functions $\mathcal{F}$ and the class $c$. In this work, we use Dudley entropy integral bound to derive the upper bounds $K_{\mathcal{F},c}$. Specifically, by Theorem F.5, for any choice of $\alpha > 0$, we can generally upper bound $\overline{N}_c^{\frac{1}{2}} \widehat{\mathfrak{R}}_{\widetilde{\mathcal{T}}_c^{\mathrm{iid}}}(\ell \circ \mathcal{F})$ by setting:

$$K_{\mathcal{F},c} = 4\alpha + 12 \int_\alpha^{\mathcal{M}} \ln^{\frac{1}{2}} 2\mathcal{N}(\ell \circ \mathcal{F}, \epsilon, \mathrm{L}_2(\widetilde{\mathcal{T}}_c^{\mathrm{iid}})) d\epsilon,$$

where $\mathcal{N}(\ell \circ \mathcal{F}, \epsilon, \mathrm{L}_2(\widetilde{\mathcal{T}}_c^{\mathrm{iid}}))$ is the $\mathrm{L}_2$ covering number of the loss class $\ell \circ \mathcal{F}$ restricted to the vector dataset $\widetilde{\mathcal{T}}_c^{\mathrm{iid}}$ (See definition D.11 for the formal definition of $\mathrm{L}_2$ norm).

For a more comprehensive summary of all the notations used throughout this paper, we refer the readers to Table 2 for quick references.

## C. U-Statistics Revisited

In this section, we revisit the definition of U-Statistics and the decoupling technique which will be used throughout the proofs of the main results.

**Definition C.1.** (U-Statistics) Given $S = \{\mathbf{x}_1, \ldots, \mathbf{x}_n\}$ drawn $i.i.d.$ from a distribution $\mathcal{P}$ over $\mathcal{X}$. Let $h : \mathcal{X}^m \to \mathbb{R}$ be a symmetric kernel in its arguments. Then, the U-Statistics $U_n(h)$ used to estimate $\theta = \mathbb{E}_{\mathbf{x}_1, \ldots, \mathbf{x}_m \sim \mathcal{P}^m}[h(\mathbf{x}_1, \ldots, \mathbf{x}_m)]$ is defined as follows:

$$\mathrm{U}_n(h) = \frac{1}{\binom{n}{m}} \sum_{i_1, \ldots, i_m \in \mathrm{C}_m[n]} h(\mathbf{x}_{i_1}, \ldots, \mathbf{x}_{i_m}), \tag{25}$$

where $\mathrm{C}_m[n]$ denotes the set of $m$-tuples selected (without replacement) from $[n]$ without order ($m$-combinations). We call $U_n(h)$ an one-sample U-Statistics of order $m$. Furthermore, let $q = \lfloor n/m \rfloor$, the *decoupled* form of $U_n(h)$ is:

$$\mathrm{U}_n(h) = \frac{1}{n!} \sum_{\pi \in \mathrm{S}[n]} V_\pi(S),$$
$$V_\pi(S) = \frac{1}{q} \sum_{i=1}^q h(\mathbf{x}_{\pi[mi-m+1]}, \ldots, \mathbf{x}_{\pi[mi]}), \tag{26}$$

**Remark C.2.** For any $1 \leqslant i \leqslant q$, we can write $\mathrm{U}_n(h) = \frac{1}{n!} \sum_{\pi \in \mathrm{S}[n]} h(\mathbf{x}_{\pi[mi-m+1]}, \ldots, \mathbf{x}_{\pi[mi]})$. Specifically,

$$\frac{1}{n!} \sum_{\pi \in \mathrm{S}[n]} h(\mathbf{x}_{\pi[mi-m+1]}, \ldots, \mathbf{x}_{\pi[mi]}) = \frac{1}{n!} \sum_{j_1, \ldots, j_m \in \mathrm{P}_m[n]} (n-m)! h(\mathbf{x}_{j_1}, \ldots, \mathbf{x}_{j_m})$$

$$= \frac{1}{n!} \sum_{i_1, \ldots, i_m \in \mathrm{C}_m[n]} m!(n-m)! h(\mathbf{x}_{i_1}, \ldots, \mathbf{x}_{i_m})$$

$$= \underbrace{\frac{1}{\binom{n}{m}} \sum_{i_1, \ldots, i_m \in \mathrm{C}_m[n]} h(\mathbf{x}_{i_1}, \ldots, \mathbf{x}_{i_m})}_{\text{Eqn. (25)}}.$$

Hence, by swapping the summations, the decoupled formula of $\mathrm{U}_n(h)$ can be understood as summing the same quantity $q$

times then dividing by $q$:

$$\underbrace{\frac{1}{n!}\sum_{\pi\in S[n]}\frac{1}{q}\sum_{i=1}^{q}h(\mathbf{x}_{\pi[mi-m+1]},\dots,\mathbf{x}_{\pi[mi]})}_{\text{Eqn. (26)}} = \frac{1}{q}\sum_{i=1}^{q}\underbrace{\frac{1}{n!}\sum_{\pi\in S[n]}h(\mathbf{x}_{\pi[mi-m+1]},\dots,\mathbf{x}_{\pi[mi]})}_{U_n(h)}$$

$$= \frac{1}{q}\sum_{i=1}^{q}U_n(h) = U_n(h).$$

## D. Generalization Bound for Empirical Risk Minimizer of U-Statistics

### D.1. Formulation of U-Statistics

Let $N_c = \min\left(\lfloor N_c^+/2\rfloor, \lfloor N_c^-/k\rfloor\right)$, which represents the maximum number of valid $i.i.d.$ tuples that can be formed for contrastive learning from the pool of training data. We restate the definition of the U-Statistics for $L_{\mathrm{un}}(f|c)$ as follows:

$$\mathcal{U}_{\mathrm{N}}(f|c) = \mathbb{1}_{\{N_c\geqslant 1\}}U_{\mathrm{N}}(f|c),$$

$$\text{where } U_{\mathrm{N}}(f|c) = \frac{1}{|\mathcal{T}_c|}\sum_{\substack{j_1,j_2\in P_2[N_c^+]\\ l_1,\dots,l_k\in C_k[N_c^-]}}\ell\left(\left\{f_{c,j_1}^\top\left[f_{c,j_2}-f_{\bar{c},l_i}\right]\right\}_{i=1}^{k}\right). \tag{27}$$

From the above equation, when $N_c = 0$, there is no valid tuple where the anchor-positive pair belongs to class $c$ (either the number of instances from class $c$ or outside of class $c$ is not sufficient). In such case, we adopt a naive estimate $\mathcal{U}_{\mathrm{N}}(f|c) = 0$. On the other hand, when $N_c \geqslant 1$, it is natural to estimate $L_{\mathrm{un}}(f|c)$ as the average loss over all valid tuples (not necessarily $i.i.d.$) where the anchor-positive pairs belong to class $c$. Finally, we define the overall U-Statistics for $L_{\mathrm{un}}(f)$ as follows:

$$\mathcal{U}_{\mathrm{N}}(f) = \sum_{c\in\mathcal{C}}\frac{N_c^+}{N}\mathcal{U}_{\mathrm{N}}(f|c). \tag{28}$$

**Remark D.1.** It is worth mentioning that the above formulation is asymptotically unbiased, i.e., $\lim_{N\to\infty}\mathcal{U}_{\mathrm{N}}(f) = L_{\mathrm{un}}(f)$. To demonstrate this briefly, denote $\mathcal{N}$ as the event that for all classes, there are enough data points to form at least one tuple, i.e., $\mathcal{N} = \left\{\forall c\in\mathcal{C}, N_c\geqslant 1\right\}$. Hence, $\lim_{N\to\infty}\mathbb{P}(\mathcal{N}) = 1$. Furthermore, we have:

$$\mathbb{E}[\mathcal{U}_{\mathrm{N}}(f)] = \mathbb{P}(\mathcal{N})\mathbb{E}[\mathcal{U}_{\mathrm{N}}(f)|\mathcal{N}] + \mathbb{P}(\mathcal{N}^c)\mathbb{E}[\mathcal{U}_{\mathrm{N}}(f)|\mathcal{N}^c]$$

$$= \mathbb{P}(\mathcal{N})\mathbb{E}\left[\sum_{c\in\mathcal{C}}\frac{N_c^+}{N}U_{\mathrm{N}}(f|c)\Big|\mathcal{N}\right] + \mathbb{P}(\mathcal{N}^c)\mathbb{E}[\mathcal{U}_{\mathrm{N}}(f)|\mathcal{N}^c]$$

$$= \mathbb{P}(\mathcal{N})\sum_{c\in\mathcal{C}}\rho(c)L_{\mathrm{un}}(f|c) + \mathbb{P}(\mathcal{N}^c)\mathbb{E}[\mathcal{U}_{\mathrm{N}}(f)|\mathcal{N}^c]$$

$$= \mathbb{P}(\mathcal{N})L_{\mathrm{un}}(f) + \mathbb{E}[\mathcal{U}_{\mathrm{N}}(f)|\mathcal{N}^c]\mathbb{P}(\mathcal{N}^c)$$

$$\to L_{\mathrm{un}}(f) \quad \text{as} \quad N\to\infty.$$

**Remark D.2.** Let $U_{\mathrm{N}}(f|c)$ be defined in Eqn. (27). With a similar decoupling argument to de la Peña & Giné (1998) and given that $N_c \geqslant 1$, we can re-write $U_{\mathrm{N}}(f|c)$ as follows:

$$U_{\mathrm{N}}(f|c) = \frac{1}{N_c^+!\times N_c^-!}\sum_{\pi_c\in S[N_c^+],\bar{\pi}_c\in S[N_c^-]}V_{\pi_c,\bar{\pi}_c}(f|c),$$

$$\text{where } V_{\pi_c,\bar{\pi}_c}(f|c) = \frac{1}{N_c}\sum_{j=1}^{N_c}\ell\left(\left\{f_{c,\pi_c[2j-1]}^\top\left[f_{c,\pi_c[2j]}-f_{\bar{c},\bar{\pi}_c[kj-k+i]}\right]\right\}_{i=1}^{k}\right), \tag{29}$$

where $S[n] = \left\{\pi : [n]\to[n]\big|\pi \text{ is bijective}\right\}$ denotes the set of all possible permutations of the indices set $[n]$ for any $n\in\mathbb{N}$. $V_{\pi_c,\bar{\pi}_c}(f|c)$ is the average over the loss evaluated on the set of independent tuples $\mathcal{T}_c^{\mathrm{iid}}[\pi_c,\bar{\pi}_c]$ selected as follows:

1. Shuffle the set of inputs $\mathcal{S}_c$ according to the permutation $\pi_c \in \mathrm{S}[\mathrm{N}_c^+]$, denote the resulting set as $\mathcal{S}_c[\pi_c]$.

2. Shuffle the set of inputs $\bar{\mathcal{S}}_c$ according to the permutation $\bar{\pi}_c \in \mathrm{S}[\mathrm{N}_c^-]$, denote the resulting set as $\bar{\mathcal{S}}_c[\bar{\pi}_c]$.

3. Pair 2-element blocks from $\mathcal{S}_c[\pi_c]$ to $k$-element blocks from $\bar{\mathcal{S}}_c[\bar{\pi}_c]$ until either set runs out of independent blocks.

$$
\mathcal{T}_c^{\mathrm{iid}}[\pi_c, \bar{\pi}_c] = \left\{ \left( \mathbf{x}_{\pi_c[2j-1]}^{(c)}, \mathbf{x}_{\pi_c[2j]}^{(c)}, \mathbf{x}_{\bar{\pi}_c[kj-k+1]}^{(\bar{c})}, \dots, \mathbf{x}_{\bar{\pi}_c[kj]}^{(\bar{c})} \right) \right\}_{j=1}^{\mathrm{N}_c}. \tag{30}
$$

Essentially, the formation of the tuples set $\mathcal{T}_c^{\mathrm{iid}}[\pi_c, \bar{\pi}_c]$ is identical to that of $\mathcal{T}_c^{\mathrm{iid}}$ except $\mathcal{S}_c$ and $\bar{\mathcal{S}}_c$ are shuffled according to permutations $\pi_c$ and $\bar{\pi}_c$ beforehand. To briefly demonstrate how the representation of U-Statistics in Eqn. (29) holds, for any two data points $\mathbf{x}_{j_1}^{(c)}, \mathbf{x}_{j_2}^{(c)} \in \mathcal{S}_c$, we define:

$$
\begin{aligned}
h(\mathbf{x}_{j_1}^{(c)}, \mathbf{x}_{j_2}^{(c)}) &= \frac{1}{|\mathrm{C}_k[n]|} \sum_{l_1,\dots,l_k \in \mathrm{C}_k[n]} \ell\left( \left\{ f_{c,j_1}^\top \left[ f_{c,j_2} - f_{\bar{c},l_i} \right] \right\}_{i=1}^k \right) \\
&= \frac{1}{\mathrm{N}_c^-!} \sum_{\bar{\pi}_c \in \mathrm{S}[\mathrm{N}_c^-]} \ell\left( \left\{ f_{c,j_1}^\top \left[ f_{c,j_2} - f_{\bar{c},\bar{\pi}_c[km-k+i]} \right] \right\}_{i=1}^k \right),
\end{aligned} \tag{31}
$$

for any $1 \leqslant m \leqslant \mathrm{N}_c$ (by remark C.2). Then, using the same decoupling argument as de la Peña & Giné (1998), we can write $\mathrm{U}_\mathrm{N}(f|c)$ as an average over $h(\mathbf{x}_{j_1}^{(c)}, \mathbf{x}_{j_2}^{(c)})$ as follows:

$$
\begin{aligned}
\mathrm{U}_\mathrm{N}(f|c) &= \frac{1}{\mathrm{P}_2[\mathrm{N}_c^+]} \sum_{j_1,j_2 \in \mathrm{P}_2[\mathrm{N}_c^+]} h(\mathbf{x}_{j_1}^{(c)}, \mathbf{x}_{j_2}^{(c)}) \\
&= \frac{1}{\mathrm{N}_c^+!} \sum_{\pi_c \in \mathrm{S}[\mathrm{N}_c^+]} \frac{1}{\mathrm{N}_c} \sum_{j=1}^{\mathrm{N}_c} h(\mathbf{x}_{\pi_c[2j-1]}^{(c)}, \mathbf{x}_{\pi_c[2j]}^{(c)}) \\
&= \frac{1}{\mathrm{N}_c^+!} \sum_{\pi_c \in \mathrm{S}[\mathrm{N}_c^+]} \frac{1}{\mathrm{N}_c} \sum_{j=1}^{\mathrm{N}_c} \frac{1}{\mathrm{N}_c^-!} \sum_{\bar{\pi}_c \in \mathrm{S}[\mathrm{N}_c^-]} \ell\left( \left\{ f_{c,\pi_c[2j-1]}^\top \left[ f_{c,\pi_c[2j]} - f_{\bar{c},\bar{\pi}_c[kj-k+i]} \right] \right\}_{i=1}^k \right) \\
&= \frac{1}{\mathrm{N}_c^+! \times \mathrm{N}_c^-!} \sum_{\substack{\pi_c \in \mathrm{S}[\mathrm{N}_c^+] \\ \bar{\pi}_c \in \mathrm{S}[\mathrm{N}_c^-]}} \frac{1}{\mathrm{N}_c} \sum_{j=1}^{\mathrm{N}_c} \ell\left( \left\{ f_{c,\pi_c[2j-1]}^\top \left[ f_{c,\pi_c[2j]} - f_{\bar{c},\bar{\pi}_c[kj-k+i]} \right] \right\}_{i=1}^k \right).
\end{aligned}
$$

**Remark D.3.** Let $g$ be a real-valued function. For any distinct pairs of indices re-arrangements $\pi_c, \pi_c' \in \mathrm{S}[\mathrm{N}_c^+]$ and $\bar{\pi}_c, \bar{\pi}_c' \in \mathrm{S}[\mathrm{N}_c^-]$, we have $\mathbb{E}_{\mathcal{S}|\mathrm{N}_c^+} g(V_{\pi_c, \bar{\pi}_c}(f|c)) = \mathbb{E}_{\mathcal{S}|\mathrm{N}_c^+} g(V_{\pi_c', \bar{\pi}_c'}(f|c))$.

### D.2. Proof of the Main Results

**Lemma D.4.** *Let $\varphi : \mathbb{R}_+ \to \mathbb{R}$ be a convex, non-decreasing function. Let $\mathbf{N} = \{\mathrm{N}_c^+\}_{c \in \mathcal{C}} \sim \mathrm{Multinom}(\mathrm{N}, \{\rho(c)\}_{c \in \mathcal{C}})$ be the multinomial random variable representing the class-wise sample sizes. Suppose that $\mathrm{N}_c \geqslant 1$ and $\boldsymbol{\Sigma}_{\mathrm{N}_c}$ denote the sequence of $\mathrm{N}_c$ i.i.d. Rademacher random variables. Then, for every $c \in \mathcal{C}$, we have:*

$$
\mathbb{E}_{\mathcal{S}|\mathrm{N}_c^+}\left[ \varphi\left( \sup_{f \in \mathcal{F}} \left| \mathrm{U}_\mathrm{N}(f|c) - \mathrm{L}_{\mathrm{un}}(f|c) \right| \right) \right] \leqslant \mathbb{E}_{\mathcal{S}, \boldsymbol{\Sigma}_{\mathrm{N}_c}|\mathrm{N}_c^+} \varphi\left[ 2\mathcal{R}_{\mathcal{F}}^{\mathcal{T}_c^{\mathrm{iid}}, \boldsymbol{\Sigma}_{\mathrm{N}_c}} \right], \tag{32}
$$

*where $\mathbb{E}_{\mathcal{S}|\mathrm{N}_c^+}$ denotes the expectation taken over the sample $\mathcal{S}$ conditioned on the random variable $\mathrm{N}_c^+$ and the random variable $\mathcal{R}_{\mathcal{F}}^{\mathcal{T}_c^{\mathrm{iid}}, \boldsymbol{\Sigma}_{\mathrm{N}_c}}$ is defined in Eqn. (24).*

**Proof.** We have:

$$\mathbb{E}_{\mathcal{S}|\mathrm{N}_c^+}\Big[\varphi\Big(\sup_{f\in\mathcal{F}}\Big|\mathrm{U}_\mathrm{N}(f|c)-\mathrm{L}_\mathrm{un}(f|c)\Big|\Big)\Big]$$

$$=\mathbb{E}_{\mathcal{S}|\mathrm{N}_c^+}\varphi\Bigg[\frac{1}{|\mathcal{T}_c|}\sup_{f\in\mathcal{F}}\Bigg|\sum_{\substack{j_1,j_2\in\mathrm{P}_2[\mathrm{N}_c^+]\\l_1,\dots,l_k\in\mathrm{C}_k[\mathrm{N}_c^-]}}\Big[\ell\Big(\big\{f_{c,j_1}^\top\big[f_{c,j_2}-f_{\bar{c},l_i}\big]\big\}_{i=1}^k\Big)-\mathrm{L}_\mathrm{un}(f|c)\Big]\Bigg|\Bigg]$$

$$=\mathbb{E}_{\mathcal{S}|\mathrm{N}_c^+}\varphi\Bigg[\frac{1}{\mathrm{N}_c^+!\times\mathrm{N}_c^-!}\sup_{f\in\mathcal{F}}\Bigg|\sum_{\pi_c\in\mathrm{S}[\mathrm{N}_c^+],\bar{\pi}_c\in\mathrm{S}[\mathrm{N}_c^-]}\Big[V_{\pi_c,\bar{\pi}_c}(f|c)-\mathrm{L}_\mathrm{un}(f|c)\Big]\Bigg|\Bigg]$$

$$\leqslant\mathbb{E}_{\mathcal{S}|\mathrm{N}_c^+}\varphi\Bigg[\frac{1}{\mathrm{N}_c^+!\times\mathrm{N}_c^-!}\sup_{f\in\mathcal{F}}\sum_{\pi_c\in\mathrm{S}[\mathrm{N}_c^+],\bar{\pi}_c\in\mathrm{S}[\mathrm{N}_c^-]}\Big|V_{\pi_c,\bar{\pi}_c}(f|c)-\mathrm{L}_\mathrm{un}(f|c)\Big|\Bigg]\quad\Big(\text{Triangle inequality}\Big)$$

$$\leqslant\mathbb{E}_{\mathcal{S}|\mathrm{N}_c^+}\varphi\Bigg[\frac{1}{\mathrm{N}_c^+!\times\mathrm{N}_c^-!}\sum_{\pi_c\in\mathrm{S}[\mathrm{N}_c^+],\bar{\pi}_c\in\mathrm{S}[\mathrm{N}_c^-]}\sup_{f\in\mathcal{F}}\Big|V_{\pi_c,\bar{\pi}_c}(f|c)-\mathrm{L}_\mathrm{un}(f|c)\Big|\Bigg]\quad\Big(\sup\sum\leqslant\sum\sup,\varphi\text{ non-decreasing}\Big)$$

$$\leqslant\frac{1}{\mathrm{N}_c^+!\times\mathrm{N}_c^-!}\sum_{\pi_c\in\mathrm{S}[\mathrm{N}_c^+],\bar{\pi}_c\in\mathrm{S}[\mathrm{N}_c^-]}\mathbb{E}_{\mathcal{S}|\mathrm{N}_c^+}\varphi\Big[\sup_{f\in\mathcal{F}}\Big|V_{\pi_c,\bar{\pi}_c}(f|c)-\mathrm{L}_\mathrm{un}(f|c)\Big|\Big]\quad\text{(Jensen's Inequality)}$$

$$=\mathbb{E}_{\mathcal{S}|\mathrm{N}_c^+}\varphi\Bigg[\sup_{f\in\mathcal{F}}\Bigg|\frac{1}{\mathrm{N}_c}\sum_{j=1}^{\mathrm{N}_c}\ell\Big(\big\{f_{c,2j-1}^\top\big[f_{c,2j}-f_{\bar{c},kj-k+i}\big]\big\}_{i=1}^k\Big)-\mathrm{L}_\mathrm{un}(f|c)\Bigg|\Bigg]\quad\text{(Remark D.3)}.$$

When conditioning on a given sample size $\mathrm{N}_c^+$ of class $c\in\mathcal{C}$, $\bar{\mathcal{S}}_c$ is an *i.i.d.* sample drawn $\mathrm{N}-\mathrm{N}_c^+$ times from the distribution $\bar{\mathcal{D}}_c$. Hence, for any $1\leqslant j\leqslant\mathrm{N}_c$, we have $\mathbb{E}_{\mathcal{S}|\mathrm{N}_c^+}\ell\Big(\big\{f_{c,2j-1}^\top\big[f_{c,2j}-f_{\bar{c},kj-k+i}\big]\big\}_{i=1}^k\Big)=\mathrm{L}_\mathrm{un}(f|c)$. Therefore, by the symmetrization trick (Lemma F.6), we have:

$$\mathbb{E}_{\mathcal{S}|\mathrm{N}_c^+}\Big[\varphi\Big(\sup_{f\in\mathcal{F}}\Big|\mathrm{U}_\mathrm{N}(f|c)-\mathrm{L}_\mathrm{un}(f|c)\Big|\Big)\Big]$$

$$\leqslant\mathbb{E}_{\mathcal{S}|\mathrm{N}_c^+}\varphi\Bigg[\sup_{f\in\mathcal{F}}\Bigg|\frac{1}{\mathrm{N}_c}\sum_{j=1}^{\mathrm{N}_c}\ell\Big(\big\{f_{c,2j-1}^\top\big[f_{c,2j}-f_{\bar{c},kj-k+i}\big]\big\}_{i=1}^k\Big)-\mathrm{L}_\mathrm{un}(f|c)\Bigg|\Bigg]$$

$$\leqslant\mathbb{E}_{\mathcal{S},\mathbf{\Sigma}_{\mathrm{N}_c}|\mathrm{N}_c^+}\varphi\Big[2\mathcal{R}_{\mathcal{F}}^{\mathcal{T}_c^\mathrm{iid},\mathbf{\Sigma}_{\mathrm{N}_c}}\Big].$$

Hence, we obtained the desired bound. $\qquad\square$

**Proposition D.5.** *Let* $\mathbf{N}=\{\mathrm{N}_c^+\}_{c\in\mathcal{C}}\sim\mathrm{Multinom}(\mathrm{N},\{\rho(c)\}_{c\in\mathcal{C}})$ *be the multinomial random variable representing the class-wise sample sizes. Let* $\mathcal{F}$ *be a class of representation functions. For any* $\delta\in(0,1)$*, the following inequality holds:*

$$\mathbb{P}\Bigg(\sup_{f\in\mathcal{F}}\Big|\mathcal{U}_\mathrm{N}(f|c)-\mathrm{L}_\mathrm{un}(f|c)\Big|\leqslant2\mathfrak{R}_{\mathrm{N}_c}(\ell\circ\mathcal{F})+8\mathcal{M}\sqrt{\frac{\ln2/\delta}{2\overline{\mathrm{N}}_c}}\Bigg|\mathrm{N}_c^+\Bigg)\geqslant1-\delta. \tag{33}$$

*where* $\overline{\mathrm{N}}_c=\max(1,\mathrm{N}_c)$ *and* $\mathfrak{R}_{\mathrm{N}_c}(\ell\circ\mathcal{F})$ *is the expected Rademacher complexity (Eqn. (24)).*

**Proof.** We divide the proof into two cases when $\mathrm{N}_c=0$ and when $\mathrm{N}_c\geqslant1$.

1. **When** $\mathrm{N}_c=0$: Then, we have $\mathcal{U}_\mathrm{N}(f|c)=0$ by default and $\sup_{f\in\mathcal{F}}\Big|\mathcal{U}_\mathrm{N}(f|c)-\mathrm{L}_\mathrm{un}(f|c)\Big|\leqslant\mathcal{M}$. Furthermore, since $8\mathcal{M}\sqrt{\frac{\ln2/\delta}{2}}>4\mathcal{M}$, the bound holds trivially.

2. **When** $N_c \geqslant 1$: By lemma D.4, let $\varphi(x) = \exp(tx)$ for $t > 0$, we have:

$$\mathbb{E}_{\mathcal{S}|N_c^+} \exp \left( t \sup_{f \in \mathcal{F}} \left| U_N(f|c) - L_{un}(f|c) \right| \right) \leqslant \mathbb{E}_{\mathcal{S}, \boldsymbol{\Sigma}_n | N_c^+} \exp \left( 2t \mathcal{R}_{\mathcal{F}}^{\mathcal{T}_c^{iid}, \boldsymbol{\Sigma}_{N_c}} \right)$$

$$\leqslant \exp \left( 2t \mathfrak{R}_{N_c}(\ell \circ \mathcal{F}) + \frac{32t^2 \mathcal{M}^2}{N_c} \right). \quad \text{(Lemma F.9)}$$

Using Markov's Inequality, for any $\lambda > 0$, we have:

$$\mathbb{P} \left( \sup_{f \in \mathcal{F}} \left| U_N(f|c) - L_{un}(f|c) \right| \geqslant \lambda \middle| N_c^+ \right) = \mathbb{P} \left( \exp \left( t \sup_{f \in \mathcal{F}} \left| U_N(f|c) - L_{un}(f|c) \right| \right) \geqslant e^{t\lambda} \middle| N_c^+ \right)$$

$$\leqslant e^{-t\lambda} \mathbb{E}_{\mathcal{S}|N_c^+} \exp \left( t \sup_{f \in \mathcal{F}} \left| U_N(f|c) - L_{un}(f|c) \right| \right)$$

$$\leqslant \exp \left( 2t \mathfrak{R}_{N_c}(\ell \circ \mathcal{F}) + \frac{32t^2 \mathcal{M}^2}{N_c} - t\lambda \right).$$

Setting $\delta = \exp \left( 2t \mathfrak{R}_{N_c}(\ell \circ \mathcal{F}) + \frac{32t^2 \mathcal{M}^2}{N_c} - t\lambda \right)$ and solve for $\lambda$, we have:

$$\lambda = 2\mathfrak{R}_{N_c}(\ell \circ \mathcal{F}) + \frac{32t\mathcal{M}^2}{N_c} + \frac{\ln 1/\delta}{t}.$$

Using the Lagrange multiplier to solve for the optimal value of $t$, we have $t = \frac{\sqrt{N_c \ln 1/\delta}}{4\mathcal{M}\sqrt{2}}$. Plugging the value of $t$ back to the formula of $\lambda$:

$$\sup_{f \in \mathcal{F}} \left| U_N(f|c) - L_{un}(f|c) \right| \leqslant 2\mathfrak{R}_{N_c}(\ell \circ \mathcal{F}) + 8\mathcal{M} \sqrt{\frac{\ln 1/\delta}{2N_c}}$$

$$= 2\mathfrak{R}_{N_c}(\ell \circ \mathcal{F}) + 8\mathcal{M} \sqrt{\frac{\ln 1/\delta}{2\overline{N}_c}}$$

$$\leqslant 2\mathfrak{R}_{N_c}(\ell \circ \mathcal{F}) + 8\mathcal{M} \sqrt{\frac{\ln 2/\delta}{2\overline{N}_c}},$$

with probability of at least $1 - \delta$, as desired.

$\square$

**Lemma D.6.** *Let* $N \sim \text{Multinom}(N, \{\rho(c)\}_{c \in \mathcal{C}})$ *be the multinomial random variable representing the class-wise sample sizes. Then, for any* $\delta \in (0, 1)$*, the events* $\left| \frac{N_c^+}{N} - \rho(c) \right| \leqslant \rho(c) \sqrt{\frac{3 \ln 2|\mathcal{C}|/\delta}{N\rho_{\min}}}$ *hold simultaneously for all* $c \in \mathcal{C}$ *with probability of at least* $1 - \delta$*. In particular, on the same high probability event:*

$$\sum_{c \in \mathcal{C}} \left| \frac{N_c^+}{N} - \rho(c) \right| \leqslant \sqrt{\frac{3 \ln 2|\mathcal{C}|/\delta}{N\rho_{\min}}}. \tag{34}$$

**Proof.** Recall that $N_c^+ \sim \text{Bin}(N, \rho(c))$. Hence, $\mathbb{E}[N_c^+] = N\rho(c)$. Then, let $0 < \lambda < 1$, by the two-sided multiplicative Chernoff bound:

$$\mathbb{P} \left( \left| \frac{N_c^+}{N} - \rho(c) \right| \geqslant \lambda \rho(c) \right) = \mathbb{P} \left( \left| N_c^+ - N\rho(c) \right| \geqslant \lambda N\rho(c) \right) \leqslant 2 \exp \left( -\frac{\lambda^2 N\rho(c)}{3} \right).$$

Then, by the Union bound, we have:

$$\mathbb{P}\left(\sum_{c\in\mathcal{C}}\left|\frac{\mathrm{N}_c^+}{\mathrm{N}}-\rho(c)\right|\geqslant\lambda\right)\leqslant\mathbb{P}\left(\bigcup_{c\in\mathcal{C}}\left\{\left|\frac{\mathrm{N}_c^+}{\mathrm{N}}-\rho(c)\right|\geqslant\rho(c)\lambda\right\}\right)$$

$$\leqslant\sum_{c\in\mathcal{C}}\mathbb{P}\left(\left|\frac{\mathrm{N}_c^+}{\mathrm{N}}-\rho(c)\right|\geqslant\rho(c)\lambda\right)$$

$$\leqslant2\sum_{c\in\mathcal{C}}\exp\left(-\frac{\lambda^2\mathrm{N}\rho(c)}{3}\right)$$

$$\leqslant2|\mathcal{C}|\exp\left(-\frac{\lambda^2\mathrm{N}\rho_{\min}}{3}\right).$$

Setting $\delta=2|\mathcal{C}|\exp\left(-\frac{\lambda^2\mathrm{N}\rho_{\min}}{3}\right)$, we have $\lambda=\sqrt{\frac{3\ln 2|\mathcal{C}|/\delta}{\mathrm{N}\rho_{\min}}}$. Therefore, with probability of at least $1-\delta$,

$$\sum_{c\in\mathcal{C}}\left|\frac{\mathrm{N}_c^+}{\mathrm{N}}-\rho(c)\right|\leqslant\sqrt{\frac{3\ln 2|\mathcal{C}|/\delta}{\mathrm{N}\rho_{\min}}}$$

and the events $\left|\frac{\mathrm{N}_c^+}{\mathrm{N}}-\rho(c)\right|\leqslant\rho(c)\sqrt{\frac{3\ln 2|\mathcal{C}|/\delta}{\mathrm{N}\rho_{\min}}}$ hold simultaneously for all $c\in\mathcal{C}$, as desired. $\qquad\square$

**Proposition D.7.** *Let $\mathcal{F}$ be a class of representation functions and $\widehat{f}_{\mathcal{U}}=\arg\min_{f\in\mathcal{F}}\mathcal{U}_{\mathrm{N}}(f)$ be the empirical minimizer of the U-Statistics (Eqn. (28)). Then, for any $\delta\in(0,1)$, with probability of at least $1-\delta$, we have:*

$$\mathrm{L}_{\mathrm{un}}(\widehat{f}_{\mathcal{U}})-\inf_{f\in\mathcal{F}}\mathrm{L}_{\mathrm{un}}(f)\leqslant2\mathcal{M}\sqrt{\frac{3\ln 4|\mathcal{C}|/\delta}{\mathrm{N}\rho_{\min}}}+\sum_{c\in\mathcal{C}}\rho(c)\left[4\mathfrak{R}_{\mathrm{N}_c}(\ell\circ\mathcal{F})+16\mathcal{M}\sqrt{\frac{\ln 2|\mathcal{C}|/\delta}{2\widehat{\mathrm{N}}_c}}\right]. \tag{35}$$

*where $\widehat{\mathrm{N}}_c=\max\left\{1,\min\left(\left\lfloor\frac{\mathrm{N}\rho(c)(1-\Lambda)}{2}\right\rfloor,\left\lfloor\frac{\mathrm{N}(1-\rho(c))(1-\Lambda)}{k}\right\rfloor\right)\right\}$, $\Lambda=\sqrt{\frac{3\ln 4|\mathcal{C}|/\delta}{\mathrm{N}\rho_{\min}}}$ (cf. Table 2) and $\mathfrak{R}_{\mathrm{N}_c}(\ell\circ\mathcal{F})$ is the expected Rademacher complexity (Eqn. (24)).*

**Proof.** From the definition of $\mathcal{U}_{\mathrm{N}}$, we have:

$$\mathrm{L}_{\mathrm{un}}(\widehat{f}_{\mathcal{U}})-\inf_{f\in\mathcal{F}}\mathrm{L}_{\mathrm{un}}(f)\leqslant2\sup_{f\in\mathcal{F}}\left|\mathcal{U}_{\mathrm{N}}(f)-\mathrm{L}_{\mathrm{un}}(f)\right|\qquad\text{(Uniform Deviation Bound)}$$

$$=2\sup_{f\in\mathcal{F}}\left|\sum_{c\in\mathcal{C}}\frac{\mathrm{N}_c^+}{\mathrm{N}}\mathcal{U}_{\mathrm{N}}(f|c)-\sum_{c\in\mathcal{C}}\rho(c)\mathrm{L}_{\mathrm{un}}(f|c)\right|$$

$$=2\sup_{f\in\mathcal{F}}\left|\sum_{c\in\mathcal{C}}\mathcal{U}_{\mathrm{N}}(f|c)\left(\frac{\mathrm{N}_c^+}{\mathrm{N}}-\rho(c)\right)+\sum_{c\in\mathcal{C}}\rho(c)\Big(\mathcal{U}_{\mathrm{N}}(f|c)-\mathrm{L}_{\mathrm{un}}(f|c)\Big)\right|$$

$$\leqslant2\sum_{c\in\mathcal{C}}\sup_{f\in\mathcal{F}}\mathcal{U}_{\mathrm{N}}(f|c)\left|\frac{\mathrm{N}_c^+}{\mathrm{N}}-\rho(c)\right|+2\sum_{c\in\mathcal{C}}\rho(c)\sup_{f\in\mathcal{F}}\left|\mathcal{U}_{\mathrm{N}}(f|c)-\mathrm{L}_{\mathrm{un}}(f|c)\right|$$

$$\leqslant2\mathcal{M}\sum_{c\in\mathcal{C}}\left|\frac{\mathrm{N}_c^+}{\mathrm{N}}-\rho(c)\right|+2\sum_{c\in\mathcal{C}}\rho(c)\sup_{f\in\mathcal{F}}\left|\mathcal{U}_{\mathrm{N}}(f|c)-\mathrm{L}_{\mathrm{un}}(f|c)\right|.$$

Let $\delta\in(0,1)$ and $\Lambda=\sqrt{\frac{3\ln 4|\mathcal{C}|/\delta}{\mathrm{N}\rho_{\min}}}$. By lemma D.6, we have:

$$\sum_{c\in\mathcal{C}}\left|\frac{\mathrm{N}_c^+}{\mathrm{N}}-\rho(c)\right|\leqslant\sqrt{\frac{3\ln 4|\mathcal{C}|/\delta}{\mathrm{N}\rho_{\min}}}\triangleq\Lambda, \tag{36}$$

with probability of at least $1 - \delta/2$. Furthermore, $\left| \frac{N_c^+}{N} - \rho(c) \right| \leqslant \rho(c)\Lambda$ holds simultaneously for all $c \in \mathcal{C}$ with probability $1 - \delta/2$. Hence, by the triangle inequality:

$$\frac{N_c^+}{N} \geqslant \rho(c) - \Lambda\rho(c) \implies \begin{cases} N_c^+ & \geqslant N\rho(c)(1 - \Lambda) \\ N_c^- & \geqslant N(1 - \rho(c))(1 - \Lambda) \end{cases},$$

simultaneously for all $c \in \mathcal{C}$ with probability of at least $1 - \delta/2$. Therefore, we can estimate $N_c$ with probability of at least $1 - \delta/2$ as follows:

$$\overline{N}_c \geqslant \widehat{N}_c \triangleq \max \left\{ 1, \min \left( \left\lfloor \frac{N\rho(c)(1 - \Lambda)}{2} \right\rfloor, \left\lfloor \frac{N(1 - \rho(c))(1 - \Lambda)}{k} \right\rfloor \right) \right\}. \tag{37}$$

By proposition D.5, we have $\sup_{f \in \mathcal{F}} \left| \mathcal{U}_N(f|c) - L_{un}(f|c) \right| \leqslant 2\mathfrak{R}_{N_c}(\ell \circ \mathcal{F}) + 8\mathcal{M}\sqrt{\frac{\ln 2/\xi}{2\overline{N}_c}}$ with probability of at least $1 - \xi$ for all $\xi \in (0, 1)$. Then, by the Union bound, we have:

$$\sum_{c \in \mathcal{C}} \rho(c) \sup_{f \in \mathcal{F}} \left| \mathcal{U}_N(f|c) - L_{un}(f|c) \right| \leqslant \sum_{c \in \mathcal{C}} \rho(c) \left[ 2\mathfrak{R}_{N_c}(\ell \circ \mathcal{F}) + 8\mathcal{M}\sqrt{\frac{\ln 4|\mathcal{C}|/\delta}{2\overline{N}_c}} \right], \tag{38}$$

with probability of at least $1 - \delta/2$. By inequalities (36), (38) and a Union bound, we have:

$$
\begin{aligned}
L_{un}(\widehat{f}_{\mathcal{U}}) - \inf_{f \in \mathcal{F}} L_{un}(f) & \leqslant 2\Lambda\mathcal{M} + \sum_{c \in \mathcal{C}} \rho(c) \left[ 4\mathfrak{R}_{N_c}(\ell \circ \mathcal{F}) + 16\mathcal{M}\sqrt{\frac{\ln 4|\mathcal{C}|/\delta}{2\overline{N}_c}} \right] \\
& \leqslant 2\Lambda\mathcal{M} + \sum_{c \in \mathcal{C}} \rho(c) \left[ 4\mathfrak{R}_{N_c}(\ell \circ \mathcal{F}) + 16\mathcal{M}\sqrt{\frac{\ln 4|\mathcal{C}|/\delta}{2\widehat{N}_c}} \right], \qquad \text{(Eqn. (37))} \\
& = 2\mathcal{M}\sqrt{\frac{3 \ln 4|\mathcal{C}|/\delta}{N\rho_{\min}}} + \sum_{c \in \mathcal{C}} \rho(c) \left[ 4\mathfrak{R}_{N_c}(\ell \circ \mathcal{F}) + 16\mathcal{M}\sqrt{\frac{\ln 4|\mathcal{C}|/\delta}{2\widehat{N}_c}} \right].
\end{aligned}
$$

with probability of at least $1 - \delta$. Hence, we obtained the desired bound. $\qquad \square$

**Remark D.8.** The above proof relies on the assumption that $\Lambda < 1$ because of the use of multiplicative Chernoff bound. In other words, $N\rho_{\min} \geqslant 3 \ln 4|\mathcal{C}|/\delta$. However, when we have $\Lambda \geqslant 1$, the right hand side of the bound becomes greater than or equal to $2\mathcal{M}$, which is already larger than the largest possible value for $L_{un}(\widehat{f}_{\mathcal{U}}) - \inf_{f \in \mathcal{F}} L_{un}(f)$. Hence, we can safely remove the assumption that $\Lambda \in (0, 1)$ from the result of the theorem.

**Remark D.9.** Let $\Lambda = \sqrt{\frac{3 \ln 4|\mathcal{C}|/\delta}{N\rho_{\min}}}$. Suppose that $N \geqslant \frac{6 \ln 4|\mathcal{C}|/\delta}{\min \left( \frac{\rho_{\min}}{2}, \frac{1 - \rho_{\max}}{k} \right)}$. We can further simplify the high probability (of at least $1 - \delta/2$) lower bound for $N_c$. Firstly, we have:

$$\Lambda = \sqrt{\frac{3 \ln 4|\mathcal{C}|/\delta}{N\rho_{\min}}} \leqslant \sqrt{\frac{\min \left( \frac{\rho_{\min}}{2}, \frac{1 - \rho_{\max}}{k} \right)}{2\rho_{\min}}} \leqslant \frac{1}{2}.$$

Note that for every $c \in \mathcal{C}$, we have $\rho(c) \geqslant \rho_{\min}$ and $1 - \rho(c) \geqslant 1 - \rho_{\max}$ where $\rho_{\max} = \max_{c \in \mathcal{C}} \rho(c)$. Hence, with probability of at least $1 - \delta/2$, we have:

$$
\begin{aligned}
\overline{N}_c & \geqslant \max \left\{ 1, \min \left( \left\lfloor \frac{N\rho_{\min}(1 - \Lambda)}{2} \right\rfloor, \left\lfloor \frac{N(1 - \Lambda)(1 - \rho_{\max})}{k} \right\rfloor \right) \right\} \\
& = \max \left\{ 1, \left\lfloor N(1 - \Lambda) \min \left( \frac{\rho_{\min}}{2}, \frac{1 - \rho_{\max}}{k} \right) \right\rfloor \right\} \\
& \geqslant N \min \left( \frac{\rho_{\min}}{8}, \frac{1 - \rho_{\max}}{4k} \right),
\end{aligned}
\tag{39}
$$

since $1 - \Lambda \geqslant \frac{1}{2}$ and $\lfloor x \rfloor \geqslant \frac{x}{2}$ for all $x \geqslant 1$. Then, by the initial assumption on the lower bound of N, we have $\mathrm{N} \min \left( \frac{\rho_{\min}}{8}, \frac{1 - \rho_{\max}}{4k} \right) \geqslant \frac{3}{2} \ln 4 |\mathcal{C}| / \delta > 1$. Therefore, with probability of at least $1 - \delta/2$, our final simplification for the lower bound of $\overline{\mathrm{N}}_c$ becomes $\overline{\mathrm{N}}_c \geqslant \mathrm{N} \min \left( \frac{\rho_{\min}}{8}, \frac{1 - \rho_{\max}}{4k} \right)$ for all $c \in \mathcal{C}$.

**Theorem D.10.** *Let $\mathcal{F}$ be a class of representation functions and $\widehat{f}_\mathcal{U} = \arg\min_{f \in \mathcal{F}} \mathcal{U}_\mathrm{N}(f)$ be the empirical minimizer of U-Statistics (Eqn. (28)). Suppose that $\widehat{\mathfrak{R}}_{\mathcal{T}_c^{\mathrm{iid}}}(\ell \circ \mathcal{F}) \leqslant \overline{\mathrm{N}}_c^{-\frac{1}{2}} K_{\mathcal{F},c}$ where $K_{\mathcal{F},c}$ depends on both the function class $\mathcal{F}$ and $c$ for all $c \in \mathcal{C}$. Then, for any $\delta \in (0,1)$, with probability of at least $1 - \delta$, we have:*

$$\mathrm{L}_{\mathrm{un}}(\widehat{f}_\mathcal{U}) - \inf_{f \in \mathcal{F}} \mathrm{L}_{\mathrm{un}}(f) \leqslant \frac{8}{\sqrt{\widetilde{\mathrm{N}}}} \sum_{c \in \mathcal{C}} \rho(c) K_{\mathcal{F},c} + 44 \mathcal{M} \sqrt{\frac{\ln 8|\mathcal{C}|/\delta}{2\widetilde{\mathrm{N}}}}, \tag{40}$$

*where $\widetilde{\mathrm{N}} = \mathrm{N} \min \left( \frac{\rho_{\min}}{2}, \frac{1 - \rho_{\max}}{k} \right)$ and $\widehat{\mathfrak{R}}_{\mathcal{T}_c^{\mathrm{iid}}}(\ell \circ \mathcal{F})$ is the empirical Rademacher complexity of the loss class restricted to the set of independent tuples $\mathcal{T}_c^{\mathrm{iid}}$ (Eqn. (24)).*

**Proof.** Fixing $\Delta \in (0,1)$. Let $\Lambda = \sqrt{\frac{3 \ln 4|\mathcal{C}|/\Delta}{\mathrm{N} \rho_{\min}}}$ and suppose that $\mathrm{N} \geqslant \frac{6 \ln 4|\mathcal{C}|/\Delta}{\min \left( \frac{\rho_{\min}}{2}, \frac{1 - \rho_{\max}}{k} \right)}$. Then, $\Lambda \leqslant \frac{1}{2}$ and the events $\overline{\mathrm{N}}_c \geqslant \widetilde{\mathrm{N}}/4$ hold simultaneously for all $c \in \mathcal{C}$ with probability of at least $1 - \Delta/2$ (Eqn. (39)). By proposition D.7 and the assumption on the lower bound of N, for any value of $\Delta \in (0,1)$, we have:

$$\begin{aligned}
\mathrm{L}_{\mathrm{un}}(\widehat{f}_\mathcal{U}) - \inf_{f \in \mathcal{F}} \mathrm{L}_{\mathrm{un}}(f) &\leqslant 2\Lambda \mathcal{M} + \sum_{c \in \mathcal{C}} \rho(c) \left[ 4 \mathfrak{R}_{\mathrm{N}_c}(\ell \circ \mathcal{F}) + 16 \mathcal{M} \sqrt{\frac{\ln 4|\mathcal{C}|/\Delta}{2\overline{\mathrm{N}}_c}} \right] \\
&\leqslant 2\Lambda \mathcal{M} + 16 \mathcal{M} \sqrt{\frac{2 \ln 4|\mathcal{C}|/\Delta}{\widetilde{\mathrm{N}}}} + 4 \sum_{c \in \mathcal{C}} \rho(c) \mathfrak{R}_{\mathrm{N}_c}(\ell \circ \mathcal{F}) \\
&= 2\mathcal{M} \sqrt{\frac{3 \ln 4|\mathcal{C}|/\Delta}{\mathrm{N} \rho_{\min}}} + 16 \mathcal{M} \sqrt{\frac{2 \ln 4|\mathcal{C}|/\Delta}{\widetilde{\mathrm{N}}}} + 4 \sum_{c \in \mathcal{C}} \rho(c) \mathfrak{R}_{\mathrm{N}_c}(\ell \circ \mathcal{F}) \\
&\leqslant 2\mathcal{M} \sqrt{\frac{3 \ln 4|\mathcal{C}|/\Delta}{2\widetilde{\mathrm{N}}}} + 16 \mathcal{M} \sqrt{\frac{2 \ln 4|\mathcal{C}|/\Delta}{\widetilde{\mathrm{N}}}} + 4 \sum_{c \in \mathcal{C}} \rho(c) \mathfrak{R}_{\mathrm{N}_c}(\ell \circ \mathcal{F}) \\
&\leqslant 18 \mathcal{M} \sqrt{\frac{2 \ln 4|\mathcal{C}|/\Delta}{\widetilde{\mathrm{N}}}} + 4 \sum_{c \in \mathcal{C}} \rho(c) \mathfrak{R}_{\mathrm{N}_c}(\ell \circ \mathcal{F}) \\
&= 36 \mathcal{M} \sqrt{\frac{\ln 4|\mathcal{C}|/\Delta}{2\widetilde{\mathrm{N}}}} + 4 \sum_{c \in \mathcal{C}} \rho(c) \mathfrak{R}_{\mathrm{N}_c}(\ell \circ \mathcal{F}),
\end{aligned} \tag{41}$$

with probability of at least $1 - \Delta$. By lemma F.2 and a Union bound, we have:

$$\begin{aligned}
\sum_{c \in \mathcal{C}} \rho(c) \mathfrak{R}_{\mathrm{N}_c}(\ell \circ \mathcal{F}) &\leqslant \sum_{c \in \mathcal{C}} \rho(c) \left[ \widehat{\mathfrak{R}}_{\mathcal{T}_c^{\mathrm{iid}}}(\ell \circ \mathcal{F}) + \mathcal{M} \sqrt{\frac{\ln |\mathcal{C}|/\Delta}{2\overline{\mathrm{N}}_c}} \right] \\
&\leqslant \sum_{c \in \mathcal{C}} \rho(c) \left[ \widehat{\mathfrak{R}}_{\mathcal{T}_c^{\mathrm{iid}}}(\ell \circ \mathcal{F}) + 2\mathcal{M} \sqrt{\frac{\ln |\mathcal{C}|/\Delta}{2\widetilde{\mathrm{N}}}} \right] \\
&= 2\mathcal{M} \sqrt{\frac{\ln |\mathcal{C}|/\Delta}{2\widetilde{\mathrm{N}}}} + \sum_{c \in \mathcal{C}} \rho(c) \widehat{\mathfrak{R}}_{\mathcal{T}_c^{\mathrm{iid}}}(\ell \circ \mathcal{F}),
\end{aligned} \tag{42}$$

with probability of at least $1 - \Delta$. From Eqn. (41), Eqn. (42) and a Union bound, we have:

$$
\begin{aligned}
\mathrm{L}_{\mathrm{un}}(\widehat{f}_{\mathcal{U}}) - \inf_{f \in \mathcal{F}} \mathrm{L}_{\mathrm{un}}(f) &\leqslant 36\mathcal{M}\sqrt{\frac{\ln 4|\mathcal{C}|/\Delta}{2\widetilde{\mathrm{N}}}} + 8\mathcal{M}\sqrt{\frac{\ln |\mathcal{C}|/\Delta}{2\widetilde{\mathrm{N}}}} + 4\sum_{c \in \mathcal{C}} \rho(c)\widehat{\mathfrak{R}}_{\mathcal{T}_c^{\mathrm{iid}}}(\ell \circ \mathcal{F}) \\
&\leqslant 44\mathcal{M}\sqrt{\frac{\ln 4|\mathcal{C}|/\Delta}{2\widetilde{\mathrm{N}}}} + 4\sum_{c \in \mathcal{C}} \rho(c)\widehat{\mathfrak{R}}_{\mathcal{T}_c^{\mathrm{iid}}}(\ell \circ \mathcal{F}) \\
&\leqslant 44\mathcal{M}\sqrt{\frac{\ln 4|\mathcal{C}|/\Delta}{2\widetilde{\mathrm{N}}}} + 4\sum_{c \in \mathcal{C}} \rho(c)\frac{K_{\mathcal{F},c}}{\sqrt{\overline{\mathrm{N}}_c}} \\
&\leqslant 44\mathcal{M}\sqrt{\frac{\ln 4|\mathcal{C}|/\Delta}{2\widetilde{\mathrm{N}}}} + \frac{8}{\sqrt{\widetilde{\mathrm{N}}}} \sum_{c \in \mathcal{C}} \rho(c)K_{\mathcal{F},c},
\end{aligned}
$$

with probability of at least $1 - 2\Delta$. Now, suppose that $\mathrm{N} < \frac{6\ln 4|\mathcal{C}|/\Delta}{\min\left(\frac{\rho_{\min}}{2}, \frac{1-\rho_{\max}}{k}\right)}$ or $\widetilde{\mathrm{N}} < 6\ln 4|\mathcal{C}|/\Delta$. Then, we have

$44\mathcal{M}\sqrt{\frac{\ln 4|\mathcal{C}|/\Delta}{2\widetilde{\mathrm{N}}}} > 44\mathcal{M}\sqrt{\frac{1}{12}}$. Therefore, the right-hand-side of Eqn. (41) will be at least $22\mathcal{M}/\sqrt{3}$, which is the greater than the largest possible upper-bound for $\mathrm{L}_{\mathrm{un}}(\widehat{f}_{\mathcal{U}}) - \inf_{f \in \mathcal{F}} \mathrm{L}_{\mathrm{un}}(f)$. Hence, the inequality in Eqn. (41) holds regardless of the assumption on N. Finally, setting $\Delta = \delta/2$ yields the desired bound. $\qquad\square$

### D.3. Applications to Common Function Classes

In the results that follows, without reiteration, we define $\widehat{f}_{\mathcal{U}} = \arg\min_{f \in \mathcal{F}} \mathcal{U}_{\mathrm{N}}(f)$ by default. We will use covering number and the Dudley entropy integral bound (Theorem F.5) as the primary tools for bounding empirical Rademacher complexities. Firstly, we define the following metrics defined on function spaces:

**Definition D.11** ($\mathrm{L}_2$ and $\mathrm{L}_{\infty,2}$ metrics). Let $\mathcal{G} = \left\{g : \mathcal{Z} \to \mathbb{R}^d\right\}$ be a class of real/vector-valued functions (with $d \geqslant 1$) from an input space $\mathcal{Z}$. Let $S = \{\mathbf{z}_1, \ldots, \mathbf{z}_n\} \subset \mathcal{Z}$ be a dataset. Then, the $\mathrm{L}_2$ and $\mathrm{L}_{\infty,2}$ metrics defined for any two functions $g, \tilde{g} \in \mathcal{G}$ is defined as:

$$
\begin{aligned}
\|g - \tilde{g}\|_{\mathrm{L}_2(S)} &= \left(\frac{1}{n}\sum_{j=1}^{n} \|g(\mathbf{z}_j) - \tilde{g}(\mathbf{z}_j)\|_2^2\right)^{1/2}, \\
\|g - \tilde{g}\|_{\mathrm{L}_{\infty,2}(S)} &= \max_{\mathbf{z}_j \in S} \|g(\mathbf{z}_j) - \tilde{g}(\mathbf{z}_j)\|_2.
\end{aligned}
\tag{43}
$$

**Lemma D.12** (Hieu et al. (2024)). *Let $\mathcal{F}$ be a class of representation functions $f : \mathcal{X} \to \mathbb{R}^d$. Let $\ell : \mathbb{R}^k \to \mathbb{R}_+$ be a contrastive loss function which is $\ell^\infty$-Lipschitz with constant $\eta > 0$ and $S = \left\{(\mathbf{x}_j, \mathbf{x}_j^+, \mathbf{x}_{j1}^-, \ldots, \mathbf{x}_{jk}^-)\right\}_{j=1}^{n} \subset \mathcal{X}^{k+2}$ be a given set of i.i.d. input tuples. Then, let $\ell \circ \mathcal{F}$ denote the loss function class and $\epsilon > 0$, we have:*

$$
\ln \mathcal{N}\left(\ell \circ \mathcal{F}, \epsilon, \mathrm{L}_2(S)\right) \leqslant \ln \mathcal{N}\left(\mathcal{F}, \frac{\epsilon}{4\Gamma\eta}, \mathrm{L}_{\infty,2}(\widetilde{S})\right),
\tag{44}
$$

*where $\widetilde{S} \subset \mathcal{X}$ is the set of all vectors including anchor, positive and negative samples and $\Gamma = \sup_{f \in \mathcal{F}, \mathbf{x} \in \widetilde{S}} \|f(\mathbf{x})\|_2$.*

#### D.3.1. LINEAR FUNCTIONS

**Proposition D.13** (Ledent et al. 2021; Hieu et al. 2024 ). *Let $a, b \in \mathbb{R}_+$ be given and let $\mathcal{X} \subseteq \mathbb{R}^m$ be an input space where $m \geqslant 2$. Define $\mathcal{F}$ as the class of functions $f : \mathcal{X} \to \mathbb{R}^d$ as follows:*

$$
\mathcal{F} = \left\{x \mapsto Ax : A \in \mathbb{R}^{d \times m}, \|A^\top\|_{2,1} \leqslant a\right\},
$$

*where $d \geqslant 2$. Given dataset $S = \{x_1, \ldots, x_n\} \in \mathcal{X}^n$ such that $\|x_i\|_2 \leqslant b, \forall 1 \leqslant i \leqslant n$. Then, for all $\epsilon > 0$, we have:*

$$
\ln \mathcal{N}\left(\mathcal{F}, \epsilon, \mathrm{L}_{\infty,2}(S)\right) \leqslant \frac{64a^2b^2}{\epsilon^2} \ln\left(\left(\frac{11ab}{\epsilon} + 7\right)nd\right).
\tag{45}
$$

**Remark D.14.** For proper credit, the above proposition is a consequence of the $L_\infty$ covering number for linear function classes from Zhang (2002).

**Theorem D.15.** *Let $a, s \in \mathbb{R}_+$ be given and $\mathcal{F}_{\mathrm{lin}}$ be the class of linear representation functions (also defined in Eqn. (17)), defined as follows:*

$$\mathcal{F}_{\mathrm{lin}} = \left\{ x \mapsto Ax : A \in \mathbb{R}^{d \times m}, \|A^\top\|_{2,1} \leqslant a, \|A\|_\sigma \leqslant s \right\}. \tag{46}$$

*Let $\widetilde{N} = N \min\left( \frac{\rho_{\min}}{2}, \frac{1 - \rho_{\max}}{k} \right)$ and $\|\mathbf{x}\|_2 \leqslant b, \forall \mathbf{x} \in \mathcal{S}$ with probability one [3]. Suppose that the contrastive loss function $\ell : \mathbb{R}^k \to [0, \mathcal{M}]$ is $\ell^\infty$-Lipschitz with constant $\eta > 0$. Then, for any $\delta \in (0, 1)$, with probability of at least $1 - \delta$, we have:*

$$\mathrm{L}_{\mathrm{un}}(\hat{f}_{\mathcal{U}}) - \inf_{f \in \mathcal{F}_{\mathrm{lin}}} \mathrm{L}_{\mathrm{un}}(f) \leqslant \frac{32}{N\sqrt{\widetilde{N}}} + 44\mathcal{M}\sqrt{\frac{\ln 8|\mathcal{C}|/\delta}{2\widetilde{N}}} + \frac{3072\sqrt{2}\eta s a b^2}{\sqrt{\widetilde{N}}} \ln\left( \left(44N\eta s a b^2 + 7\right) N(k+2)d \right) \ln(N\mathcal{M}). \tag{47}$$

**Proof.** Given a certain class $c \in \mathcal{C}$. Suppose that $N_c \geqslant 1$. In other words, the set of *i.i.d.* tuples whose anchor-positive pairs belong to class $c$, $\mathcal{T}_c^{\mathrm{iid}}$, is not empty. Let $\widetilde{\mathcal{T}}_c^{\mathrm{iid}}$ be the set of all vectors including anchor, positive and negative samples from $\mathcal{T}_c^{\mathrm{iid}}$. Let $\Gamma_c = \sup_{f \in \mathcal{F}_{\mathrm{lin}}, \mathbf{x} \in \widetilde{\mathcal{T}}_c^{\mathrm{iid}}} \|f(\mathbf{x})\|_2$. For any $\epsilon > 0$, we have:

$$
\begin{aligned}
\ln \mathcal{N}\left( \ell \circ \mathcal{F}_{\mathrm{lin}}, \epsilon, \mathrm{L}_2(\mathcal{T}_c^{\mathrm{iid}}) \right) &\leqslant \ln \mathcal{N}\left( \mathcal{F}_{\mathrm{lin}}, \frac{\epsilon}{4\Gamma_c \eta}, \mathrm{L}_{\infty, 2}(\widetilde{\mathcal{T}}_c^{\mathrm{iid}}) \right) && \text{(Lemma D.12)} \\
&\leqslant \frac{1024\eta^2 a^2 b^2 \Gamma_c^2}{\epsilon^2} \ln\left( \left( \frac{44\eta\Gamma_c ab}{\epsilon} + 7 \right) N_c(k+2)d \right) && \text{(Proposition D.13)} \\
&\leqslant \frac{1024\eta^2 s^2 a^2 b^4}{\epsilon^2} \ln\left( \left( \frac{44\eta s a b^2}{\epsilon} + 7 \right) N(k+2)d \right) && (\Gamma_c \leqslant sb, \forall c \in \mathcal{C}).
\end{aligned}
$$

Let $\alpha = N^{-1}$. Without loss of generality, assume that $\mathcal{N}\left( \ell \circ \mathcal{F}_{\mathrm{lin}}, \epsilon, \mathrm{L}_2(\mathcal{T}_c^{\mathrm{iid}}) \right) \geqslant 2$ for all $\alpha \leqslant \epsilon \leqslant \mathcal{M}$. Then, using Theorem F.5, we have:

$$
\begin{aligned}
\widehat{\mathfrak{R}}_{\mathcal{T}_c^{\mathrm{iid}}}(\ell \circ \mathcal{F}_{\mathrm{lin}}) &\leqslant 4\alpha + \frac{12}{\sqrt{N_c}} \int_\alpha^{\mathcal{M}} \sqrt{2 \ln \mathcal{N}\left( \ell \circ \mathcal{F}_{\mathrm{lin}}, \epsilon, \mathrm{L}_2(\mathcal{T}_c^{\mathrm{iid}}) \right)} \, d\epsilon \\
&\leqslant 4\alpha + \frac{12}{\sqrt{N_c}} \int_\alpha^{\mathcal{M}} \sqrt{\frac{2048\eta^2 s^2 a^2 b^4}{\epsilon^2} \ln\left( \left( \frac{44\eta s a b^2}{\epsilon} + 7 \right) N(k+2)d \right)} \, d\epsilon \\
&\leqslant 4\alpha + \frac{384\sqrt{2}\eta s a b^2}{\sqrt{N_c}} \ln\left( \left( \frac{44\eta s a b^2}{\alpha} + 7 \right) N(k+2)d \right) \int_\alpha^{\mathcal{M}} \frac{1}{\epsilon} \, d\epsilon \\
&= 4\alpha + \frac{384\sqrt{2}\eta s a b^2}{\sqrt{N_c}} \ln\left( \left( \frac{44\eta s a b^2}{\alpha} + 7 \right) N(k+2)d \right) \ln(\mathcal{M}/\alpha) \\
&= \frac{4}{N} + \frac{384\sqrt{2}\eta s a b^2}{\sqrt{N_c}} \ln\left( \left( 44N\eta s a b^2 + 7 \right) N(k+2)d \right) \ln(N\mathcal{M}).
\end{aligned}
$$

Setting $\ln\left( \left( 44N\eta s a b^2 + 7 \right) N(k+2)d \right) \ln(N\mathcal{M}) = \phi$ for brevity. Then, for all $c \in \mathcal{C}$, we have:

$$\widehat{\mathfrak{R}}_{\mathcal{T}_c^{\mathrm{iid}}}(\ell \circ \mathcal{F}_{\mathrm{lin}}) \leqslant \frac{4}{N} + \frac{384\sqrt{2}\eta s a b^2 \phi}{\sqrt{N_c}}.$$

---

[3]The bound on input's norm with probability one is imposed so that for any draw of $\mathcal{S}$, we can bound the $L_{\infty,2}$ covering number of $\mathcal{F}_{\mathrm{lin}}$ restricted to any subset of $\mathcal{S}$ using proposition D.13 with the same value $b$.

Then, by Theorem D.10, setting $K_{\mathcal{F},c} = \frac{4}{N} + 384\sqrt{2}\eta s a b^2 \phi$ for all $c \in \mathcal{C}$, we have:

$$\mathrm{L}_{\mathrm{un}}(\widehat{f}_{\mathcal{U}}) - \inf_{f \in \mathcal{F}_{\mathrm{lin}}} \mathrm{L}_{\mathrm{un}}(f) \leqslant \frac{32}{N\sqrt{\widetilde{N}}} + 44\mathcal{M}\sqrt{\frac{\ln 8|\mathcal{C}|/\delta}{2\widetilde{N}}} + \frac{3072\sqrt{2}\eta s a b^2}{\sqrt{\widetilde{N}}} \ln\left(\left(44N\eta s a b^2 + 7\right) N(k+2)d\right) \ln(N\mathcal{M}),$$

with probability of at least $1 - \delta$ for any $\delta \in (0, 1)$, as desired. $\qquad\square$

### D.3.2. NEURAL NETWORKS

We are interested in the following class of neural networks: Let $L \geqslant 1$ be a natural number representing number of layers. Let $d_0, d_1, \ldots, d_L \in \mathbb{N}$ such that $d_l \geqslant 1, \forall 0 \leqslant l \leqslant L$ be layer widths. Let $s_1, \ldots, s_L$ be a sequence of real positive numbers and define the following parameter spaces:

$$\mathcal{B}_l = \left\{ A^{(l)} \in \mathbb{R}^{d_{l-1} \times d_l} : \|A^{(l)}\|_\sigma \leqslant s_l \right\}, \tag{48}$$

where $\|\cdot\|_\sigma$ denotes the spectral norm. Denote $\mathcal{A} = \mathcal{B}_1 \times \cdots \times \mathcal{B}_L$ as the parameter space for the class of neural networks $\mathcal{F}_{\mathrm{nn}}^{\mathcal{A}}$ (also defined in Eqn. (18)), defined as follows:

$$\mathcal{F}_{\mathrm{nn}}^{\mathcal{A}} = \mathcal{F}_L \circ \mathcal{F}_{L-1} \circ \cdots \circ \mathcal{F}_1, \tag{49}$$

where for all $1 \leqslant l \leqslant L$, $\mathcal{F}_l = \varphi_l \circ \mathcal{V}_l$ such that:

1. $\varphi_l : \mathbb{R}^{d_l} \to \mathbb{R}^{d_l}$ is an $\ell^2$-Lipschitz activation function with constant $\xi_l$ fixed a priori.

2. $\mathcal{V}_l = \left\{ x \mapsto A^{(l)}x : A^{(l)} \in \mathcal{B}_l \right\}$ represents the pre-activated linear layers.

**Lemma D.16** (Long & Sedghi (2020), Lemma A.8). *The internal covering number of a $d$-dimensional ball with radius $\kappa$, denoted as $\mathcal{B}_\kappa$, with respect to any norm $\|\cdot\|$ can be bounded by:*

$$\mathcal{N}\left(\mathcal{B}_\kappa, \epsilon, \|\cdot\|\right) \leqslant \left\lceil \frac{3\kappa}{\epsilon} \right\rceil^d \leqslant \left(\frac{3\kappa}{\epsilon} + 1\right)^d. \tag{50}$$

**Theorem D.17.** *Let $\mathcal{F}_{\mathrm{nn}}^{\mathcal{A}}$ be the class of neural networks defined in Eqn. (18). Let $\widetilde{N} = N \min\left(\frac{\rho_{\min}}{2}, \frac{1-\rho_{\max}}{k}\right)$ and $\|\mathbf{x}\|_2 \leqslant b, \forall \mathbf{x} \in \mathcal{S}$ with probability one. Suppose that the contrastive loss $\ell : \mathbb{R}^k \to \mathbb{R}_+$ is $\ell^\infty$-Lipschitz with constant $\eta > 0$. Then, for any $\delta \in (0, 1)$, with probability of at least $1 - \delta$, we have:*

$$\mathrm{L}_{\mathrm{un}}(\widehat{f}_{\mathcal{U}}) - \inf_{f \in \mathcal{F}_{\mathrm{nn}}^{\mathcal{A}}} \mathrm{L}_{\mathrm{un}}(f) \leqslant \frac{32}{N\sqrt{\widetilde{N}}} + 44\mathcal{M}\sqrt{\frac{\ln 8|\mathcal{C}|/\delta}{2\widetilde{N}}} + 192\mathcal{M}\sqrt{\frac{\mathcal{W}}{\widetilde{N}} \ln\left(12\eta N L b^2 \prod_{m=1}^{L} \xi_m^2 s_m^2 + 1\right)}, \tag{51}$$

*where $\mathcal{W} = \sum_{l=1}^{L} d_l$ is the total number of neurons in the neural networks.*

**Proof.** Let $\mathbf{A} = \{A^{(l)}\}_{l=1}^{L}$ be a set of weight matrices from the parameter space $\mathcal{A}$ and we denote $F_{\mathbf{A}} \in \mathcal{F}_{\mathrm{nn}}^{\mathcal{A}}$ as the neural network parameterized by $\mathbf{A}$. In other words,

$$F_{\mathbf{A}}(\mathbf{x}) = \varphi_L\left(A^{(L)}\varphi_{L-1}\left(\ldots\varphi_1(A^{(1)}\mathbf{x})\ldots\right)\right), \quad \forall \mathbf{x} \in \mathcal{X}.$$

Additionally, let $1 \leqslant l_1 \leqslant l_2 \leqslant L$, define $F_{\mathbf{A}}^{l_1 \to l_2}$ as the sub-network from layer $l_1$ to $l_2$. Specifically,

$$F_{\mathbf{A}}^{l_1 \to l_2}(\mathbf{z}) = \varphi_{l_2}\left(A^{(l_2)}\varphi_{l_2-1}\left(\ldots\varphi_{l_1}(A^{(l_1)}\mathbf{z})\ldots\right)\right), \quad \forall \mathbf{z} \in \mathbb{R}^{d_{l_1}-1}.$$

Let $\widetilde{\mathbf{A}} = \{\widetilde{A}^{(l)}\}_{l=1}^{L} \in \mathcal{A}$ be another set of weight matrices different from $\mathbf{A}$. Additionally, for $1 \leqslant l \leqslant L$, define $\widetilde{\mathbf{A}}_l = \{A^{(1)}, \ldots, A^{(l-1)}, \widetilde{A}^{(l)}, \ldots, \widetilde{A}^{(L)}\}$ as the set of weight matrices obtained by replacing the $l^{th}$ to the $L^{th}$ elements of $\mathbf{A}$ with those of $\widetilde{\mathbf{A}}$. Then, for all $1 \leqslant l \leqslant L$, we have:

$$
\begin{cases}
F_{\widetilde{\mathbf{A}}_l}^{l+1 \to L}(\mathbf{z}) & = F_{\widetilde{\mathbf{A}}_{l+1}}^{l+1 \to L}(\mathbf{z}), \quad \forall \mathbf{z} \in \mathbb{R}^{d_l} \\
F_{\widetilde{\mathbf{A}}_l}^{1 \to l-1}(\mathbf{x}) & = F_{\widetilde{\mathbf{A}}_{l+1}}^{1 \to l-1}(\mathbf{x}), \quad \forall \mathbf{x} \in \mathcal{X}
\end{cases} .
$$

Given a certain class $c \in \mathcal{C}$ and suppose that $\mathrm{N}_c \geqslant 1$. In other words, $\widetilde{\mathcal{T}}_c^{\mathrm{iid}} \neq \varnothing$. For any $\mathbf{x} \in \widetilde{\mathcal{T}}_c^{\mathrm{iid}}$ and $1 \leqslant l \leqslant L$, we have:

$$
\begin{aligned}
\left\| F_{\widetilde{\mathbf{A}}_l}(\mathbf{x}) - F_{\widetilde{\mathbf{A}}_{l+1}}(\mathbf{x}) \right\|_2 &= \left\| F_{\widetilde{\mathbf{A}}_l}^{l+1 \to L} \circ \varphi_l \left( A^{(l)} F_{\widetilde{\mathbf{A}}_l}^{1 \to l-1}(\mathbf{x}) \right) - F_{\widetilde{\mathbf{A}}_{l+1}}^{l+1 \to L} \circ \varphi_l \left( \widetilde{A}^{(l)} F_{\widetilde{\mathbf{A}}_{l+1}}^{1 \to l-1}(\mathbf{x}) \right) \right\|_2 \\
&\leqslant \xi_l \prod_{m=l+1}^{L} \xi_m s_m \left\| A^{(l)} F_{\widetilde{\mathbf{A}}_l}^{1 \to l-1}(\mathbf{x}) - \widetilde{A}^{(l)} F_{\widetilde{\mathbf{A}}_{l+1}}^{1 \to l-1}(\mathbf{x}) \right\|_2 \\
&\leqslant \xi_l \prod_{m=l+1}^{L} \xi_m s_m \left\| A^{(l)} - \widetilde{A}^{(l)} \right\|_\sigma \cdot \left\| F_{\widetilde{\mathbf{A}}_l}^{1 \to l-1}(\mathbf{x}) - F_{\widetilde{\mathbf{A}}_{l+1}}^{1 \to l-1}(\mathbf{x}) \right\| \\
&\leqslant \xi_l \prod_{m=l+1}^{L} \xi_m s_m \left\| A^{(l)} - \widetilde{A}^{(l)} \right\|_\sigma \cdot \|\mathbf{x}\|_2 \prod_{u=1}^{l-1} \xi_u s_u \\
&\leqslant \frac{b}{s_l} \prod_{m=1}^{L} \xi_m s_m \left\| A^{(l)} - \widetilde{A}^{(l)} \right\|_\sigma .
\end{aligned}
$$

By the triangle inequality, for all $\mathbf{x} \in \widetilde{\mathcal{T}}_c^{\mathrm{iid}}$, we have:

$$
\begin{aligned}
\left\| F_{\mathbf{A}}(\mathbf{x}) - F_{\widetilde{\mathbf{A}}}(\mathbf{x}) \right\|_2 &\leqslant \sum_{l=1}^{L} \left\| F_{\widetilde{\mathbf{A}}_l}(\mathbf{x}) - F_{\widetilde{\mathbf{A}}_{l+1}}(\mathbf{x}) \right\|_2 \\
&\leqslant b \prod_{m=1}^{L} \xi_m s_m \sum_{l=1}^{L} \frac{\|A^{(l)} - \widetilde{A}^{(l)}\|_\sigma}{s_l} .
\end{aligned}
$$

Let $\epsilon > 0$ and $\varepsilon_1, \ldots, \varepsilon_L$ be a sequence of positive real numbers such that $\epsilon = \sum_{l=1}^{L} \beta_l \varepsilon_l$ where $\sum_{l=1}^{L} \beta_l = 1$. For all $1 \leqslant l \leqslant L$, we construct an internal $\varepsilon_l$-cover, denoted as $\mathcal{C}(\mathcal{B}_l, \varepsilon_l)$, with respect to the $\|\cdot\|_\sigma$ norm for the parameter space at the $l^{th}$ layer, $\mathcal{B}_l$. Let $\mathcal{C}_{\mathcal{A}} = \bigotimes_{l=1}^{L} \mathcal{C}(\mathcal{B}_l, \varepsilon_l)$ be the Cartesian product of all constructed covers. Then, for all $\mathbf{A} = \{A^{(l)}\}_{l=1}^{L} \in \mathcal{A}$, there exists $\widetilde{\mathbf{A}} = \{\widetilde{A}^{(l)}\}_{l=1}^{L} \in \mathcal{C}_{\mathcal{A}}$ such that:

$$
\begin{aligned}
\left\| F_{\mathbf{A}} - F_{\widetilde{\mathbf{A}}} \right\|_{\mathrm{L}_{\infty,2}(\widetilde{\mathcal{T}}_c^{\mathrm{iid}})} &= \max_{\mathbf{x} \in \widetilde{\mathcal{T}}_c^{\mathrm{iid}}} \left\| F_{\mathbf{A}}(\mathbf{x}) - F_{\widetilde{\mathbf{A}}}(\mathbf{x}) \right\|_2 \\
&\leqslant b \prod_{m=1}^{L} \xi_m s_m \sum_{l=1}^{L} \frac{\|A^{(l)} - \widetilde{A}^{(l)}\|_\sigma}{s_l} \\
&\leqslant b \prod_{m=1}^{L} \xi_m s_m \sum_{l=1}^{L} \frac{\varepsilon_l}{s_l} .
\end{aligned}
$$

For each $1 \leqslant l \leqslant L$, setting $\varepsilon_l = \frac{\beta_l s_l \epsilon}{b \prod_{m=1}^{L} \xi_m s_m}$ yields $\|F_{\mathbf{A}} - F_{\widetilde{\mathbf{A}}}\|_{\mathrm{L}_{\infty,2}(\widetilde{\mathcal{T}}_c^{\mathrm{iid}})} \leqslant \epsilon$. Hence, $\mathcal{C}_{\mathcal{A}}$ becomes the $\epsilon$-cover for $\mathcal{A}$ (or

equivalently, $\mathcal{F}_{\mathrm{nn}}^{\mathcal{A}}$) with respect to the $\mathrm{L}_{\infty,2}(\widetilde{\mathcal{T}}_c^{\mathrm{iid}})$ metric. Setting $\beta_l = L^{-1}$ for all $l \in [L]$ [4], we have:

$$
\begin{aligned}
\ln \mathcal{N}\left(\mathcal{F}_{\mathrm{nn}}^{\mathcal{A}}, \epsilon, \mathrm{L}_{\infty,2}(\widetilde{\mathcal{T}}_c^{\mathrm{iid}})\right) &= \sum_{l=1}^{L} \ln |\mathcal{C}(\mathcal{B}_l, \varepsilon_l)| \\
&\leqslant \sum_{l=1}^{L} d_l \ln \left( \frac{3 s_l}{\varepsilon_l} + 1 \right) \qquad \text{(Lemma D.16)} \\
&= \sum_{l=1}^{L} d_l \ln \left( \frac{3b \prod_{m=1}^{L} \xi_m s_m}{\beta_l \epsilon} + 1 \right) \\
&= \mathcal{W} \ln \left( \frac{3Lb \prod_{m=1}^{L} \xi_m s_m}{\epsilon} + 1 \right).
\end{aligned}
$$

Let $\alpha = \mathrm{N}^{-1}$ and $\Gamma_c = \sup_{\mathbf{A} \in \mathcal{A}, \mathbf{x} \in \widetilde{\mathcal{T}}_c^{\mathrm{iid}}} \|F_{\mathbf{A}}(\mathbf{x})\|_2$. Without loss of generality, assume that $\mathcal{N}\left(\ell \circ \mathcal{F}_{\mathrm{nn}}^{\mathcal{A}}, \epsilon, \mathrm{L}_2(\mathcal{T}_c^{\mathrm{iid}})\right) \geqslant 2$ for all $\alpha \leqslant \epsilon \leqslant \mathcal{M}$. By Theorem F.5, we have:

$$
\begin{aligned}
\widehat{\mathfrak{R}}_{\mathcal{T}_c^{\mathrm{iid}}}(\ell \circ \mathcal{F}_{\mathrm{nn}}^{\mathcal{A}}) &\leqslant 4\alpha + \frac{12}{\sqrt{\mathrm{N}_c}} \int_{\alpha}^{\mathcal{M}} \sqrt{2 \ln \mathcal{N}\left(\ell \circ \mathcal{F}_{\mathrm{nn}}^{\mathcal{A}}, \epsilon, \mathrm{L}_2(\mathcal{T}_c^{\mathrm{iid}})\right)} d\epsilon \\
&\leqslant 4\alpha + \frac{12}{\sqrt{\mathrm{N}_c}} \int_{\alpha}^{\mathcal{M}} \sqrt{2 \ln \mathcal{N}\left(\mathcal{F}_{\mathrm{nn}}^{\mathcal{A}}, \frac{\epsilon}{4\eta \Gamma_c}, \mathrm{L}_{\infty,2}(\widetilde{\mathcal{T}}_c^{\mathrm{iid}})\right)} d\epsilon \qquad \text{(Lemma D.12)} \\
&\leqslant 4\alpha + \frac{12}{\sqrt{\mathrm{N}_c}} \int_{\alpha}^{\mathcal{M}} \sqrt{2\mathcal{W} \ln \left( \frac{12\eta L \Gamma_c b \prod_{m=1}^{L} \xi_m s_m}{\epsilon} + 1 \right)} d\epsilon \\
&\leqslant 4\alpha + \frac{12}{\sqrt{\mathrm{N}_c}} \int_{\alpha}^{\mathcal{M}} \sqrt{2\mathcal{W} \ln \left( \frac{12\eta L b^2 \prod_{m=1}^{L} \xi_m^2 s_m^2}{\epsilon} + 1 \right)} d\epsilon \qquad (\Gamma_c \leqslant b \prod_{m=1}^{L} \xi_m s_m, \forall c \in \mathcal{C}) \\
&\leqslant 4\alpha + \frac{12}{\sqrt{\mathrm{N}_c}}(\mathcal{M} - \alpha)\sqrt{2\mathcal{W} \ln \left( \frac{12\eta L b^2 \prod_{m=1}^{L} \xi_m^2 s_m^2}{\alpha} + 1 \right)} \\
&\leqslant \frac{4}{\mathrm{N}} + 12\mathcal{M}\sqrt{\frac{2\mathcal{W}}{\mathrm{N}_c} \ln \left( 12\eta \mathrm{N} L b^2 \prod_{m=1}^{L} \xi_m^2 s_m^2 + 1 \right)}.
\end{aligned}
$$

Then, for all $c \in \mathcal{C}$, we have:

$$
\widehat{\mathfrak{R}}_{\mathcal{T}_c^{\mathrm{iid}}}(\ell \circ \mathcal{F}_{\mathrm{nn}}^{\mathcal{A}}) \leqslant \frac{4}{\mathrm{N}} + 24\mathcal{M}\sqrt{\frac{\mathcal{W}}{2\mathrm{N}_c} \ln \left( 12\eta \mathrm{N} L b^2 \prod_{m=1}^{L} \xi_m^2 s_m^2 + 1 \right)}.
$$

Let $K_{\mathcal{F},c} = \frac{4}{\mathrm{N}} + 24\mathcal{M}\sqrt{\mathcal{W} \ln \left( 12\eta \mathrm{N} L b^2 \prod_{m=1}^{L} \xi_m^2 s_m^2 + 1 \right)}$ for all $c \in \mathcal{C}$, applying Theorem D.10 yields:

$$
\mathrm{L}_{\mathrm{un}}(\widehat{f}_{\mathcal{U}}) - \inf_{f \in \mathcal{F}_{\mathrm{nn}}^{\mathcal{A}}} \mathrm{L}_{\mathrm{un}}(f) \leqslant \frac{32}{\mathrm{N}\sqrt{\widetilde{\mathrm{N}}}} + 44\mathcal{M}\sqrt{\frac{\ln 8|\mathcal{C}|/\delta}{2\widetilde{\mathrm{N}}}} + 192\mathcal{M}\sqrt{\frac{\mathcal{W}}{\widetilde{\mathrm{N}}} \ln \left( 12\eta \mathrm{N} L b^2 \prod_{m=1}^{L} \xi_m^2 s_m^2 + 1 \right)},
$$

with probability of at least $1 - \delta$ for any $\delta \in (0, 1)$, as desired. $\qquad \square$

---

[4] Even though we can use Lagrange multiplier to solve for the optimal values of $\beta_l$, these values are only present in logarithmic terms. Therefore, we can afford to be somewhat less "strict" with their selection.

# E. Generalization Bound for Empirical Sub-sampled Risk Minimizer

## E.1. Sub-sampling Procedure

Given a labeled dataset $\mathcal{S} = \left\{ (\mathbf{x}_j, \mathbf{y}_j) \right\}_{j=1}^{N}$. Let $\mathcal{T}$ be the set of all valid tuples that can be formed from the dataset $\mathcal{S}$ and let $\mathcal{T}_c$ be the set of valid tuples whose anchor-positive pairs belong to class $c$. Then, we have $\mathcal{T} = \bigcup_{c \in \mathcal{C}} \mathcal{T}_c$ and $|\mathcal{T}| = \sum_{c \in \mathcal{C}} |\mathcal{T}_c|$ where $|\mathcal{T}_c| = 2\binom{N_c^+}{2}\binom{N - N_c^+}{k}$. The procedure for sampling subsets of tuples described in Section 3.3 is equivalent to drawing $M$ samples $i.i.d.$ from the discrete distribution $\nu$ over $\mathcal{T}$ defined as:

$$
\forall T \in \mathcal{T}, c \in \mathcal{C} : \nu(T) = \frac{N_c^+/N}{|\mathcal{T}_c|}, \quad \text{if } T \in \mathcal{T}_c
$$
$$
\mathcal{T}_{\text{sub}} \sim \nu^M.
$$
(52)

In other words, for all $c \in \mathcal{C}$, the probability of picking from $\mathcal{T}$ a tuple $T$ whose anchor-positive pair belongs to class $c$ is $\nu(\{T \in \mathcal{T}_c\}) = \frac{N_c^+}{N}$, which is an unbiased estimator of the class probabilities themselves, i.e., $\mathbb{E}_{\mathbf{N}}[\nu(\{T \in \mathcal{T}_c\})] = \rho(c)$.

## E.2. Proof of the Main Result

Let $\mathcal{T}_{\text{sub}} = \left\{ (\mathbf{x}_j, \mathbf{x}_j^+, \mathbf{x}_{j1}^-, \ldots, \mathbf{x}_{jk}^-) \right\}_{j=1}^{M} \sim \nu^M$ be the sample of tuples drawn $i.i.d.$ from the distribution $\nu$ over all valid tuples $\mathcal{T}$. Let $\mathcal{F}$ be a class of representation functions and $\ell : \mathbb{R}^k \to \mathbb{R}_+$ be an $\ell^\infty$-Lipschitz contrastive loss function. For any $f \in \mathcal{F}$, we define the sub-sampled empirical risk evaluated on the tuples set $\mathcal{T}_{\text{sub}}$ as follows:

$$
\widehat{\mathcal{L}}(f; \mathcal{T}_{\text{sub}}) = \frac{1}{M} \sum_{j=1}^{M} \ell\left( \left\{ f(\mathbf{x}_j)^\top \left[ f(\mathbf{x}_j^+) - f(\mathbf{x}_{ji}^-) \right] \right\}_{i=1}^{k} \right).
$$
(53)

**Theorem E.1.** *Let $\mathcal{F}$ be a class of representation functions and $\widehat{f}_{\text{sub}} = \arg\min_{f \in \mathcal{F}} \widehat{\mathcal{L}}(f; \mathcal{T}_{\text{sub}})$ be the empirical sub-sampled risk minimizer (Eqn. (53)). Suppose that $\widehat{\mathfrak{R}}_{\mathcal{T}_c^{\text{iid}}}(\ell \circ \mathcal{F}) \leqslant \overline{N}_c^{-\frac{1}{2}} K_{\mathcal{F},c}$ where $K_{\mathcal{F},c}$ depends on both the function class $\mathcal{F}$ and $c$ for all $c \in \mathcal{C}$. Then, for any $\delta \in (0, 1)$, with probability of at least $1 - \delta$, we have:*

$$
L_{\text{un}}(\widehat{f}_{\text{sub}}) - \inf_{f \in \mathcal{F}} L_{\text{un}}(f) \leqslant 4\widehat{\mathfrak{R}}_{\mathcal{T}_{\text{sub}}}(\ell \circ \mathcal{F}) + \frac{8}{\sqrt{\widetilde{N}}} \sum_{c \in \mathcal{C}} \rho(c) K_{\mathcal{F},c} + 6\mathcal{M}\sqrt{\frac{\ln 8/\delta}{2M}} + 44\mathcal{M}\sqrt{\frac{\ln 16|\mathcal{C}|/\delta}{2\widetilde{N}}},
$$
(54)

*where $\widetilde{N} = N \min\left( \frac{\rho_{\min}}{2}, \frac{1 - \rho_{\max}}{k} \right)$ and $\widehat{\mathfrak{R}}_{\mathcal{T}_c^{\text{iid}}}(\ell \circ \mathcal{F})$ is the empirical Rademacher complexity of the loss class restricted to the set of independent tuples $\mathcal{T}_c^{\text{iid}}$ (Eqn. (24)).*

**Proof.** Let $\Delta \in (0, 1)$. By the uniform deviation bound, we have:

$$
L_{\text{un}}(\widehat{f}_{\text{sub}}) - \inf_{f \in \mathcal{F}} L_{\text{un}}(f) \leqslant 2 \sup_{f \in \mathcal{F}} \left| \widehat{\mathcal{L}}_{\text{un}}(f; \mathcal{T}_{\text{sub}}) - L_{\text{un}}(f) \right|
$$
$$
\leqslant 2 \sup_{f \in \mathcal{F}} \left| \widehat{\mathcal{L}}_{\text{un}}(f; \mathcal{T}_{\text{sub}}) - \mathcal{U}_{\mathbf{N}}(f) \right| + 2 \sup_{f \in \mathcal{F}} \left| \mathcal{U}_{\mathbf{N}}(f) - L_{\text{un}}(f) \right|.
$$

With a slight abuse of notation, for any $T = (\mathbf{x}, \mathbf{x}^+, \mathbf{x}_1^-, \ldots, \mathbf{x}_k^-) \in \mathcal{T}$, denote $\ell\left( \left\{ f(\mathbf{x})^\top \left[ f(\mathbf{x}^+) - f(\mathbf{x}_i^-) \right] \right\}_{i=1}^{k} \right)$ as $\ell(T)$

for brevity. Denote $\widetilde{\mathcal{C}} = \left\{ c \in \mathcal{C} : \mathrm{N}_c \geqslant 1 \right\}$. Then, we have:

$$\mathbb{E}_{\mathcal{T}_{\mathrm{sub}} \sim \nu^{\mathrm{M}}} \left[ \widehat{\mathcal{L}}_{\mathrm{un}}(f; \mathcal{T}_{\mathrm{sub}}) \Big| \mathcal{S} \right] = \mathbb{E}_{\{T_j\}_{j=1}^{\mathrm{M}} \sim \nu^{\mathrm{M}}} \left[ \widehat{\mathcal{L}}_{\mathrm{un}}(f; \{T_j\}_{j=1}^{\mathrm{M}}) \Big| \mathcal{S} \right]$$

$$= \underbrace{\mathbb{E}_{\{T_j\}_{j=1}^{\mathrm{M}} \sim \nu^{\mathrm{M}}} \left[ \frac{1}{\mathrm{M}} \sum_{j=1}^{\mathrm{M}} \ell(T_j) \Big| \mathcal{S} \right]}_{\text{simplifies to } \mathbb{E}_{T \sim \nu}[\ell(T)|\mathcal{S}]}$$

$$= \sum_{T \in \mathcal{T}} \nu(T)\ell(T)$$

$$= \sum_{c \in \widetilde{\mathcal{C}}} \sum_{T_c \in \mathcal{T}_c} \nu(T_c)\ell(T_c)$$

$$= \sum_{c \in \widetilde{\mathcal{C}}} \frac{\mathrm{N}_c^+}{\mathrm{N}} \underbrace{\frac{1}{|\mathcal{T}_c|} \sum_{T_c \in \mathcal{T}_c} \ell(T_c)}_{\mathcal{U}_{\mathrm{N}}(f|c) \text{ - Eqn. (27)}}$$

$$= \sum_{c \in \mathcal{C}} \frac{\mathrm{N}_c^+}{\mathrm{N}} \mathcal{U}_{\mathrm{N}}(f|c) = \mathcal{U}_{\mathrm{N}}(f).$$

Hence, by Proposition F.3, for any choice of $\mathcal{S}$ we have with probability of at least $1 - \Delta$ (with respect to the draw of $\mathcal{T}_{\mathrm{sub}}$):

$$\sup_{f \in \mathcal{F}} \left| \widehat{\mathcal{L}}_{\mathrm{un}}(f; \mathcal{T}_{\mathrm{sub}}) - \mathcal{U}_{\mathrm{N}}(f) \right| \leqslant 2 \widehat{\mathfrak{R}}_{\mathcal{T}_{\mathrm{sub}}}(\ell \circ \mathcal{F}) + 3\mathcal{M}\sqrt{\frac{\ln 4/\Delta}{2\mathrm{M}}}. \tag{55}$$

In particular, we have $\mathbb{P}\left(\text{Eqn. (55) does not hold}\right) = \mathbb{E}_{\mathcal{S}} \mathbb{P}\left(\text{Eqn. (55) does not hold} \Big| \mathcal{S}\right) \leqslant \mathbb{E}_{\mathcal{S}}[\Delta] = \Delta$: the overall failure probability of Eqn. (55) is less than $\Delta$. Furthermore, by Theorem D.10, with probability of at least $1 - \Delta$ (with respect to the draw of $\mathcal{S}$), we have:

$$\sup_{f \in \mathcal{F}} \left| \mathcal{U}_{\mathrm{N}}(f) - \mathrm{L}_{\mathrm{un}}(f) \right| \leqslant \frac{4}{\sqrt{\widetilde{\mathrm{N}}}} \sum_{c \in \mathcal{C}} \rho(c) K_{\mathcal{F},c} + 22\mathcal{M}\sqrt{\frac{\ln 8|\mathcal{C}|/\Delta}{2\widetilde{\mathrm{N}}}}. \tag{56}$$

Combining Eqn. (55) and Eqn. (56) using a Union bound, with probability of at least $1 - 2\Delta$ (with respect to the draw of $\mathcal{S}$ and $\mathcal{T}_{\mathrm{sub}}$), we have:

$$\mathrm{L}_{\mathrm{un}}(\widehat{f}_{\mathrm{sub}}) - \inf_{f \in \mathcal{F}} \mathrm{L}_{\mathrm{un}}(f) \leqslant 4 \widehat{\mathfrak{R}}_{\mathcal{T}_{\mathrm{sub}}}(\ell \circ \mathcal{F}) + \frac{8}{\sqrt{\widetilde{\mathrm{N}}}} \sum_{c \in \mathcal{C}} \rho(c) K_{\mathcal{F},c} + 6\mathcal{M}\sqrt{\frac{\ln 4/\Delta}{2\mathrm{M}}} + 44\mathcal{M}\sqrt{\frac{\ln 8|\mathcal{C}|/\Delta}{2\widetilde{\mathrm{N}}}}.$$

Setting $\Delta = \delta/2$ yields the desired bound. □

### E.3. Applications to Common Function Classes

Without reiteration, we define $\widehat{f}_{\mathrm{sub}} = \arg\min_{f \in \mathcal{F}} \widehat{\mathcal{L}}(f; \mathcal{T}_{\mathrm{sub}})$ by default. Applying the above result, we can easily obtain the generalization bounds for the common classes (linear functions, neural networks) by bounding $\widehat{\mathfrak{R}}_{\mathcal{T}_{\mathrm{sub}}}(\ell \circ \mathcal{F})$ using Dudley entropy integral (Theorem F.5) and applying Theorem D.10.

#### E.3.1. LINEAR FUNCTIONS

**Theorem E.2** (Linear Functions)**.** *Let* $a, s \in \mathbb{R}_+$ *be given and* $\mathcal{F}_{\mathrm{lin}}$ *be the class of linear representation functions (also defined in Eqn. (17)), defined as follows:*

$$\mathcal{F}_{\mathrm{lin}} = \left\{ x \mapsto Ax : A \in \mathbb{R}^{d \times m}, \|A^\top\|_{2,1} \leqslant a, \|A\|_\sigma \leqslant s \right\}. \tag{57}$$

*Let $\widetilde{N} = N \min\left(\frac{\rho_{\min}}{2}, \frac{1-\rho_{\max}}{k}\right)$ and $\|\mathbf{x}\|_2 \leqslant b, \forall \mathbf{x} \in \mathcal{S}$ with probability one. Suppose that the loss $\ell : \mathbb{R}^k \to \mathbb{R}_+$ is $\ell^\infty$-Lipschitz with constant $\eta > 0$. Then, for any $\delta \in (0,1)$, with probability of at least $1 - \delta$, we have:*

$$
L_{\mathrm{un}}(\widehat{f}_{\mathrm{sub}}) - \inf_{f \in \mathcal{F}_{\mathrm{lin}}} L_{\mathrm{un}}(f) \leqslant \frac{4}{M} + \frac{32}{N\sqrt{\widetilde{N}}} + 3072\sqrt{2}\eta sab^2 \ln\left(\left(44N\eta sab^2 + 7\right)N(k+2)d\right)\ln(N\mathcal{M})\left(\frac{1}{\sqrt{M}} + \frac{1}{\sqrt{\widetilde{N}}}\right)
$$
$$
+ 6\mathcal{M}\sqrt{\frac{\ln 8/\delta}{2M}} + 44\mathcal{M}\sqrt{\frac{\ln 16|\mathcal{C}|/\delta}{2\widetilde{N}}}.
$$

$$(58)$$

**Proof.** By Theorem D.15, for all $c \in \mathcal{C}$, we have $\widehat{\mathfrak{R}}_{\mathcal{T}_c^{\mathrm{iid}}}(\ell \circ \mathcal{F}_{\mathrm{lin}}) \leqslant 4N^{-1} + \overline{N_c}^{-\frac{1}{2}} 384\sqrt{2}\eta sab^2 \phi$ where $\phi$ is a logarithmic function of $N, \mathcal{M}, \eta, s, a, b, k$ and $d$, defined by $\phi = \ln\left(\left(44N\eta sab^2 + 7\right)N(k+2)d\right)\ln(N\mathcal{M})$. Then, by Theorem E.1, with probability of at least $1 - \delta$ for any $\delta \in (0,1)$, we have:

$$
L_{\mathrm{un}}(\widehat{f}_{\mathrm{sub}}) - \inf_{f \in \mathcal{F}_{\mathrm{lin}}} L_{\mathrm{un}}(f) \leqslant \widehat{\mathfrak{R}}_{\mathcal{T}_{\mathrm{sub}}}(\ell \circ \mathcal{F}_{\mathrm{lin}}) + \frac{32}{N\sqrt{\widetilde{N}}} + \frac{3072\sqrt{2}\eta sab^2 \phi}{\sqrt{\widetilde{N}}} + 6\mathcal{M}\sqrt{\frac{\ln 8/\delta}{2M}} + 44\mathcal{M}\sqrt{\frac{\ln 16|\mathcal{C}|/\delta}{2\widetilde{N}}}. \quad (59)
$$

Using Dudley entropy integral bound (Theorem F.5) in the same way as Theorem D.15, we have:

$$
\widehat{\mathfrak{R}}_{\mathcal{T}_{\mathrm{sub}}}(\ell \circ \mathcal{F}_{\mathrm{lin}}) \leqslant \frac{4}{M} + \frac{384\sqrt{2}\eta sab^2 \phi}{\sqrt{M}}. \quad (60)
$$

Combining Eqn. (59) and Eqn. (60), with probability of at least $1 - \delta$, we have:

$$
L_{\mathrm{un}}(\widehat{f}_{\mathrm{sub}}) - \inf_{f \in \mathcal{F}_{\mathrm{lin}}} L_{\mathrm{un}}(f) \leqslant \frac{4}{M} + \frac{32}{N\sqrt{\widetilde{N}}} + 3072\sqrt{2}\eta sab^2 \phi\left(\frac{1}{\sqrt{M}} + \frac{1}{\sqrt{\widetilde{N}}}\right) + 6\mathcal{M}\sqrt{\frac{\ln 8/\delta}{2M}} + 44\mathcal{M}\sqrt{\frac{\ln 16|\mathcal{C}|/\delta}{2\widetilde{N}}},
$$

as desired. $\qquad\square$

### E.3.2. NEURAL NETWORKS

**Theorem E.3** (Neural Networks). *Let $L \geqslant 1$ and $d_0, d_1, \ldots, d_L$ be positive integers representing layer widths. For $1 \leqslant 1 \leqslant L$, let $\mathcal{B}_l = \left\{A^{(l)} \in \mathbb{R}^{d_{l-1} \times d_l} : \|A^{(l)}\|_\sigma \leqslant s_l\right\}$ be parameter spaces where $\{s_l\}_{l=1}^L$ is a sequence of known real positive numbers and $\{\varphi_l : \mathbb{R}^{d_l} \to \mathbb{R}^{d_l}\}_{l=1}^L$ be the sequence of activation functions that are fixed a priori and $\ell^2$-Lipschitz with constants $\{\xi_l\}_{l=1}^L$. Let $\mathcal{F}_{\mathrm{nn}}^{\mathcal{A}}$ be the class of neural networks (also defined in Eqn. (18)) defined as follows:*

$$
\mathcal{F}_{\mathrm{nn}}^{\mathcal{A}} = \mathcal{F}_L \circ \mathcal{F}_{L-1} \circ \cdots \circ \mathcal{F}_1, \text{ where } \mathcal{F}_l = \left\{x \mapsto \varphi_l\left(A^{(l)}x\right) : A^{(l)} \in \mathcal{B}_l\right\} \quad (61)
$$

*Let $\widetilde{N} = N \min\left(\frac{\rho_{\min}}{2}, \frac{1-\rho_{\max}}{k}\right)$ and $\|\mathbf{x}\|_2 \leqslant b, \forall \mathbf{x} \in \mathcal{S}$ with probability one. Suppose that the contrastive loss $\ell : \mathbb{R}^k \to \mathbb{R}_+$ is $\ell^\infty$-Lipschitz with constant $\eta > 0$. Then, for any $\delta \in (0,1)$, with probability of at least $1 - \delta$, we have:*

$$
L_{\mathrm{un}}(\widehat{f}_{\mathrm{sub}}) - \inf_{f \in \mathcal{F}_{\mathrm{nn}}^{\mathcal{A}}} L_{\mathrm{un}}(f) \leqslant \frac{4}{M} + \frac{32}{N\sqrt{\widetilde{N}}} + 24\mathcal{M}\mathcal{W}^{\frac{1}{2}}\ln^{\frac{1}{2}}\left(12\eta NLb^2 \prod_{m=1}^L \xi_m^2 s_m^2 + 1\right)\left(\frac{1}{\sqrt{\widetilde{N}}} + \frac{1}{\sqrt{M}}\right)
$$
$$
+ 6\mathcal{M}\sqrt{\frac{\ln 8/\delta}{2M}} + 44\mathcal{M}\sqrt{\frac{\ln 16|\mathcal{C}|/\delta}{2\widetilde{N}}},
$$

$$(62)$$

*where $\mathcal{W} = \sum_{l=1}^L d_l$ is the total number of neurons in the neural networks.*

**Proof.** By Theorem D.17, for all $c \in \mathcal{C}$, we have $\widehat{\mathfrak{R}}_{\mathcal{T}_c^{\mathrm{iid}}}(\ell \circ \mathcal{F}_{\mathrm{nn}}^{\mathcal{A}}) \leqslant 4\mathrm{N}^{-1} + \overline{\mathrm{N}}_c^{-\frac{1}{2}} 24\mathcal{M}\mathcal{W}^{\frac{1}{2}} \ln^{\frac{1}{2}} \left( 12\eta \mathrm{N} L b^2 \prod_{m=1}^L \xi_m^2 s_m^2 + 1 \right)$. Applying Theorem E.1 yields:

$$
\begin{aligned}
\mathrm{L}_{\mathrm{un}}(\widehat{f}_{\mathrm{sub}}) - \inf_{f \in \mathcal{F}_{\mathrm{nn}}^{\mathcal{A}}} \mathrm{L}_{\mathrm{un}}(f) \leqslant{} & \widehat{\mathfrak{R}}_{\mathcal{T}_{\mathrm{sub}}}(\ell \circ \mathcal{F}_{\mathrm{nn}}^{\mathcal{A}}) + \frac{32}{\mathrm{N}\sqrt{\widetilde{\mathrm{N}}}} + \frac{24\mathcal{M}\mathcal{W}^{\frac{1}{2}} \ln^{\frac{1}{2}} \left( 12\eta \mathrm{N} L b^2 \prod_{m=1}^L \xi_m^2 s_m^2 + 1 \right)}{\sqrt{\widetilde{\mathrm{N}}}} \\
& + 6\mathcal{M}\sqrt{\frac{\ln 8/\delta}{2\mathrm{M}}} + 44\mathcal{M}\sqrt{\frac{\ln 16|\mathcal{C}|/\delta}{2\widetilde{\mathrm{N}}}},
\end{aligned}
\tag{63}
$$

with probability of at least $1 - \delta$ for any $\delta \in (0, 1)$. Applying Dudley entropy integral bound (Theorem F.5) again for $\widehat{\mathfrak{R}}_{\mathcal{T}_{\mathrm{sub}}}(\ell \circ \mathcal{F}_{\mathrm{nn}}^{\mathcal{A}})$, we have:

$$
\widehat{\mathfrak{R}}_{\mathcal{T}_{\mathrm{sub}}}(\ell \circ \mathcal{F}_{\mathrm{nn}}^{\mathcal{A}}) \leqslant \frac{4}{\mathrm{M}} + \frac{24\mathcal{M}\mathcal{W}^{\frac{1}{2}} \ln^{\frac{1}{2}} \left( 12\eta \mathrm{N} L b^2 \prod_{m=1}^L \xi_m^2 s_m^2 + 1 \right)}{\sqrt{\mathrm{M}}}.
\tag{64}
$$

Combining Eqn. (63) and Eqn. (64), we have:

$$
\begin{aligned}
\mathrm{L}_{\mathrm{un}}(\widehat{f}_{\mathrm{sub}}) - \inf_{f \in \mathcal{F}_{\mathrm{nn}}^{\mathcal{A}}} \mathrm{L}_{\mathrm{un}}(f) \leqslant{} & \frac{4}{\mathrm{M}} + \frac{32}{\mathrm{N}\sqrt{\widetilde{\mathrm{N}}}} + 24\mathcal{M}\mathcal{W}^{\frac{1}{2}} \ln^{\frac{1}{2}} \left( 12\eta \mathrm{N} L b^2 \prod_{m=1}^L \xi_m^2 s_m^2 + 1 \right) \left( \frac{1}{\sqrt{\widetilde{\mathrm{N}}}} + \frac{1}{\sqrt{\mathrm{M}}} \right) \\
& + 6\mathcal{M}\sqrt{\frac{\ln 8/\delta}{2\mathrm{M}}} + 44\mathcal{M}\sqrt{\frac{\ln 16|\mathcal{C}|/\delta}{2\widetilde{\mathrm{N}}}},
\end{aligned}
$$

with probability of at least $1 - \delta$ for any $\delta \in (0, 1)$, as desired. $\qquad \square$

# F. Classic Learning Theory Results

## F.1. Rademacher Complexity

**Definition F.1** (Rademacher Complexity). Let $\mathcal{Z}$ be a vector space and $\mathcal{D}$ be a distribution over $\mathcal{Z}$. Let $\mathcal{G}$ be a class of functions $g : \mathcal{Z} \to [a, b]$ where $a, b \in \mathbb{R}$ and $a < b$. Let $S = \{\mathbf{z}_1, \ldots, \mathbf{z}_n\}$ be a dataset drawn *i.i.d.* from $\mathcal{D}$. Then, the *empirical Rademacher complexity* of $\mathcal{G}$ is defined as follows:

$$
\widehat{\mathfrak{R}}_S(\mathcal{G}) = \mathbb{E}_{\mathbf{\Sigma}_n} \left[ \sup_{g \in \mathcal{G}} \frac{1}{n} \left| \sum_{i=1}^n \sigma_i g(\mathbf{z}_i) \right| \right],
\tag{65}
$$

where $\mathbf{\Sigma}_n = (\sigma_1, \ldots, \sigma_n)$ is a vector of $n$ independent Rademacher variables. Additionally, the *expected Rademacher complexity* of $\mathcal{G}$ is defined as follows:

$$
\mathfrak{R}_n(\mathcal{G}) = \mathbb{E}_S \left[ \widehat{\mathfrak{R}}_S(\mathcal{G}) \right].
\tag{66}
$$

Intuitively, the Rademacher complexity is a measure of a function class' richness. If a class $\mathcal{G}$ is sufficiently diverse, there is a higher chance that given a random sequence of signs (represented by the sequence of Rademacher variables), we will be able to find a function $g \in \mathcal{G}$ that matches the signs. Hence, the Rademacher complexity will be large.

**Lemma F.2.** *Let $\mathcal{Z}$ be a vector space and $\mathcal{D}$ be a distribution over $\mathcal{Z}$. Let $\mathcal{G}$ be a class of functions $g : \mathcal{Z} \to [a, b]$ where $a, b \in \mathbb{R}$ and $a < b$. Let $S = \{\mathbf{z}_1, \ldots, \mathbf{z}_n\}$ be a dataset drawn i.i.d. from $\mathcal{D}$. Then, for any $\delta \in (0, 1)$, with probability of at least $1 - \delta$, we have:*

$$
\mathfrak{R}_n(\mathcal{G}) \leqslant \widehat{\mathfrak{R}}_S(\mathcal{G}) + (b - a)\sqrt{\frac{\ln 1/\delta}{2n}}
\tag{67}
$$

**Proof.** Let $\phi : \mathcal{Z}^n \to \mathbb{R}_+$ be defined as $\phi(x_1, \ldots, x_n) = \mathbb{E}_{\mathbf{\Sigma}_n} \left[ \sup_{g \in \mathcal{G}} \frac{1}{n} \left| \sum_{j=1}^n \sigma_j g(x_j) \right| \right]$ where $x_i \in \mathcal{Z}$ for all $1 \leqslant i \leqslant n$. Then, for all $1 \leqslant 1 \leqslant n$, we have:

$$
\sup_{x_i, x_i' \in \mathcal{Z}} \left| \phi(x_1, \ldots, x_i, \ldots, x_n) - \phi(x_1, \ldots, x_i', \ldots, x_n) \right| \leqslant \frac{b - a}{n}.
$$

Hence, by McDiarmid's inequality (McDiarmid, 1989), for any $\epsilon > 0$, we have:

$$\mathbb{P}\bigg( \underbrace{\mathbb{E}\phi(\mathbf{z}_1, \ldots, \mathbf{z}_n)}_{\mathfrak{R}_n(\mathcal{G})} - \underbrace{\phi(\mathbf{z}_1, \ldots, \mathbf{z}_n)}_{\widehat{\mathfrak{R}}_S(\mathcal{G})} \geqslant \epsilon \bigg) \leqslant \exp\bigg( -\frac{2n\epsilon^2}{(b-a)^2} \bigg).$$

Setting the right-hand-side to $\delta$, we have $\epsilon = (b-a)\sqrt{\frac{\ln 1/\delta}{2n}}$ and we obtained the desired bound. $\qquad\square$

**Proposition F.3** (Rademacher Complexity Bound). *Let $\mathcal{Z}$ be a vector space and $\mathcal{D}$ be a distribution over $\mathcal{Z}$. Let $\mathcal{G}$ be a class of functions $g : \mathcal{Z} \to [a, b]$ where $a, b \in \mathbb{R}$ and $a < b$. Let $S = \{\mathbf{z}_1, \ldots, \mathbf{z}_n\}$ be a dataset drawn i.i.d. from $\mathcal{D}$. Then, for any $\delta \in (0, 1)$, with probability of at least $1 - \delta$, we have:*

$$\sup_{g \in \mathcal{G}} \left| \mathbb{E}_{\mathbf{z} \sim \mathcal{D}}[g(\mathbf{z})] - \frac{1}{n}\sum_{i=1}^{n} g(\mathbf{z}_i) \right| \leqslant 2\widehat{\mathfrak{R}}_S(\mathcal{G}) + 3(b-a)\sqrt{\frac{\ln 4/\delta}{2n}}. \tag{68}$$

**Proof.** Let $\phi : \mathcal{Z}^n \to \mathbb{R}$ be defined as $\phi(x_1, \ldots, x_n) = \sup_{g \in \mathcal{G}}\big[ \mathbb{E}_{\mathbf{z} \sim \mathcal{D}}[g(\mathbf{z})] - \frac{1}{n}\sum_{i=1}^{n} g(x_i) \big]$. Using McDiarmid's inequality, for all $\Delta \in (0, 1)$, the following inequality holds with probability of at least $1 - \Delta/2$:

$$\phi(\mathbf{z}_1, \ldots, \mathbf{z}_n) \leqslant \mathbb{E}_S\phi(\mathbf{z}_1, \ldots, \mathbf{z}_n) + (b-a)\sqrt{\frac{\ln 2/\Delta}{2n}}$$

$$\leqslant 2\mathfrak{R}_n(\mathcal{G}) + (b-a)\sqrt{\frac{\ln 2/\Delta}{2n}}. \qquad \text{(Symmetrization - Lemma F.6)}$$

Furthermore, we have $\mathfrak{R}_n(\mathcal{G}) \leqslant \widehat{\mathfrak{R}}_S(\mathcal{G}) + (b-a)\sqrt{\frac{\ln 2/\Delta}{2n}}$ with probability of at least $1 - \Delta/2$ (Lemma F.2). Hence, by the Union bound, with probability of at least $1 - \Delta$, we have:

$$\sup_{g \in \mathcal{G}} \left[ \mathbb{E}_{\mathbf{z} \sim \mathcal{D}}[g(\mathbf{z})] - \frac{1}{n}\sum_{i=1}^{n} g(x_i) \right] \leqslant 2\widehat{\mathfrak{R}}_S(\mathcal{G}) + 3(b-a)\sqrt{\frac{\ln 2/\Delta}{2n}}.$$

Let $\phi(x_1, \ldots, x_n) = \sup_{g \in \mathcal{G}}\big[ \frac{1}{n}\sum_{i=1}^{n} g(x_i) - \mathbb{E}_{\mathbf{z} \sim \mathcal{D}}[g(\mathbf{z})] \big]$ and repeat the above argument, we have the inequality in the other direction with probability of at least $1 - \Delta$. Hence, by the Union bound, with probability of at least $1 - 2\Delta$, we have the following two-sided inequality:

$$\sup_{g \in \mathcal{G}} \left| \mathbb{E}_{\mathbf{z} \sim \mathcal{D}}[g(\mathbf{z})] - \frac{1}{n}\sum_{i=1}^{n} g(\mathbf{z}_i) \right| \leqslant 2\widehat{\mathfrak{R}}_S(\mathcal{G}) + 3(b-a)\sqrt{\frac{\ln 2/\Delta}{2n}}.$$

Setting $\Delta = \delta/2$ completes the proof. $\qquad\square$

### F.2. Massart Lemma & Dudley's Entropy Integral

**Lemma F.4** (Massart's Finite Lemma). *Given $S = \{\mathbf{z}_1, \ldots, \mathbf{z}_n\}$ be an i.i.d. sample from a given distribution. Let $\mathcal{G}$ be a finite function class. Then, we have:*

$$\mathbb{E}_{\boldsymbol{\Sigma}_n}\left[ \sup_{g \in \mathcal{G}} \frac{1}{n}\left| \sum_{i=1}^{n} \sigma_i g(\mathbf{z}_i) \right| \right] \leqslant B\sqrt{\frac{2\ln 2|\mathcal{G}|}{n}}, \tag{69}$$

*where $\boldsymbol{\Sigma}_n = \{\sigma_1, \ldots, \sigma_n\}$ is a sequence of i.i.d. Rademacher variables and $B = \sup_{g \in \mathcal{G}} \big( \frac{1}{n}\sum_{i=1}^{n} |g(\mathbf{z}_i)|^2 \big)^{1/2}$.*

**Proof.** For each $h \in \mathcal{G}$, we denote $\theta_h = \frac{1}{n}\sum_{i=1}^{n}\sigma_i g(\mathbf{z}_i)$. Then, for $\lambda > 0$, we have:

$$\lambda\mathbb{E}_{\boldsymbol{\Sigma}_n}\left[\sup_{h\in\mathcal{G}}\frac{1}{n}\left|\sum_{i=1}^{n}\sigma_i g(\mathbf{z}_i)\right|\right] = \lambda\mathbb{E}_{\boldsymbol{\Sigma}_n}\left[\max_{h\in\mathcal{G}}|\theta_h|\right] \quad (\text{sup} \to \max \text{ due to finite } \mathcal{G})$$

$$= \ln\exp\lambda\mathbb{E}_{\boldsymbol{\Sigma}_n}\left[\max_{h\in\mathcal{G}}|\theta_h|\right]$$

$$= \ln\exp\lambda\mathbb{E}_{\boldsymbol{\Sigma}_n}\left[\max_{h\in\mathcal{G}}\max(\theta_h, -\theta_h)\right]$$

$$\leqslant \ln\mathbb{E}_{\boldsymbol{\Sigma}_n}\exp\left(\lambda\max_{h\in\mathcal{G}}\max(\theta_h, -\theta_h)\right) \quad (\text{Jensen's Inequality})$$

$$= \ln\mathbb{E}_{\boldsymbol{\Sigma}_n}\max_{h\in\mathcal{G}}\exp\left(\lambda\max(\theta_h, -\theta_h)\right) \quad (\exp(\lambda x) \text{ is an increasing function})$$

$$\leqslant \ln\sum_{h\in\mathcal{G}}\mathbb{E}_{\boldsymbol{\Sigma}_n}\left[\exp(\lambda\theta_h) + \exp(-\lambda\theta_h)\right] \quad (\exp\max \leqslant \exp\text{ sum})$$

$$= \ln 2\sum_{h\in\mathcal{G}}\mathbb{E}_{\boldsymbol{\Sigma}_n}\left[\exp(\lambda\theta_h)\right] \quad (\text{By symmetry of } \sigma)$$

$$= \ln 2\sum_{h\in\mathcal{G}}\prod_{i=1}^{n}\mathbb{E}_{\boldsymbol{\Sigma}_n}\left[\exp\left(\frac{\lambda}{n}\sigma_i g(\mathbf{z}_i)\right)\right] \quad (\text{Since all } \sigma_i\text{'s are independent})$$

$$= \ln 2\sum_{h\in\mathcal{G}}\prod_{i=1}^{n}\frac{\exp\left(-\frac{\lambda}{n}g(\mathbf{z}_i)\right) + \exp\left(\frac{\lambda}{n}g(\mathbf{z}_i)\right)}{2}.$$

Using the inequality $e^x + e^{-x} \leqslant 2e^{\frac{x^2}{2}}$, we have:

$$\lambda\mathbb{E}_{\boldsymbol{\Sigma}_n}\left[\sup_{h\in\mathcal{G}}\frac{1}{n}\left|\sum_{i=1}^{n}\sigma_i g(\mathbf{z}_i)\right|\right] \leqslant \ln 2\sum_{h\in\mathcal{G}}\prod_{i=1}^{n}\exp\left(\frac{\lambda^2 g(\mathbf{z}_i)^2}{2n^2}\right)$$

$$= \ln 2\sum_{h\in\mathcal{G}}\exp\left(\frac{\lambda^2\sum_{i=1}^{n}g(\mathbf{z}_i)^2}{2n^2}\right)$$

$$\leqslant \ln 2|\mathcal{G}| \cdot \exp\left(\frac{\lambda^2 B^2}{2n}\right)$$

$$= \ln 2|\mathcal{G}| + \frac{\lambda^2 B^2}{2n}.$$

From the above, let $\lambda = B^{-1}\sqrt{2n\ln 2|\mathcal{G}|}$ and we obtain the desired bound. $\qquad\square$

**Theorem F.5** (Dudley's Entropy Integral). *Let $\mathcal{G}$ be a real-valued function class and $S = \{\mathbf{z}_1, \ldots, \mathbf{z}_n\}$ be a sample drawn i.i.d. from a fixed distribution. Then, we have:*

$$\widehat{\mathfrak{R}}_S(\mathcal{G}) \leqslant \inf_{\alpha>0}\left(4\alpha + 12\int_{\alpha}^{B}\sqrt{\frac{\ln 2\mathcal{N}(\mathcal{G}, \epsilon, \mathrm{L}_2(S))}{n}}d\epsilon\right), \tag{70}$$

*where $B = \sup_{g\in\mathcal{G}}\left(\frac{1}{n}\sum_{i=1}^{n}g(\mathbf{z}_i)^2\right)^{1/2}$ and $\widehat{\mathfrak{R}}_S(\mathcal{G})$ is the empirical Rademacher complexity of the function class $\mathcal{G}$.*

**Proof.** The above result is derived by a standard chaining argument (See, for example, Ledent et al. (2021, Proposition 22) or Bartlett et al. (2017, Lemma A.5)). Then, apply Massart's finite lemma. The difference is that instead of using the standard Massart's lemma for the regular notion of Rademacher Complexity (without the absolute value surrounding the sum), we apply lemma F.4. $\qquad\square$

### F.3. Symmetrization Inequality

**Lemma F.6** (Symmetrization). *Let* $S = \{\mathbf{z}_1, \ldots, \mathbf{z}_n\}$ *be a sample drawn i.i.d. from a distribution* $\mathcal{P}$. *Let* $\mathcal{G}$ *denote a function class. Then, for any real-valued non-decreasing function* $\varphi$, *we have:*

$$\mathbb{E}_S \varphi \left[ \sup_{g \in \mathcal{G}} \left| \frac{1}{n} \sum_{i=1}^{n} g(\mathbf{z}_i) - \mathbb{E}_{\mathbf{z} \sim \mathcal{P}}[g(\mathbf{z})] \right| \right] \leqslant \mathbb{E}_{S, \mathbf{\Sigma}_n} \varphi \left[ 2\mathcal{R}_{\mathcal{G}}^{S, \mathbf{\Sigma}_n} \right], \tag{71}$$

*where* $\mathbf{\Sigma}_n = \{\sigma_1, \ldots, \sigma_n\}$ *is the sequence of i.i.d. Rademacher variables.*

**Proof.** Denote $\mathbb{E}_S$ as the expectation taken over the sample $S$. We introduce another sample $S' = \{\mathbf{z}'_1, \ldots, \mathbf{z}'_n\}$ that is identically distributed as $S$. We have:

$$\mathbb{E}_S \varphi \left[ \sup_{g \in \mathcal{G}} \frac{1}{n} \left| \sum_{i=1}^{n} (g(\mathbf{z}_i) - \mathbb{E}_S[g(\mathbf{z}_i)]) \right| \right]$$

$$= \mathbb{E}_S \varphi \left[ \sup_{g \in \mathcal{G}} \frac{1}{n} \left| \sum_{i=1}^{n} (g(\mathbf{z}_i) - \mathbb{E}_{S'}[g(\mathbf{z}'_i)]) \right| \right]$$

$$= \mathbb{E}_S \varphi \left[ \sup_{g \in \mathcal{G}} \frac{1}{n} \left| \sum_{i=1}^{n} g(\mathbf{z}_i) - \sum_{i=1}^{n} \mathbb{E}_{S'}[g(\mathbf{z}'_i)] \right| \right]$$

$$\leqslant \mathbb{E}_{S, S'} \varphi \left[ \sup_{g \in \mathcal{G}} \frac{1}{n} \left| \sum_{i=1}^{n} (g(\mathbf{z}_i) - f(\mathbf{z}'_i)) \right| \right] \quad \text{(Jensen's Inequality for } \varphi \circ \sup |\,.\,|)$$

$$= \mathbb{E}_{S, S', \mathbf{\Sigma}_n} \varphi \left[ \sup_{g \in \mathcal{G}} \frac{1}{n} \left| \sum_{i=1}^{n} \sigma_i (g(\mathbf{z}_i) - g(\mathbf{z}'_i)) \right| \right] \quad (g(\mathbf{z}_i) - g(\mathbf{z}'_i) \text{ is symmetric)}$$

$$\leqslant \frac{1}{2} \mathbb{E}_{S, \mathbf{\Sigma}_n} \varphi \left[ \sup_{g \in \mathcal{G}} \frac{2}{n} \left| \sum_{i=1}^{n} \sigma_i g(\mathbf{z}_i) \right| \right] + \frac{1}{2} \mathbb{E}_{S', \mathbf{\Sigma}_n} \varphi \left[ \sup_{g \in \mathcal{G}} \frac{2}{n} \left| \sum_{i=1}^{n} (-\sigma_i) g(\mathbf{z}'_i) \right| \right]$$

$$= \frac{1}{2} \mathbb{E}_{S, \mathbf{\Sigma}_n} \varphi \left[ \sup_{g \in \mathcal{G}} \frac{2}{n} \left| \sum_{i=1}^{n} \sigma_i g(\mathbf{z}_i) \right| \right] + \frac{1}{2} \mathbb{E}_{S', \mathbf{\Sigma}_n} \varphi \left[ \sup_{g \in \mathcal{G}} \frac{2}{n} \left| \sum_{i=1}^{n} \sigma_i g(\mathbf{z}'_i) \right| \right] \quad \text{(Rademacher variables are symmetric)}$$

$$= \mathbb{E}_{S, \mathbf{\Sigma}_n} \varphi \left[ \sup_{g \in \mathcal{G}} \frac{2}{n} \left| \sum_{i=1}^{n} \sigma_i g(\mathbf{z}_i) \right| \right] \quad (S \text{ and } S' \text{ are identically distributed)}$$

$$= \mathbb{E}_{S, \mathbf{\Sigma}_n} \varphi \left[ 2\mathcal{R}_{\mathcal{G}}^{S, \mathbf{\Sigma}_n} \right].$$

$\square$

### F.4. Sub-Gaussianity of Rademacher Complexity

**Definition F.7** (Sub-Gaussian Random Variable). Let $X$ be a random variable with mean $\mathbb{E}[X] = \mu$. $X$ is then called sub-Gaussian if there exists $\xi > 0$ such that:

$$M_X(t) \leqslant \exp \left( t\mu + \frac{t^2 \xi^2}{2} \right), \quad \forall t > 0, \tag{72}$$

where $M_X$ denotes the moment generating function of $X$. We call $X$ a sub-Gaussian random variable with variance proxy $\xi^2$, denoted as $X \in \mathcal{SG}(\xi^2)$.

**Lemma F.8** ((Boucheron et al., 2013), Theorem 2.1). *Let* $X$ *be a random variable with mean* $\mathbb{E}[X] = \mu$. *If there exists* $\xi > 0$ *such that the following holds:*

$$\mathbb{P}(|X - \mu| \geqslant t) \leqslant 2\exp \left( -\frac{t^2}{2\xi^2} \right), \tag{73}$$

*then the random variable $X$ is sub-Gaussian. Specifically, $X \in \mathcal{SG}(16\xi^2)$.*

**Proof.** Let $Z = X - \mu$ be the centered random variable derived by translating $X$ by its mean. Firstly, we prove that Eqn. (73) implies that $\mathbb{E}|Z|^{2q} \leqslant q!(4\xi^2)^q$ for all integers $q \geqslant 1$. Using the identity $\mathbb{E}|Z|^q = \int_0^\infty qt^{q-1}\mathbb{P}(|Z| \geqslant t)dt$, we have:

$$\mathbb{E}|Z|^{2q} = 2q \int_0^\infty t^{2q-1}\mathbb{P}(|Z| \geqslant t)dt$$

$$\leqslant 4q \int_0^\infty t^{2q-1} \exp\left(-\frac{t^2}{2\xi^2}\right)dt.$$

Letting $u = \frac{t^2}{2\xi^2}$, hence $t^2 = 2u\xi^2$ and $dt = \frac{\xi^2 du}{t}$, the above integral becomes:

$$\mathbb{E}|Z|^{2q} \leqslant 4q\xi^2 \int_0^\infty t^{2q-2}e^{-u}du$$

$$= 4q\xi^2 \int_0^\infty (2u\xi^2)^{q-1}e^{-u}du$$

$$= 2q \cdot (2\xi^2)^q \underbrace{\int_0^\infty u^{q-1}e^{-u}du}_{\Gamma(q)}$$

$$= 2q!(2\xi^2)^q \leqslant q!(4\xi^2)^q.$$

Let $\tilde{Z}$ be the *i.i.d.* copy of $Z$. Hence, $Z - \tilde{Z}$ is symmetric about 0, which means that $\mathbb{E}[(Z - \tilde{Z})^p] = 0$ for odd-order $p$-moments. Therefore, For all $\lambda > 0$, we have:

$$M_Z(\lambda)M_{-\tilde{Z}}(\lambda) = M_{Z-\tilde{Z}}(\lambda) \quad \text{(Due to independence)}$$

$$= \mathbb{E}\exp\left(\lambda(Z - \tilde{Z})\right)$$

$$= 1 + \sum_{q=1}^\infty \frac{\lambda^{2q}\mathbb{E}\left[(Z - \tilde{Z})^{2q}\right]}{(2q)!}.$$

By the convexity of $f(z) = z^{2q}$, for all $t \in (0, 1)$, we have:

$$\left[tZ + (1-t)(-\tilde{Z})\right]^{2q} \leqslant tZ^{2q} + (1-t)\tilde{Z}^{2q}.$$

Setting $t = \frac{1}{2}$, we have:

$$\left[\frac{Z - \tilde{Z}}{2}\right]^{2q} \leqslant \frac{Z^{2q} + \tilde{Z}^{2q}}{2} \implies (Z - \tilde{Z})^{2q} \leqslant 2^{2q-1}(Z^{2q} + \tilde{Z}^{2q}).$$

As a result, we have $\mathbb{E}[(Z - \tilde{Z})^{2q}] \leqslant 2^{2q-1}(\mathbb{E}[Z^{2q}] + \mathbb{E}[\tilde{Z}^{2q}]) = 2^{2q}\mathbb{E}[Z^{2q}]$. Plugging this back to the formula of $M_{Z-\tilde{Z}}(\lambda)$, we have:

$$\mathbb{E}[e^{\lambda Z}]\mathbb{E}[e^{-\lambda\tilde{Z}}] \leqslant 1 + \sum_{q=1}^\infty \frac{\lambda^{2q}2^{2q}\mathbb{E}[Z^{2q}]}{(2q)!}$$

$$\leqslant 1 + \sum_{q=1}^\infty \frac{\lambda^{2q}2^{2q}(4\xi^2)^q q!}{(2q)!}.$$

Since $\mathbb{E}[e^{-\lambda \tilde{Z}}] \geqslant 1$ for all $\lambda > 0$ and

$$\frac{(2q)!}{q!} = \prod_{j=1}^{q}(q+j) \geqslant \prod_{j=1}^{q}(2j) = 2^q q!,$$

We have:

$$\mathbb{E}[e^{\lambda Z}] \leqslant 1 + \sum_{q=1}^{\infty}\frac{(2\lambda^2 \cdot 4\xi^2)^q}{q!} = 1 + \sum_{q=1}^{\infty}\frac{(8\lambda^2\xi^2)^q}{q!} = e^{8\lambda^2\xi^2}.$$

Therefore, we have:

$$M_X(\lambda) = e^{\lambda\mu}\mathbb{E}[e^{\lambda Z}] \leqslant \exp\left(\lambda\mu + \frac{16\lambda^2\xi^2}{2}\right).$$

Hence, by definition, we have $X \in \mathcal{SG}(16\xi^2)$ as desired. $\qquad\square$

**Lemma F.9.** *Let $\mathcal{F}$ be a class of bounded functions $f : \mathcal{X} \to [0, \mathcal{M}]$ and let $S = \{\mathbf{x}_1, \dots, \mathbf{x}_n\}$ be sampled i.i.d. from a given distribution $\mathcal{P}$. Let $\mathbf{\Sigma}_n = \{\sigma_1, \dots, \sigma_n\}$ be a sequence of independent Rademacher variables and define the following random variable:*

$$\mathcal{R}_{\mathcal{F}}^{S,\mathbf{\Sigma}_n} = \sup_{f\in\mathcal{F}}\left|\frac{1}{n}\sum_{j=1}^{n}\sigma_j f(\mathbf{x}_j)\right|, \tag{74}$$

*which is a function of both $S$ and $\mathbf{\Sigma}_n$. Then, we have:*

$$\mathbb{E}_{S,\mathbf{\Sigma}_n}\exp\left(t\mathcal{R}_{\mathcal{F}}^{S,\mathbf{\Sigma}_n}\right) \leqslant \exp\left(t\mathfrak{R}_n(\mathcal{F}) + \frac{16t^2\mathcal{M}^2}{2n}\right), \quad \forall t > 0,$$

*where $\mathfrak{R}_n(\mathcal{F})$ is the expected Rademacher complexity, defined as $\mathfrak{R}_n(\mathcal{F}) = \mathbb{E}_{S,\mathbf{\Sigma}_n}\left[\mathcal{R}_{\mathcal{F}}^{S,\mathbf{\Sigma}_n}\right]$.*

**Proof.** Let $S_i$ be the copy of $S$ with the $i^{th}$ element replaced with $\mathbf{x}'_i$, an i.i.d. copy of $\mathbf{x}_i$. Similarly, let $\mathbf{\Sigma}_n^{(l)}$ be the copy of $\mathbf{\Sigma}_n$ where the $l^{th}$ element is replaced with $\sigma'_l$, an i.i.d. copy of $\sigma_l$. Then, we have:

$$\left|\mathcal{R}_{\mathcal{F}}^{S,\mathbf{\Sigma}_n} - \mathcal{R}_{\mathcal{F}}^{S_i,\mathbf{\Sigma}_n}\right| \leqslant \sup_{f\in\mathcal{F}}\left|\frac{1}{n}\sigma_i(f(\mathbf{x}_i) - f(\mathbf{x}'_i))\right| \leqslant \frac{2\mathcal{M}}{n},$$

$$\left|\mathcal{R}_{\mathcal{F}}^{S,\mathbf{\Sigma}_n} - \mathcal{R}_{\mathcal{F}}^{S,\mathbf{\Sigma}_n^{(l)}}\right| \leqslant \sup_{f\in\mathcal{F}}\left|\frac{1}{n}f(\mathbf{x}_l)(\sigma_l - \sigma'_l)\right| \leqslant \frac{2\mathcal{M}}{n}.$$

Hence, by McDiarmid's inequality, we have:

$$\mathbb{P}\left(\left|\mathcal{R}_{\mathcal{F}}^{S,\mathbf{\Sigma}_n} - \mathfrak{R}_n(\mathcal{F})\right| \geqslant t\right) \leqslant 2\exp\left(-\frac{t^2}{4n^{-1}\mathcal{M}^2}\right), \quad \forall t > 0.$$

Let $\xi^2 = \frac{2\mathcal{M}^2}{n}$. Then, by lemma F.8, the random variable $\mathcal{R}_{\mathcal{F}}^{S,\mathbf{\Sigma}_n}$ is sub-Gaussian with variance proxy of $16\xi^2 = \frac{32\mathcal{M}^2}{n}$. Hence, we have:

$$\mathbb{E}_{S,\mathbf{\Sigma}_n}\exp\left(t\mathcal{R}_{\mathcal{F}}^{S,\mathbf{\Sigma}_n}\right) \leqslant \exp\left(t\mathfrak{R}_n(\mathcal{F}) + \frac{16t^2\mathcal{M}^2}{n}\right), \quad \forall t > 0,$$

as desired. $\qquad\square$

# G. Further Discussions

## G.1. Some Comments on the Differences between Our Proof Strategy and that of Lei et al. (2023)

The core similarity of our work and Lei et al. (2023) is that we both bound the (empirical) Rademacher complexity of the loss class, denoted as $\widehat{\mathfrak{R}}_S(\mathcal{G})$ through the $\mathrm{L}_\infty$ covering number of the following class:

$$\mathcal{H} = \left\{ (\mathbf{x}, \mathbf{x}^+, \mathbf{x}^-) \mapsto f(\mathbf{x})^\top \left[ f(\mathbf{x}^+) - f(\mathbf{x}^-) \right] : f \in \mathcal{F} \right\}. \tag{75}$$

Specifically, suppose that the loss is $\ell^\infty$-Lipschitz with constant $\eta > 0$ and bounded by $\mathcal{M}$. By Dudley's entropy integral bound, we have:

$$\widehat{\mathfrak{R}}_S(\mathcal{G}) \lesssim \int_\alpha^{\mathcal{M}} \sqrt{\frac{\ln \mathcal{N}(\mathcal{G}, \epsilon, \mathrm{L}_2(S))}{n}} d\epsilon \leq \int_\alpha^{\mathcal{M}} \sqrt{\frac{\ln \mathcal{N}(\mathcal{H}, \epsilon/\eta, \mathrm{L}_\infty(S_{\mathcal{H}}))}{n}} d\epsilon, \quad \alpha > 0. \tag{76}$$

Where $S$ is a set of $n$ independent tuples and $S_{\mathcal{H}}$ is the set of triplets incurred from $S$ (See Lei et al. (2023), Section 4.2). At this point, the difference lies in the methods by which we estimate $\mathcal{N}(\mathcal{H}, \epsilon/\eta, \mathrm{L}_\infty(S_{\mathcal{H}}))$. In our work, the $\mathrm{L}_\infty$ covering number of $\mathcal{H}$ is **directly estimated** by the $\mathrm{L}_{\infty,2}$ covering number of the representation function class $\mathcal{F}$ (Lemma D.12). In Lei et al. (2023), the estimation is done via fat-shattering dimension and worst-case Rademacher complexity. Below, we show how the additional costs in terms of $n, k$ and their logarithmic terms propagate through complexity measures:

$$\mathfrak{R}_{WC}(\mathcal{H})^2 \xrightarrow{nk/\epsilon^2} \mathrm{fat}_\epsilon(\mathcal{H}) \xrightarrow{\ln^2(nk/\epsilon^2)} \ln \mathcal{N}_\infty(\mathcal{H}, \epsilon, \mathrm{L}_\infty(S_{\mathcal{H}})), \tag{77}$$

where $\mathfrak{R}_{WC}(\mathcal{H})$ denotes the worst-case Rademacher complexity of $\mathcal{H}$ (note that $\mathfrak{R}_{WC}(\mathcal{H}) \in \widetilde{\mathcal{O}}(1/\sqrt{nk})$) and $\mathrm{fat}_\epsilon(\mathcal{H})$ denotes the fat-shattering dimension. This yields the final inequality (See Lei et al. (2023), Eqn. (C.6)):

$$\ln \mathcal{N}(\mathcal{H}, \epsilon/\eta, \mathrm{L}_\infty(S_{\mathcal{H}})) \lesssim \frac{nk\eta^2 \mathfrak{R}_{WC}^2(\mathcal{H})}{\epsilon^2} \ln^2 \left( \frac{\eta^2 nk}{\epsilon^2} \right). \tag{78}$$

Effectively, this introduces an additional multiplicative factor of $\ln(k)$ in the main results of Lei et al. (2023) (See Theorems 4.8 and 4.9 of Lei et al. (2023)).

We also note that the works in the similar lines of research as Arora et al. (2019) and Lei et al. (2023) deal with the $i.i.d.$ tuples regime while our work is concerned with non-$i.i.d.$ regimes. In this paper, we use similar arguments as Hoeffding (1948) and Clémençon et al. (2008) to handle the non-$i.i.d.$ nature of the tuples with recycled data. This requires estimating each class-conditional risk independently, resulting in excess risk bounds of order $\widetilde{\mathcal{O}}\left( 1 \big/ \sqrt{\mathrm{N} \min\left( \rho_{\min}, \frac{1-\rho_{\max}}{k} \right)} \right)$, which are substantially different than those of Lei et al. (2023). In fact, the number $\mathrm{N} \min\left( \rho_{\min}, \frac{1-\rho_{\max}}{k} \right)$ in our bounds plays a similar role as the number "$n$" in Lei et al. (2023) - the sample complexity for the number of disjoint tuples.

## G.2. Extension of the Sub-sampling Procedure

In the procedure described in Section 3.3, the tuples are sub-sampled independently from the pool of "valid" tuples where samples are unique within tuples. This means that the samples within each tuple are dependent conditionally given the full sample $\mathbf{x}_1, \ldots, \mathbf{x}_{\mathrm{N}}$ (but independent without condition), whilst the tuples are independent with or without conditioning. **This approach corresponds to calculating U-statistics**.

It is also possible to allow replacement within tuples, in which case the samples within each tuple would instead be independent conditioned on the full sample $\mathbf{x}_1, \ldots, \mathbf{x}_{\mathrm{N}}$ but dependent if no conditioning is applied. **This approach corresponds to calculating V-statistics**. Our techniques readily extend to this case and would only incur an additive term of $\widetilde{\mathcal{O}}\left( \frac{1}{\mathrm{N} \min(\rho_{\min}/2, (1-\rho_{\max})/k)} \right)$, which does not worsen the order of magnitude in our bounds. To see why, let us consider the one-sample case for simplicity. Specifically, let $S = \{\mathbf{x}_1, \ldots, \mathbf{x}_n\}$ be a random sample drawn $i.i.d.$ from a distribution

$\mathcal{P}$ over $\mathcal{X}$ and let $h : \mathcal{X}^m \to \mathbb{R}$ be a symmetric kernel. The V-Statistics of order $m$ of the kernel $h$ is defined as:

$$V_n(h) = \frac{1}{n^m} \sum_{i_1=1}^{n} \cdots \sum_{i_m=1}^{n} h(\mathbf{x}_{i_1}, \ldots, \mathbf{x}_{i_m}). \tag{79}$$

Unlike $U_n(h)$, $V_n(h)$ is generally not an unbiased estimator of $\mathbb{E}h(\mathbf{x}_1, \ldots, \mathbf{x}_m)$ due to the terms with repeated samples. By Hoeffding (1948), we can write $V_n(h)$ as follows:

$$n^m V_n(h) = \binom{n}{m} U_n(h) + \sum_{i_{1:m} \in [n]^m \setminus \mathrm{C}_m[n]} h(\mathbf{x}_{i_1}, \ldots, \mathbf{x}_{i_m}). \tag{80}$$

As a result:

$$V_n(h) - U_n(h) = \frac{1}{n^m} \sum_{i_{1:m} \in [n]^m \setminus \mathrm{C}_m[n]} [h(\mathbf{x}_{i_1}, \ldots, \mathbf{x}_{i_m}) - U_n(h)], \tag{81}$$

where $|[n]^m \setminus C_m[n]| \in \mathcal{O}(n^{m-1})$. Therefore, if $|h(x_1, \ldots, x_m)| \leqslant M$ for all $x_1, \ldots, x_m \in \mathcal{X}$, then $V_n(h) - U_n(h) \in \mathcal{O}(M/n)$, which means the effect of biasedness dissipates when $n$ increases. Therefore, while biasedness slightly complicates analysis, we can leverage results on the concentration of U-statistics to study the concentration of V-statistics around the population risk, with a small incurred cost.

### G.3. Relevance to Semi-Supervised Contrastive Learning

The methods developed in this work is not directly tailored for semi-supervised or self-supervised regimes. However, we note that they are reasonably extendable to those settings. In this section, we outline how the analytical techniques used in this paper can be adapted to semi/self-supervised representation learning. Suppose that there exists a distribution $\mathcal{P}$ over $\mathcal{X} \times \mathcal{C}$ where $\mathcal{X}$ denotes the data space and $\mathcal{C}$ denotes the labels space (which is accessible to the learner). In semi-supervised classification, two sets of data $S = \{(x_j, y_j)\}_{j=1}^n \sim \mathcal{P}^n$ and $S_U = \{(u_j, \bar{y}_j)\}_{j=1}^m \sim \mathcal{P}^m$ are given. While both $S, S_U$ are drawn i.i.d. from $\mathcal{P}$, the labels $(\bar{y}_j)_{j \in [m]}$ are hidden, making $S_U$ unlabeled. In general, the overall risk for a representation functions $f \in \mathcal{F}$ is a combination of both supervised and unsupervised risks. As the supervised risk has been well-studied (Bartlett et al., 2017; Golowich et al., 2017; Long & Sedghi, 2020; Wei & Ma, 2020; Ledent et al., 2021), we focus on the possible formulations of unsupervised risk below instead.

**Consistency regularization (Sajjadi et al., 2016; Chen et al., 2020)**: Under this regime, the unsupervised loss aims to ensure consistency of representations under different augmentations of inputs. Specifically, let $\mathcal{A}$ be a distribution over augmentation schemes, the unsupervised risk can be defined as

$$\mathrm{L}_{\mathrm{un}}(f) = \mathbb{E}_{\substack{u \sim \mathcal{P} \\ \alpha, \alpha^+ \sim \mathcal{A}^2}} \left[ \left\| f(\alpha[u]) - f(\alpha^+[u]) \right\| \right] \tag{82}$$

where $\| \cdot \|$ is a distance measure in $\mathbb{R}^d$. Under this regime, we are implicitly given a set of augmented pairs $S_U^{\mathrm{aug}} = \{(\alpha_j[u_j], \alpha_j^+[u_j])\}_{j=1}^m$ drawn i.i.d. from a distribution dependent on both $\mathcal{P}, \mathcal{A}$ (where $\alpha_j$'s are drawn i.i.d. from $\mathcal{A}$). Here, the loss function only relies on augmented views $\alpha(x)$ and $\alpha^+(x)$ which comes from the **same** natural sample, which precludes reusing natural samples in different pairs/tuples. Thus, we can analyze this regime using standard results in learning theory without the decoupling technique.

**Self-supervised CL (HaoChen et al., 2021; Wang et al., 2022; Huang et al., 2023)**: In this case, the unsupervised risk can be defined as

$$\mathrm{L}_{\mathrm{un}}(f) = \mathbb{E}_{\substack{u, u_{1:k}^- \sim \mathcal{P}^{k+1} \\ \alpha, \alpha^+, \beta_{1:k} \sim \mathcal{A}^{k+2}}} \left[ \ell\left( \left\{ f(\alpha[u])^\top \left[ f(\alpha^+[u]) - f(\beta_i[u_i^-]) \right] \right\}_{i=1}^k \right) \right]. \tag{83}$$

Intuitively, the augmented views of the same input should be similar to each other while being dissimilar to augmented views of other samples. Since $S_U$ consists of i.i.d. data points in this setting, it is natural to formulate a $(k+2)$-order U-Statistics from $S_U$, generalizing our results to this setting. While such a learning setting involves reused samples, the lack of supervised information removes much of the subtleties of class-collision in our work. Thus, the analysis would be close to (a $k$-wise extension of) Clémençon et al. (2008), making the result simpler than our case when class constraints are present.

