# OpenReview forum: "Generalization Analysis for Supervised Contrastive Representation Learning under Non-IID Settings"
_ICML.cc/2025/Conference — ICML 2025 poster_

### Official Review · Reviewer_t8eS · 2025-02-28

**Overall Recommendation:** 3

**Summary:**

This paper proposes a modified framework where ERM is performed using a small subset of input tuples assembled from a fixed pool of label data points. It derives generalization bounds for the empirical minimizer of U-Statistics and a sub-sampled risk. The results are applied to obtain bounds for common classes of representation functions such as linear functions and neural networks.

**Claims And Evidence:**

Yes.

**Essential References Not Discussed:**

No.

**Experimental Designs Or Analyses:**

No.

**Methods And Evaluation Criteria:**

Yes.

**Other Comments Or Suggestions:**

I have only read Section 1,2,3 of this manuscript because I am very confused about the motivation and settings of this paper. If the authors can remove my doubts, I will carefully check the subsequent content.

**Other Strengths And Weaknesses:**

Strengths:

None.

Weaknesses:

1.There are some issues for references. For example, Lei et al., 2022 the authors cited should be “Lei, Y., Yang, T., Ying, Y., and Zhou, D.-X. Generalization analysis for contrastive representation learning. In International Conference in Machine Learning, **2023**.”

2.Firstly, some previous works such as Arora et al., 2019 and Lei et al., 2023 have explored **conditional** i.i.d. setting. Secondly, this paper is not the first work provide a generalization analysis for the CRL framework under non-i.i.d. settings. The authors didn’t mention those non-i.i.d. works such as HaoChen et al., 2021, Wang et al., 2022, Huang et al., 2023.

HaoChen et al., Provable Guarantees for Self-Supervised Deep Learning with Spectral Contrastive Loss. NeurIPS, 2021.
Wang et al., Chaos is a ladder：A new theoretical understanding of contrastive learning via augmentation overlap. ICLR, 2022.

Huang, et al., Towards the generalization of contrastive self-supervised learning. ICLR, 2023.

3.The sampling process adopted by the authors is unreasonable since all pre-training samples are unlabeled.

**Questions For Authors:**

1.Why do the authors define $\bar{\mathcal(D)}_c$ as Equation (2)? It does not follow the set-ups of most practical use cases since the latent label of input is unavailable. I suggest that the authors adopt the definition of Arora et al., 2019 (Equation (2) on page 2). If so, is the theoretical analysis in this paper affected by this definition, and where?

2.In the lines 141-143, the authors state that $S$ is a dataset of $n$ i.i.d. input tuples. This sentence is a little confusing. The relationship between any two tuples is certainly independent. The authors also describe their own independent sampling process on the right hand of page 3. Do the authors want to state that the $k+2$ samples in each tuple are i.i.d. ? In my opinion, the authors didn’t clarify the drawback of the i.i.d. condition of previous works. It may be better for readers to understand via an intuitive example.

3.In the part 3.3, why do the authors define $\mathcal{S}$ with some labeled samples?  Should these samples be unlabeled? If these samples are labeled, the contrastive learning studied in this paper is supervised, not unsupervised. However, the authors state that they derive some generalization bounds of unsupervised contrastive learning. It is very confusing.

**Relation To Broader Scientific Literature:**

Please see the above Summary.

**Theoretical Claims:**

No. I have only read Section 1,2,3 of this manuscript because I am very confused about the motivation and settings of this paper. If the authors can remove my doubts, I will carefully check the subsequent content.

---

> ### Author Rebuttal · Authors · 2025-04-01
>
> We truly value your thoughtful feedback. Below, we do our best to allay your concerns within the character limit, and stand ready to answer any questions in further detail during the rolling discussion.
>
> Firstly, we clarify that our work assumes $N$ **labeled** examples $x_{1:N}$ are used to form valid tuples $(x,x^+,x^-_{1:k})$ (where $x,x^+$ share a label) for training an unsupervised loss. Thus, we agree that our setting aligns more with **supervised contrastive learning**, differing from [4, 5, 6], which rely on data augmentation for positive pairs. We promise to clarify these distinctions in our revision.
>
> However, we note that using CRL loss in a supervised setting is valid. For instance, both [2, 3] assume access to i.i.d. tuples where $x,x^+$ come from the same conditional distribution, which **cannot be simulated without labels**. Empirically, using the unsupervised loss at the feature representation stage enhances downstream classification performance (See Fig. 2), aligning with previous findings [7, 10, 11].
>
> **Our work vs. [4, 5, 6]**:
>
> - The main results in [5] are Thms. 4.2 and 4.8, which bound the **population downstream classification risk** in terms of **population contrastive learning risk**. In [6], bounds are derived for the error rate of nearest-neighbor classifiers built on representation functions in terms of misalignment probability $R_\epsilon$. Thus, results in [5,6] are all at population level. In contrast, we focus on the *excess contrastive risk*.
>
> - Thm. 4.1 in [4] tackles excess contrastive risk for a loss taking *two* arguments $x,x^-$. Somewhat like ours, they construct the empirical contrastive risk from a fixed dataset $\bar x_{1:n_{pre}}$ using all pairs. Since there are only two arguments, their proof closely resembles [12], ignoring dependence on $k$. Crucially, the lack of labels allows **all pairs** to be included, removing the nuance of class-collision. In contrast, our work focuses on the interaction between $k$ and class labels, raising many technical challenges. Meanwhile, [4] emphasizes the link between population classification risk and contrastive risk, as well as the impact of data augmentation, which we do not consider.
>
> **Q1a: Re formula of $\mathcal{\bar D}\_c$.**
>
> **Re**: In Eqn. (2) of [2], the negative distribution is defined as $\mathcal{D}\_\mathrm{neg}(x)=\mathbb{E}\_{c\sim\rho}[\mathcal{D}_c(x)]$, which allows negative samples to be drawn from any class regardless of the anchor's class, effectively **allowing class-collision**. In our definition, class-collision is excluded, resulting in the definition in Eqn. (2) of our work. Disallowing class collision is associated to better performance [2, 8, 9].
>
> **Q1b: Are results affected by Eqn. (2)?**
>
> **Re**: Yes, our definition of negative distributions results in a class-collision free version of the population risk in Eqn. (3). Thus, our results concern the concentration of empirical risks around this version. However, our techniques readily extend to class-collision case: we believe the quantity $\widetilde N$ (Thm. 5.1 and 5.2) would then be replaced by $N\min(\rho_{\min}/2,1/2k)$ and we expect the results on the estimation error to be marginally tighter (although the practical performance would suffer from reduced alignment between the unsupervised loss and classification objective [2, 9]).
>
> **Q2: Re Sections 3.2 and 3.3.**
>
> **Re**: Our Sec. 3 compares the tuples sampling methods in [2] and our work. Under [2] (Sec 3.2), the tuples are drawn i.i.d. from a distribution over $\mathcal{X}^{k+2}$, i.e., each whole tuple is observed independently as the reviewer stated and no data reuse occurs. Under our framework (Sec. 3.3), the procedure in P. 3 is used to select tuples from a labeled dataset $\mathcal{S}$ drawn i.i.d. from a distribution over $\mathcal{X}\times\mathcal{C}$. Hence, while **independence holds for elements of $\mathcal{S}$, it does not hold for the selected tuples**. For instance, if $k=1, \mathcal{S}=(x,y,v,z)$ with labels $1,1,1,2$, then both $(x,y,z)$ and $(u,v,z)$ are valid tuples but they are dependent due to a shared negative sample $z$. This process allows data reuse across tuples, adhering to practice [7] where recycling samples is desired to create large tuples datasets.
>
> **Q3: Why is $S$ labeled?**
>
> **Re**: As clarified above, our work indeed targets **supervised contrastive learning** rather than truly unsupervised settings. We promise to emphasize this in the revision.
>
> We thank you again for your suggestions, which we believe will help us better frame our contribution in a broader context beyond [2, 3]. We promise to thoroughly discuss all the proposed references.  If the reviewer and AC deem it appropriate, we are open to changing the title to `Generalization Analysis for supervised contrastive learning in non i.i.d. settings'. We hope our rebuttal has helped improve your opinion of our contributions. Please do not hesitate to let us know if you have any further questions.

---

> > ### Comment · Reviewer_t8eS · 2025-04-02
> >
> > Thanks for the authors' response. I still have some doubts. I have understood the authors aim to study the theoretical performance of supervised CRL. What puzzles me the most is that the authors didn't explain why they mentioned that they proposed some bounds of unsupervised risk.  Besides, CRL was proposed to deal with the case where label information is not available. if label information is available, why don't we conduct the common supervised learning but supervised CRL? Can the authors provide some examples to illustrate that in certain situations, the performance of supervised CRL is greater than supervised learning? Finally, I have also listed some different viewpoints from your response as follows.
> >
> > **1.** The authors stated that the sampling process of [2,3] cannot be simulated without labels. I don't agree this viewpoint. The sampling process of [2,3] is most realistic when data augmentation doesn't alter the original semantics of samples. It doesn't need the real label of samples but the consistency of the latent labels (semantics) of $x,x^+$. Compared with [2,3], the sampling process of this work is not realistic for unsupervised CRL.
> >
> > **2.** The **Re** of **Q1a** stated that the negative distribution defined in [2] allows negative samples to be drawn from any class regardless of the anchor's class, allowing class collision. I agree with it since, in practice, we cannot ensure the latent labels of negative samples are different from their corresponding anchor sample without real label information. In the supervised CRL setting, the negative sampling process of this paper can be achieved.
> >
> > **3.** The **Re** of **Q2** stated that each whole tuple in [2] is observed independently. Then, why do the authors ensure that the sampling method of [2] has no data reuse? From my perspective, the simple example the authors provided is also likely to occur in the case of [2].

---

> > > ### Author Response · Authors · 2025-04-02
> > >
> > > Dear reviewer,
> > >
> > > **Many thanks for your very fast response**. To the best of our understanding, the discussion will unfortunately be ended by OpenReview after this message. We try our very best to **clarify** the **remaining misunderstandings** within the character limit and are ready to answer further questions via other proposed means if necessary. In short, we respectfully stand by our rebuttal and are very grateful in advance for your time reading our detailed answer below and our original rebuttal. Note that the references are in our answer to reviewers 81JF/bLbY.
> > >
> > >
> > >
> > > **Re: Why supervised CRL but not common supervised learning directly and practical examples of supervised CRL.**
> > >
> > > We apologize for not making this clearer in the last sentence of the third paragraph of our rebuttal. **Yes**, the practical performance of supervised CRL can be superior. This phenomenon is explicitly described and thoroughly documented for CIFAR10, CIFAR 100 and ImageNet (see Figure 3, Figure 4 and Table 2) as one of the main messages in the paper "Supervised contrastive learning" [7] which has more than 5000 citations. In addition, our own experiments (cf. Figure 2 in our paper) corroborate this fact on the MNIST dataset.
> > > Like the rest of the literature, we do not claim to capture this phenomenon theoretically, and instead prove bounds on the generalization error of the unsupervised risk, an important and already **highly non-trivial** step in the study of supervised CRL.
> > >
> > > **Re (Comment 1)**: Whilst it is not impossible that the original intention in [2,3] may have been to create a simplified model which could metaphorically illustrate the performance of unsupervised CRL in the presence of data augmentation, mathematically, it is a fact that both papers make assumptions which are extremely close to assuming the presence of labels. The results are inapplicable to the "practical" case of data augmentation.
> > >
> > > To clarify this, note that there is **no data augmentation** in either [2] or [3], and the results of [2], which establish a relationship between the supervised and unsupervised risks, explicitly assume and **mathematically rely on** the fact that the anchor $x$ and positive $x^+$ samples are not only sampled from the same class but sampled **independently of each other**. Thus, the results in [2] **do not work** for data augmentation: it is **not enough** for the "**semantics**" to be preserved, what is required is the ability to draw a new sample $x^+$ from the conditional distribution over the class of $x$ with absolutely no other dependence on $x$ other than through the label. We do not think this is in any way more "realistic" than assuming the presence of labels (if the sampling of $x^+$ conditionally given $x$, which corresponds to the data augmentation step in the references you provide, can be disassociated from the draw of $x$ and applied indefinitely, then it implies the ability to draw indefinitely many samples from the class of $x$).
> > >
> > > The **assumption is present** in Eqn. (1) of [2] and at the beginning of the "problem formulation" in [3] (the distribution $\mathcal{D}_{sim}$ is factorized into independent components for $x$ and $x^+$). A similar assumption is also made in the analogous 'block' setting in Section 6.3 of [2].
> > >
> > > In addition, the **proofs** of Theorem 4.1 (via Lemma 4.3),  and Proposition 6.2 **all rely on the independence between $x$ and $x^+$** when using Jensen's inequality: the key step (see step (b) in the proof of Lemma 4.3 or similarly, the first line of the proof of Proposition 6.2 in the appendix) relies on moving the expectation over $x^+,x^-$ (but not the expectation over $x$) inside the loss function formulation $\ell(f(x)^\top [f(x^+)-f(x^-)])$. This is possible pointwise for all fixed values of $x$, whose expectation remains outside the loss function because the function $\mathbb{R}^2\ni(f(x^+),f(x^-))\mapsto\ell(f(x)^\top[f(x^+)-f(x^-)])$ is convex in $(f(x^+),f(x^-))\in\mathbb{R}^2$ for any fixed value of $f(x)$. It is not possible to perform the same step if the distributions of $x$ and $x^+$ are dependent (as in the case of augmentation) since the function $\mathbb{R}^3\ni(f(x),f(x^+),f(x^-))\mapsto\ell(f(x)^\top [f(x^+)-f(x^-)])$ is not convex in $(f(x),f(x^+),f(x^-))\in\mathbb{R}^3$.
> > >
> > > **Re (Comment 3)**: Thank you for your comment. We believe you are referring to the reuse of the same samples in different tuples. In short, [2,3] require **independence** over tuples, which does not hold in our setting because of the reuse of the same fixed empirical pool of samples $x_1,\ldots,x_N$ to construct many tuples. In the setting of [2,3], **if the population distribution over samples is finite**, it is indeed possible for the same samples to **coincidentally** appear in the same tuple despite independence. If the distribution is non-atomic, this will happen with probability zero. In both cases, it is both far less likely and a distinct question from independence.

---

### Official Review · Reviewer_bLbY · 2025-03-08

**Overall Recommendation:** 4

**Summary:**

This paper provides the first statistical generalization theory of Contrastive Representation Learning (CRL) where data does not follow the strong assumption of independently and identically distributed (i.i.d.). Previous theoretical research on CRL assumes that data points used for training are drawn independently. However, this is not true in real-life applications. In reality, datasets are often created by reusing labeled samples across multiple tuples via resampling. Such kind of reuse invalidates the independence assumption, making existing theoretical analyses less applicable to real-world scenarios.

To address this issue, the paper introduces a revised statistical theory framework that applies to non-i.i.d. data. It establishes generalization bounds for CRL using statistical techniques from U-Statistics, showing that the required number of samples per class depends on the logarithm of the covering number of the learnable feature space. The paper formulates the population risk in CRL as a U-statistics problem and prove that the generalization gap between the empirical minimizer and the Bayes risk decreases at a rate that depends on the dataset size and the structure of the contrastive loss. This provides a more rigorous understanding of how generalization behaves under data reuse. The authors also derive generalization bounds for empirical risk minimization under both U-Statistics and subsampled risk settings, demonstrating how these bounds change with dataset size and subsampling factors. Finally, the paper applies its theoretical findings to common function classes such as linear models and neural networks. By providing a more realistic theoretical foundation, the study offers insights into how CRL generalizes when data is limited and frequently reused.

## update after rebuttal

The authors have addressed all my questions in my original review. I thus decide to maintain my original rating of 'accept'.

**Claims And Evidence:**

I think all the four major claims in the introduction are supported by theoretical results.
Specifically, there are two main technical results. The first one is the generalization bound for the empirical minimizer of U-statistics. This claim is supported by section 5.1 and the detailed analysis is presented in Appendix B.2. The second one is Thm 5.2 in Section 5.2, with detailed proof included in Appendix C.2.

I read the proof of the main results in Appendix B and C and skimmed through the technical lemmas. It seems to me that the claims are supported.

**Essential References Not Discussed:**

From my opinion the most relevant works are discussed in the Related Work section.

**Experimental Designs Or Analyses:**

N/A

**Methods And Evaluation Criteria:**

N/A. This is a purely theoretical work that proposes new statistical generalization analysis for contrastive representation learning.There is no empirical evaluation.

**Other Comments Or Suggestions:**

N/A

**Other Strengths And Weaknesses:**

**1** Clarity: I think this paper is clear. The claims are clear to understand. As a theoretical work, the section of proof strategy significantly helps readers like me to better understand the major steps in the analysis.

**2** Originality: this paper is the first theoretical work in generalization theory of contrastive learning.

**Questions For Authors:**

Can the results or analysis framework of the paper be extended to semi-supervised contrastive representation learning?
To be specific, in many use case such as AI for Health (e.g. segmentation task of medical images), practitioners often adopt semi-supervised learning where the total loss consists of a contrastive loss and a supervised loss.
So I guess to what extent does the major results depend on the unsupervised risk formulation in Definition 3.1? Can it be of other forms? Thanks!

**Relation To Broader Scientific Literature:**

This paper belongs to the scientific area of statistical generalization theory, where advanced statistical methods are used to analyze the accuracy and robustness of statistical and machine learning models. It specifically focuses on contrastive representation learning. I think this paper is a good complement to this area. Specifically, the previous theoretical work in this area all follows the framework of Arora et al. 2019 which assumes the ideal setting of i.i.d. Data tuples. However, in CRL use case, this often does not hold. This paper relaxes this strong assumption via a U-statistics technique. Therefore, I think this paper makes a valid contribution to its field.

**Theoretical Claims:**

Yes. I read section 4 on the proof strategy. I also skimmed Appendix B.2, where I focused on the proof of the main result in Thm B.10. I checked the proof of Thm C.1.
I did not check the details of the proof for other auxiliary lemmas.

---

> ### Author Rebuttal · Authors · 2025-03-31
>
> Dear reviewer, we truly appreciate your **recognition of the strength and novelty** of our contributions. We are also particularly grateful for your detailed review and the effort you put into carefully **checking our technical proofs** in Appendices B and C.
>
> To answer your question - yes - the techniques presented in our paper can be useful in semi-supervised learning depending on the unsupervised risk formulation. Below, we list several examples.
>
> **Overview**: Suppose that there exists a distribution $\mathcal{P}$ over $\mathcal{X}\times\mathcal{C}$ where $\mathcal{X}$ denotes the data space and $\mathcal{C}$ denotes the labels space (which is accessible to the learner). In semi-supervised classification, two sets of data $S=\\{(x\_j, y\_j)\\}\_{j=1}^{n}\sim\mathcal{P}^n$ and $S_U = \\{(u\_j, \bar y\_j)\\}\_{j=1}^m\sim\mathcal{P}^m$ are given. While both $S, S_U$ are drawn i.i.d. from $\mathcal{P}$, the labels $(\bar y_j)_{j\in[m]}$ are hidden, making $S_U$ unlabelled. In general, the overall risk for a representation functions $f\in\mathcal{F}$ is a combination of both supervised and unsupervised risks. As the supervised risk has been well-studied [13, 16, 17], we focus on possible formulations of unsupervised risk below instead.
>
> **Consistency regularization [14, 15]**: Under this regime, the unsupervised loss aims to ensure consistency of representations under different augmentations of inputs. Specifically, let $\mathcal{A}$ be a distribution over augmentation schemes, the unsupervised risk can be defined as
> $$L_\mathrm{un}(f)=\mathbb{E}_{\substack{u\sim\mathcal{P} \\\ \alpha,\alpha^+\sim\mathcal{A}^2}}\Big[\\|f(\alpha[u])-f(\alpha^+[u])\\|\Big]$$
>
> where $\\|\cdot\\|$ is a distance measure in $\mathbb{R}^d$. Under this regime, we are implicitly given a set of augmented pairs $S_U^\mathrm{aug}=\{(\alpha_j[u_j], \alpha_j^+[u_j])\}_{j=1}^{m}$  drawn i.i.d. from a distribution dependent on both $\mathcal{P, A}$ (where $a_j$'s are drawn i.i.d. from $\mathcal{A}$). Here, the loss function only relies on augmented views $\alpha(x)$ and $\alpha^+(x)$ which comes from the *same* natural sample, which precludes reusing natural samples in different pairs/tuples. Thus, we can analyze this regime using standard results in learning theory without the decoupling technique.
>
> **Self-supervised CL [4, 5, 6]**: In this case, the unsupervised risk can be defined as
> $$L\_\mathrm{un}(f)=\mathbb{E}\_{\substack{u, u^-\_{1:k}\sim\mathcal{P}^{k+1}\\\ \alpha,\alpha^+, \beta\_{1:k}\sim\mathcal{A}^{k+2}}}\Big[\ell\Big(\Big\\{f(\alpha[u])^\top[f(\alpha^+[u])-f(\beta\_i[u\_i^-])]\Big\\}_{i=1}^k\Big)\Big].$$
>
> Intuitively, the augmented views of the same input should be similar to each other while being dissimilar to augmented views of other samples. Since $S_U$ consists of i.i.d. data points in this setting, it is natural to formulate a $(k+2)$-order U-Statistics from $S_U$, generalizing our results to this setting. While such a learning setting involves reused samples, the lack of supervised information removes much of the subtleties of class-collision in our work. Thus, the analysis would be close to (a $k$-wise extension of) [12], making the result simpler than our case when class constraints are present.
>
> **Extension**: Extension to semantic segmentation is natural: we then have $\mathcal{X}\subseteq\mathbb{R}^{C\times H\times W}$, $\mathcal{C}\subseteq \\{0, 1\\}^{C\times H\times W}$ where $C, H, W$ represents image channels, height, width and $\mathcal{F}$ is a class of CNNs. Under consistency regularization, $\\|\cdot\\|$ can be the MSE loss between tensors. Under self-supervised contrastive learning, the contrastive loss can be any common loss functions (hinge/logistic).
>
> Lastly, as explained in the response to Reviewer t8eS, we note that our results concern the *excess unsupervised risk*: the connection between the unsupervised (population) risk and the (population)  downstream classification risk is an orthogonal line of research which has been studied in various works [4, 5, 6].
>
> **Conclusion**: Once again, we thank the reviewer for the interesting suggestion that extends our work to potential new directions. In our revised manuscript, we promise to discuss such extension to semi-supervised learning at length and propose future directions accordingly. Please do let us know if you have any further questions or points to discuss.
>
> **References**:
>
> [12] Clemencon et al., Ranking and Empirical Minimization of U-Statistics. Ann. Statist., 2006.
>
> [13] Bartlett et al., Spectrally-normalized Margin Bounds for Neural Networks. NIPS, 2017.
>
> [14] Ting Chen et al., A Simple Framework for Contrastive Learning of Visual Representations. ICML, 2020.
>
> [15] Sajjadi et al., Regularization With Stochastic Transformations and
> Perturbations. NIPS, 2016.
>
> [16] Long \& Sedghi, Generalization bounds for deep convolutional neural networks. ICLR, 2020.
>
> [17] Golowich et al., Size-Independent Sample Complexity of Neural Networks. COLT, 2018.

---

### Official Review · Reviewer_81JF · 2025-03-14

**Overall Recommendation:** 3

**Summary:**

For the generalization analysis of contrastive representation learning (CRL) in non-IID settings, several new theory bounds are proposed in this paper. First, this paper proposes a revised theoretical framework for CRL. Then, a U-Statistics formulation for the population unsupervised risk is proposed, and bounds for the empirical minimizer of U-Statistics and a sub-sampled risk are derived. Finally, excess risk bounds for classes of linear functions and neural networks are derived based on the above theoretical results.

**Claims And Evidence:**

$\bullet$ This paper mainly proposes the claim: Training datasets are often limited to fixed pools of labeled samples, previous results on generalization bounds for CRL might not comply with most practical use cases where data is limited.

$\bullet$ The theoretical results in this paper provide strong and clear evidence for the claim.

**Essential References Not Discussed:**

No. All the essential references have been adequately discussed.

**Experimental Designs Or Analyses:**

Yes. The experimental designs used to verify the theoretical results and the analyses on the theoretical results are all valid.

**Methods And Evaluation Criteria:**

Yes. The analytical method on U-Statistics provides effective support for theoretical analysis for CRL under non-IID settings.

**Other Comments Or Suggestions:**

Typos: Line 321, "$O(1/\sqrt{n})$" --> "$\tilde{O}(1/\sqrt{n})$".

**Other Strengths And Weaknesses:**

$\bullet$ Strengths:

The theory of U-Processes is often considered to deal with pairwise learning or ranking learning. Using it to deal with pairwise relationships between anchor-positive pairs and tuples of negative samples is a novel perspective, which provides a good example of the potential application and development of U-Statistics.

$\bullet$ Weaknesses:

1. The discussion on reducing the dependency of the generalization bounds on $k$ is insufficient. Previous work used the Lipschitz continuity of the loss functions with respect to the $\ell_\infty$ norm or the self-bounding Lipschitz continuity (essentially smoothness) to reduce the dependency on $k$ from square-root to logarithmic. When the label distribution is perfectly balanced, the discussion in lines 314 to 326 is that due to $N=nk$, the bound here is similar to the $\tilde{O}(1/\sqrt{n})$ bounds in previous works. However, in this case, the bound here is tighter than the existing $\tilde{O}(1/\sqrt{n})$ bounds by a logarithmic factor of $k$, What is the reason for the tightness of this logarithmic factor?

2. Regarding the reasons why formulating the (k+2)-order U-Statistics in the case of CRL is not straightforward, it seems to me that the two points listed are actually stating the same thing together, and I suggest that the two points can be combined or that some appropriate explanation should be added to the first point.

**Questions For Authors:**

Please refer to Weaknesses.

**Relation To Broader Scientific Literature:**

Compared with previous works which require an IID assumption across input tuples, the theoretical results in this paper for non-IID settings are more consistent with practical situations, and these theoretical results broaden the limitations of existing works.

**Theoretical Claims:**

Yes. I have reviewed most of the proofs and the theoretical claims are correct.

---

> ### Author Rebuttal · Authors · 2025-03-30
>
> Many thanks for your thorough review. We especially appreciate your efforts in checking the correctness of our theoretical analysis as well as raising an interesting question regarding the proof techniques. Please refer to our response below regarding the difference between the technicality of our work and that of [3].
>
> **Re (Weakness 1)**: Many thanks for your keen observation! Indeed, our Theorem 5.1 is free of any logarithmic factors in either $N$ or $k$. However, the right hand side involves the quantities $K_{\mathcal{F},c}$, which each bounds the Rademacher complexity of the loss class associated to tuples corresponding to positive class $c$: applying the bounds to concrete function classes, as we do in Appendix B.3, does incur logarithmic factors of both $k$ and $N$ when bounding these complexities via covering numbers and Dudley's entropy formula.
>
> In contrast, bounds in [3] incur logarithmic factors as early as in Theorem 4.8, even though the Rademacher complexities $\mathfrak{R}\_{S_{\mathcal{H}},nk}(\mathcal{H})$ have yet to be bounded. This is because [3] relies on inequalities between different complexity measures (Lemma C.3 in [3]) earlier in the proof. Still, since our analysis on concrete function classes subsequently incurs log factors, we have chosen not to dwell on this particular improvement to not obscure our main contributions.
>
> **Details on Proof Technique**: The core similarity of our work and [3] is the fact that we both bound the (empirical) Rademacher complexity of the loss class, denoted as $\mathfrak{\widehat R}\_{S}(\mathcal{G})$ through the $L_\infty$ covering number of $\mathcal{H}=\\{(x, x^+, x^-)\mapsto f(x)^\top[f(x^+) - f(x^-)]:f\in\mathcal{F}\\}$. Specifically, suppose that the loss is $\ell^\infty$-Lipschitz with constant $\eta>0$:
> $$
> \mathfrak{\widehat R}\_S(\mathcal{G}) \lesssim \int_\alpha^\mathcal{M}\sqrt{\frac{\ln \mathcal{N}(\mathcal{H}, \epsilon/\eta, L_\infty(S_\mathcal{H}))}{n}}d\epsilon, \quad \alpha>0.
> $$
>
> Where $S$ is a set of $n$ independent tuples and $S_\mathcal{H}$ is the set of triplets incurred from $S$ (See [3], Section 4.2). At this point, the difference lies in the methods by which we estimate $\mathcal{N}(\mathcal{H}, \epsilon/\eta, L_\infty(S_\mathcal{H}))$. In our work, the $L_\infty$ covering number of $\mathcal{H}$ is **directly estimated** by the $L_{\infty, 2}$ covering number of $\mathcal{F}$ (Lemma B.12). In [3], the estimation is done via fat-shattering dimension and worst-case Rademacher complexity. Below, we show how the additional costs in terms of $n, k$ and their logarithmic terms propagate through complexity measures:
> $$
> \mathfrak{R}\_{WC}(\mathcal{H})^2 \xrightarrow{nk/\epsilon^2} \mathrm{fat}\_{\epsilon}(\mathcal{H}) \xrightarrow{\ln^2(nk/\epsilon^2)} \ln\mathcal{N}\_\infty(\mathcal{H}, \epsilon, L\_\infty(S_\mathcal{H})),
> $$
>
> where $\mathfrak{R}\_{WC}(\mathcal{H})$ denotes the worst-case Rademacher complexity of $\mathcal{H}$ (note that $\mathfrak{R}\_{WC}(\mathcal{H})\in\widetilde O(1/\sqrt{nk})$) and $\mathrm{fat}\_\epsilon(\mathcal{H})$ denotes the fat-shattering dimension. This yields the final inequality (See [3], Eqn. (C.6)):
> $$
> \ln\mathcal{N}(\mathcal{H}, \epsilon/\eta, L_\infty(S_\mathcal{H}))\lesssim \frac{nk\eta^2\mathfrak{R}^2\_{WC}(\mathcal{H})}{\epsilon^2}\ln^2\Bigg(\frac{\eta^2 nk}{\epsilon^2}\Bigg).
> $$
>
> **Re (Weakness 2)**: We thank the reviewer for their suggestion. In our revised manuscript, we will certainly re-phrase the subtlety of formulating a $(k+2)$-order U-Statistics more succinctly.
>
> **Conclusion**: Aside from the concerns addressed above, we promise to correct any other existing typos in our revised manuscript. We thank the reviewer once again for their effort in making our work clearer for readers.
> We stand ready to answer any further questions you may have and hope that our answer has helped further improve your opinion of our work.
>
> **References**:
>
> [2] Arora et al., A Theoretical Analysis of Contrastive Unsupervised Representation Learning. ICML, 2019.
>
> [3] Lei et al., Generalization Analysis for Contrastive Representation Learning. ICML, 2023.
>
> [4] HaoChen et al., Provable Guarantees for Self-Supervised Deep Learning with Spectral Contrastive Loss. NeurIPS, 2021.
>
> [5] Wang et al., Chaos is a ladder: A new theoretical understanding of Contrastive Learning via augmentation overlap. ICLR, 2022.
>
> [6] Huang et al., Towards the generalization of contrastive Self-supervised Learning. ICLR, 2023.
>
> [7] Khosla et al., Supervised Contrastive Learning. NeurIPS, 2020.
>
> [8] Awasthi et al., Do More Negative Samples Necessarily Hurt In Contrastive Learning? ICML, 2022.
>
> [9] Ash et al., Investigating the Role of Negatives in Contrastive Representation Learning. AISTATS, 2022.
>
> [10] Schroff et al., FaceNet: A Unified Embedding for Face Recognition and Clustering. CVPR, 2015.
>
> [11] Sohn, Improved Deep Metric Learning with Multi-class N-pair Loss Objective. NIPS, 2016.

---

### Official Review · Reviewer_sw2u · 2025-03-16

**Overall Recommendation:** 3

**Summary:**

This paper revisits contrastive representation learning by relaxing the traditional i.i.d. assumption on training tuples. Instead of assuming independent tuples, the authors analyze a practical setting where a fixed pool of labeled data is recycled to form multiple tuples. They derive generalization bounds using a U‑Statistics framework for both the full risk and a sub-sampled empirical risk. They applied the obtained resutls to linear models and neural networks, and experiment on MNIST and synthetic data.

**Claims And Evidence:**

Yes.

**Essential References Not Discussed:**

I don't believe there are essential related works missing from the paper's citations.

**Experimental Designs Or Analyses:**

I checked the to experiments in section 6.1 on how a method based on sub-sampling strategy can effectively approximate the performance of the all-tuple method, and the experiments in section 6.2 showing that the number of samples needed scales linearly with the number of classes and negative samples.

**Methods And Evaluation Criteria:**

Yes.

**Other Comments Or Suggestions:**

For eq (3) I think it's better to reply what is k before eq 3. You mentioned that in the introduction, but I think it's better to restate it before equation 3 instead of before equation 5.

Line 141, it is written "where $\mathcal{S}$..." but $\mathcal{S}$ does not appear in the equation above (eq 5).

typo: C[m]  in equation 11 not defined

**Other Strengths And Weaknesses:**

Strengths:
- The paper extends the theoretical analysis of contrastive representation learning by explicitly handling the non-i.i.d. nature of real-world data developing a framework based on U‑Statistics and decoupling techniques.
- The experiments (although conducted on MNIST and synthetic datasets) demonstrate that the proposed sub-sampling regime can approximate the performance of using all possible tuples, thereby validating the theoretical claims.

Weaknesses:

- The clarity of the paper’s exposition is not as strong as it could be. The presentation sometimes lacks sufficient intuition or detailed explanation of certain steps, particularly in the derivations involving U‑Statistics and decoupling techniques
- One limitation is that treating the tuples as they where independent, as discussed in 4.1, can be done only if we use the tuple sampling strategy described in section 3.3. In a real-world scenario the within-tuple selections may not be  independent.
Of course the dependency is negligible when the number of available samples is very large.
- While the paper describes the overall approach, details on experimental setup are missing.

**Questions For Authors:**

1) Question regarding remark 4.2: In your decoupling argument, you partition the data into disjoint blocks to form the U‑Statistics. Could you please clarify how the symmetry of the kernel ensures that we can remove the sum over q? and write U_{n} in that way as per remark 4.2?  I guess the writing is true only in expectation.

**Relation To Broader Scientific Literature:**

The authors overcome the non-i.i.d. issue by reformulating the risk estimation problem in terms of U‑Statistics and then applying decoupling techniques.
More in detail: instead of computing the empirical risk over tuples of iid samples, they express the population unsupervised risk as a U‑Statistic. This formulation averages the contrastive loss over all valid tuples that can be formed from the dataset. Although these tuples are generated by reusing the same data points, the U‑Statistics framework provides an unbiased estimator of the true risk.
To handle the dependencies, the authors use a decoupling technique inspired by works of Peña and Ginè. The key idea is to break down the U‑Statistic into sums over independent blocks.
They reorganize the sum over all possible tuples into sums over several groups (or blocks) where, by design, each block consists of tuples that are constructed in a disjoint manner from the fixed dataset.

**Theoretical Claims:**

Yes, I have some questions on Remark 4.2 (see questions)

---

> ### Author Rebuttal · Authors · 2025-03-30
>
> Many thanks for the thoughtful and detailed response. We sincerely appreciate your recognition of the novelty in our decoupling techniques. Below, we provide our responses to address your concerns.
>
> **Re (Remark 4.2)**: To clarify your concern, suppose we have an one-sample, $m$-order U-Statistic as defined in Eqn. (12) and pick an arbitrary integer $i\in[q]$ where $q=\lfloor n/m\rfloor$. Additionally, denote $P_m[n]$ and $C_m[n]$ as the sets of $m$-permutations and $m$-combinations selected from $[n]$, respectively. Then, for any $l_1, \dots, l_m \in P_m[n]$, there are $(n-m)!$ re-arrangements $\pi\in S[n]$ (line 220) such that $\pi[mi-m+u]=l_u, \forall u\in[m]$. In other words, when we shuffle $[n]$ brute-forcedly, the tuple $l_1, \dots, l_m$ appears $(n-m)!$ times (in exact order) from position $mi-m+1$ to $mi$. Hence, we can write:
> $$
>     \frac{1}{n!}\sum_{\pi\in S[n]} h(x_{\pi[mi-m+1]}, \dots, x_{\pi[mi]}) = \frac{1}{n!}\sum_{l_{1:m}\in P_m[n]}(n-m)!h(x_{l_1}, \dots, x_{l_m}).
> $$
>
> When $h$ is symmetric, any permutation of its arguments does not alter its value. Thus, we can further simplify the right-hand-side to a sum over the set of $m$-combinations:
> $$
>     \sum_{l_{1:m}\in P_m[n]}h(x_{l_1}, \dots, x_{l_m}) = \sum_{p_{1:m}\in C_m[n]}m! h(x_{p_1}, \dots, x_{p_m}).
> $$
>
> Combining all of the above:
> $$
> \frac{1}{n!}\sum_{\pi\in S[n]} h(x_{\pi[mi-m+1]}, \dots, x_{\pi[mi]}) = \frac{m!(n-m)!}{n!}\sum_{p_{1:m}\in C_m[n]} h(x_{p_1}, \dots, x_{p_m}),
> $$
>
> which is exactly Eqn. (11).
>
> **Re (Weakness 1)**: We thank the reviewer for the suggestion to improve our presentation. To summarize, our U-Statistics formulation is motivated by the task of creating an (asymptotically) unbiased estimator for $\mathrm{L_{un}}(f)$ by estimating each conditional risk $\mathrm{L_{un}}(f|c)$ then combining the estimators, resulting in Eqn. (8). The reason for this separation lies in the nuance of estimating $\mathrm{L_{un}}(f)$ directly through a $(k+2)$-order U-Statistics (Lines 240 - 245). We acknowledge that we can make the intuition clearer for the readers and we will make sure to improve our presentation in the revised manuscript.
>
> **Re (Weakness 2)**: Thank you for the keen observation. In the procedure described in Section 3.3, the tuples are subsampled independently from the pool of 'valid' tuples where samples are unique within tuples. This means that the samples within each tuple are dependent conditionally given the full sample $x_1,\ldots,x_N$ (but independent without condition), whilst the tuples are independent with or without conditioning. **This approach corresponds to calculating U-statistics**.
>
> It is also possible to allow replacement within tuples, in which case the samples within each tuple would instead be independent conditioned on the full sample $x_1,\ldots,x_N$ but dependent if no conditioning is applied. **This approach corresponds to calculating V-statistics**. Our techniques readily extend to this case and would only incur an additive term of $\widetilde{O}\Big(\frac{1}{ N\min(\rho_{\min}/2, (1-\rho_{\max})/k)}\Big)$, which does not worsen the order of magnitude in our bounds. To see why, let us consider the one-sample case (described before Eqn. (11)) for simplicity. The V-Statistics of order $m$ for some kernel $h$ is defined as:
> $$
>     V_n(h)=\frac{1}{n^m}\sum_{i_1=1}^n\dots\sum_{i_m=1}^n h(x_{i_1}, \dots, x_{i_m}).
> $$
>
> Generally, $V_n(h)$ is not an unbiased estimator of $\mathbb{E}h(x_1, \dots, x_m)$ due to the terms with repeated samples. By [1], we can write $V_n(h)$ as follows:
> $$
>     n^m V_n(h) = \binom{n}{m}U_n(h) +\sum_{i_{1:m}\in[n]^m\setminus C_m[n]}h(x_{i_1}, \dots, x_{i_m}).
> $$
>
> As a result:
> $$
> V_n(h) - U_n(h) = \frac{1}{n^m} \sum_{i_{1:m}\in[n]^m\setminus C_m[n]}[h(x_{i_1}, \dots, x_{i_m}) - U_n(h)],
> $$
>
> where $|[n]^m\setminus C_m[n]|\in O(n^{m-1})$.  Therefore, if $|h(x_1, \dots, x_m)|\le M$, then $V_n(h)-U_n(h)\in O(M/n)$, which means the effect of biasedness dissipates when $n$ increases (as the reviewer remarked). Thus, while biasedness slightly complicates analysis, we can leverage results on the concentration of U-statistics to study the concentration of V-statistics around the population risk, with a small incurred cost. Once again, we really appreciate the reviewer's feedback that help extend the scope of our work to a very interesting regime.
>
> **Other Modifications**: As per reviewer's suggestions, we have made the following changes:
>
> - Reiterate the definition of $k$ in definition 3.1 (Eqn. (3)).
>
> - Move the definition of the i.i.d. dataset $S$ before Eqn. (5) for better flow.
>
> - Define $C_m[n]$ before Eqn. (11).
>
> Furthermore, we promise to add a more detailed experiment set-up description in the appendix as well as a pointer to the appendix in the main text. We thank the reviewer once again for their efforts in making our work clearer for readers.
>
> **References**:
>
> [1] Wassily Hoeffding, A Class of Statistics with Asymptotically Normal Distribution. Ann. Math. Statist., 1948.

---

> > ### Comment · Reviewer_sw2u · 2025-04-07
> >
> > Thank you for your detailed response. I appreciate that you addressed almost all of my concerns.
> > I still have one doubt, my question was not about the equality between eq. (11) and eq. (12), but rather on why $U_n$ can be written exactly as stated in Remark 4.2.

---

> > > ### Author Response · Authors · 2025-04-08
> > >
> > > We are delighted that we have addressed almost all of your concerns. We are very grateful for your feedback and believe our scientific discussion will help us further improve the presentation and impact of our work.
> > >
> > > Regarding your follow-up question, we understand that you are seeking clarification as to why the sum over $q$ disjoint blocks in Eqn. (12) is no longer present in Remark 4.2. We apologize for the confusion. As we should have stated more explicitly, the original calculation in our rebuttal is indeed meant to establish the equality between the expression in Remark 4.2 (which is the starting point of the calculation) and the U-Statistics formula in Eqn. (11).
> > >
> > > To summarize concisely, for all $1\le i \le q$ where $q=\lfloor n/m\rfloor$, we have
> > >
> > > $$
> > > \begin{align*}
> > >     \underbrace{\frac{1}{n!}\sum\_{\pi\in S[n]}h(x\_{\pi[mi-m+1]},\dots,x\_{\pi[mi]})}\_{\textup{Remark 4.2}} &=\frac{1}{n!}\sum\_{l\_{1:m}\in P\_m[n]}(n-m)!h(x\_{l\_1}, \dots, x\_{l\_m}) \\\ &=\frac{1}{n!}\sum\_{p\_{1:m}\in C\_m[n]} m!(n-m)!h(x\_{p\_1}, \dots, x\_{p_m}) \\\ &=\underbrace{\frac{1}{\binom{n}{m}}\sum\_{p\_{1:m}\in C\_m[n]}h(x\_{p\_1}, \dots, x\_{p\_m})}\_{\textup{Eqn. (11)}}.
> > > \end{align*}
> > > $$
> > >
> > > Intuitively, the expression in Remark 4.2 (**without** the average over $q$ blocks) can be viewed as averaging over all permutations of $n$ samples, where for each permutation, we extract a block of $m$ consecutive elements (starting from position $mi-m+1$ to position $mi$) as inputs to the kernel $h$. As demonstrated above, this is exactly equivalent to averaging over all possible $m$-permutations (or $m$-combinations if $h$ is symmetric) of the $n$ samples, which is precisely the definition of $U_n(h)$ in Eqn. (11). Since this holds for each one of the $q$ disjoint blocks, we remarked that the representation in Eqn. (12) (**with** the average over $q$ blocks) is the same as summing $U_n(h)$ for $q$ times then dividing by $q$ again. Specifically,
> > >
> > > $$
> > > \begin{align*}
> > > \underbrace{\frac{1}{n!}\sum\_{\pi\in S[n]}\frac{1}{q}\sum\_{i=1}^qh(x\_{\pi[mi-m+1]},\dots,x\_{\pi[mi]})}\_{\textup{Eqn. (12)}}
> > >  &= \frac{1}{q}\sum\_{i=1}^q\underbrace{\frac{1}{n!}\sum\_{\pi\in S[n]}h(x\_{\pi[mi-m+1]},\dots,x\_{\pi[mi]})}\_{\textup{Remark 4.2}} \\\ &= \frac{1}{q}\sum_{i=1}^q U_n(h)=U_n(h).
> > > \end{align*}
> > > $$
> > >
> > > We thank you again for your help in improving our manuscript and promise to include a more thorough explanation as above in the revision.

---

### Decision · Program_Chairs · 2025-05-01

**Decision:**

Accept (poster)

**Comment:**

This paper revisits contrastive representation learning by relaxing the traditional i.i.d. assumption on training tuples (Arora et al. (2019) and lei et al. (2023)). In contrast, the authors analyze a practical setting where a fixed pool of labeled data is recycled to form multiple tuples. They derive generalization bounds using a U‑Statistics framework. The authors use a decoupling technique (Peña and Ginè) which breaks down the U‑Statistic into sums over independent blocks.

The reviewers all agree the motivation for this theoretical work is clear and reasonable and it  is more consistent with practical setting, and the results are new.  In the rebuttal, the authors addressed the questions satisfactorily in their rebuttal.  I concur with the reviewers and recommend its acceptance.

I would strongly suggest to incorporate all the suggestions made by the reviewers in the final version.